# Learning to compute Gröbner bases

**Hiroshi Kera**[*]
Chiba University,
Zuse Institute Berlin
kera@chiba-u.jp

**Yuki Ishihara**
Nihon University
ishihara.yuki@nihon-u.ac.jp

**Yuta Kambe**
Mitsubishi Electric
kambe.yuta@bx.mitsubishielectric.co.jp

**Tristan Vaccon**
Limoges University
tristan.vaccon@unilim.fr

**Kazuhiro Yokoyama**
Rikkyo University
kazuhiro@rikkyo.ac.jp

## Abstract

Solving a polynomial system, or computing an associated Gröbner basis, has been a fundamental task in computational algebra. However, it is also known for its notorious doubly exponential time complexity in the number of variables in the worst case. This paper is the first to address the learning of Gröbner basis computation with Transformers. The training requires many pairs of a polynomial system and the associated Gröbner basis, raising two novel algebraic problems: random generation of Gröbner bases and transforming them into non-Gröbner ones, termed as backward Gröbner problem. We resolve these problems with 0-dimensional radical ideals, the ideals appearing in various applications. Further, we propose a hybrid input embedding to handle coefficient tokens with continuity bias and avoid the growth of the vocabulary set. The experiments show that our dataset generation method is a few orders of magnitude faster than a naive approach, overcoming a crucial challenge in learning to compute Gröbner bases, and Gröbner computation is learnable in a particular class.

## 1 Introduction

Understanding the properties of polynomial systems and solving them have been a fundamental problem in computational algebra and algebraic geometry with vast applications in cryptography [8, 93], control theory [71], statistics [27, 41], computer vision [78], systems biology [60], and so forth. Special sets of polynomials called Gröbner bases [16] play a key role to this end. In linear algebra, the Gaussian elimination simplifies or solves a system of linear equations by transforming its coefficient matrix into the reduced row echelon form. Similarly, a Gröbner basis can be regarded as a reduced form of a given polynomial system, and its computation is a generalization of the Gaussian elimination to general polynomial systems. However, computing a Gröbner basis is known for its notoriously bad computational cost in theory and practice. It is an NP-hard problem with the doubly exponential worst-case time complexity in the number of variables [29, 67]. Nevertheless, because of its importance, various algorithms have been proposed in computational algebra to obtain Gröbner bases in better runtime. Examples include Faugère's F4/F5 algorithms [33, 34] and M4GB [66].

---

[*]corresponding author

In this study, we investigate Gröbner basis computation from a learning perspective, envisioning it as a practical compromise to address large-scale polynomial system solving and understanding when mathematical algorithms are computationally intractable. The learning approach does not require explicit design of computational procedures, and we only need to train a model using a large amount of (non-Gröbner set, Gröbner basis) pairs. Further, if we restrict ourselves to a particular class of Gröbner bases (or associated *ideals*), the model may internally find some patterns useful for prediction. The success of learning indicates the existence of such patterns, which encourages the improvement of mathematical algorithms and heuristics. Several recent studies have already addressed mathematical tasks via learning, particularly using Transformers [14, 19, 58]. For example, [58] showed that Transformers can learn symbolic integration simply by observing many $(\mathrm{d}f/\mathrm{d}x, f)$ pairs in training. The training samples are generated by first randomly generating $f$ and computing its derivative $\mathrm{d}f/\mathrm{d}x$ and/or by the reverse process.

However, a crucial challenge in the learning of Gröbner basis computation is that it is mathematically unknown how to efficiently generate many (non-Gröbner set, Gröbner basis) pairs. We need an efficient backward approach (i.e., *solution-to-problem* computation) because, as discussed above, the forward approach (i.e., *problem-to-solution* computation) is prohibitively expensive. To this end, we frame two problems: (i) a random generation of Gröbner bases and (ii) a backward transformation from a Gröbner basis to an associated non-Gröbner set. To our knowledge, neither of them has been addressed in the study of Gröbner bases because of the lack of motivations; all the efforts have been dedicated to the forward computation from a non-Gröbner set to Gröbner basis.

Another challenge in the learning approach using Transformers lies in the tokenization of polynomials on infinite fields, such as $\mathbb{R}$ and $\mathbb{Q}$. To cover a wide range of coefficients, one has to either prepare numerous number tokens or split a number into digits. The former requires a large embedding matrix, and the latter incurs large attention matrices due to the lengthy sequences. To resolve this, we introduce a continuous embedding scheme, which embeds coefficient tokens by a small network and avoids the tradeoff between vocabulary size and sequence length. The continuity of the function realized by the network naturally implements the continuity of the numbers in the embedding.

We summarize the contributions as follows.

- We investigate the first learning approach to the Gröbner computation using Transformers and experimentally show its learnability. Unlike most prior studies, our results indicate that training a Transformer may be a compromise to NP-hard problems to which no efficient (even approximate or probabilistic) algorithms have been designed.

- We uncovered two unexplored algebraic problems—random generation of Gröbner bases and backward Gröbner problem and propose efficient methods to address them in the 0-dimensional case. The problems are essential to the learning approach but also algebraically interesting and need interaction between computational algebra and machine learning.

- We propose a new input embedding to efficiently handle a large range of coefficients without the tradeoff between the size of the embedding matrix and attention maps.

Our experiments show that the proposed dataset generation is highly efficient and faster than a baseline method by a few orders of magnitude. Further, we observe a learnability gap between polynomials on finite fields and infinite fields while predicting polynomial supports are more tractable.

## 2 Related Work

**Gröbner basis computation.** Gröbner basis is one of the fundamental concepts in algebraic geometry and commutative ring theory [24, 39]. By its computational aspect, Gröbner basis is a very useful tool for analyzing the mathematical structures of solutions of algebraic constraints. Notably, the form of Gröbner bases is suited for finding solutions and allows parametric coefficients, and thus, it is vital to make Gröbner basis computation efficient and practical in applications. Following the definition of Gröbner bases in [16], the original algorithm to compute them can be presented as (i) create potential new leading terms by constructing *S-polynomials*, (ii) reduce them either to zero or to new polynomials for the Gröbner basis, and (iii) repeat until no new S-polynomials can be constructed. Plenty of work has been developed to surpass this algorithm. There are four main strategies: (a) avoiding unnecessary S-polynomials based on the F5 algorithm and the more general *signature-based algorithms* [9, 34]. Machine learning appeared for this task in [73]. (b) More efficient

reduction using efficient linear algebraic computations using [33] and the very recent GPU-using [12]. (c) Performing *modular computations*, following [6, 70], to prevent *coefficient growth* during the computation. (d) Using the structure of the ideal, e.g., [13, 35] for change of term ordering for 0-dimensional ideals or [83] when the Hilbert function is known. In this study, we present the fifth strategy: (e) Gröbner basis computation fully via learning without specifying any mathematical procedures.

**Transformers for mathematics.** Recent studies have revealed that Transformers can be used for mathematical reasoning and symbolic computation. The training only requires samples (i.e., problem–solution pairs), and no explicit mathematical procedures need to be specified. In [58], the first study that uses Transformers for mathematical problems is presented. It showed that Transformers can learn symbolic integration and differential equation solving with training with sufficiently many and diverse samples. Since then, Transformers have been applied to checking local stability and controllability of differential equations [22], polynomial simplification [3], linear algebra [19, 20], symbolic regression [14, 26, 44, 45], Lyapunov function design [4] and attacking the LWE cryptography [61, 89]. In [75], comprehensive experiments over various mathematical tasks are provided. In contrast to the aforementioned studies, we found that Gröbner basis computation, an NP-hard problem, even has an algebraic challenge in the dataset generation. This paper introduces unexplored algebraic problems and provides an efficient algorithm for a special but important case (i.e., 0-dimensional ideals), thereby realizing an experimental validation of the learning of Gröbner basis computation.

## 3 Notations and Definitions

We introduce the necessary notations and definitions in this Section. The reader interested in a gentle introduction to Gröbner basis theory can refer to the classical book [25]. For their comfort, we have moreover compiled most elementary additional definitions and notations in App. A.

We consider a polynomial ring $k[x_1, \ldots, x_n]$ with a field $k$ and variables $x_1, \ldots, x_n$. For a set $F \subset k[x_1, \ldots, x_n]$, the ideal generated by $F$ is denoted by $\langle F \rangle$. Once a term order on the terms of $k[x_1, \ldots, x_n]$ is fixed, one can define leading terms and Gröbner bases.

**Definition 3.1** (Leading term). Let $F = \{f_1, \ldots, f_s\} \subset k[x_1, \ldots, x_n]$ and let $\prec$ be a term order. The leading term $\mathrm{LT}(f_i)$ of $f_i$ is the largest term in $f_i$ in ordering $\prec$. The leading term set of $F$ is $\mathrm{LT}(F) = \{\mathrm{LT}(f_1), \ldots, \mathrm{LT}(f_s)\}$.

**Definition 3.2** (Gröbner basis). Fix a term order $\prec$. A finite subset $G$ of an ideal $I$ is said to be a $\prec$-*Gröbner basis* of $I$ if $\langle \mathrm{LT}(G) \rangle = \langle \mathrm{LT}(I) \rangle$.

The condition $\langle \mathrm{LT}(G) \rangle = \langle \mathrm{LT}(I) \rangle$ means that for any element $h \in I$, the leading term $\mathrm{LT}(h)$ is divided by the leading term $\mathrm{LT}(g)$ of an element $g \in G$. It gives a complete test whether a given polynomial $h$ is in $I$ or not by polynomial division with $G$, similar to Gaussian elimination by a basis of a vector space. The remainder is 0 means that $h \in I$, and otherwise means that $h \notin I$. This is related to finding solutions. Roughly speaking, if $h \in I = \langle f_1, \ldots, f_s \rangle$, then we have a form $h = \sum_{i=1}^{s} h_i f_i$ and the system $f_1(x_1, \ldots, x_n) = \cdots = f_s(x_1, \ldots, x_n) = 0$ shares solutions with the equation $h(x_1, \ldots, x_n) = 0$.

Note that $\langle \mathrm{LT}(G) \rangle \subset \langle \mathrm{LT}(I) \rangle$ is trivial from $G \subset I$. The nontriviality of the Gröbner basis lies in $\langle \mathrm{LT}(G) \rangle \supset \langle \mathrm{LT}(I) \rangle$; that is, a finite number of leading terms can generate the leading term of any polynomial in the infinite set $I$. The Hilbert basis theorem [25] guarantees that every ideal $I \neq \{0\}$ has a Gröbner basis. Moreover, using the multivariate division algorithm, one gets that any Gröbner basis $G$ of an ideal $I$ generates $I$. We are particularly interested in the *reduced* Gröbner basis $G$ of $I = \langle F \rangle$, which is unique once the term order is fixed.

**Intuition of Gröbner bases and system solving.** Let $G = \{g_1, \ldots, g_t\}$ be a Gröbner basis of an ideal $\langle F \rangle = \langle f_1, \ldots, f_s \rangle$. The polynomial system $g_1(x_1, \ldots, x_n) = \cdots = g_t(x_1, \ldots, x_n) = 0$ is a simplified form of $f_1(x_1, \ldots, x_n) = \cdots = f_s(x_1, \ldots, x_n) = 0$ with the same solution set. With the term order $\prec_{\mathrm{lex}}$, $G$ has a form $g_1 \in k[x_{n_1}, \ldots, x_n], g_2 \in k[x_{n_2}, \ldots, x_n], \ldots, g_t \in k[x_{n_t}, \ldots, x_n]$ with $n_1 \leq n_2 \leq \ldots \leq n_t$, which may be regarded as the "reduced row echelon form" of a polynomial system. In our particular case (i.e., 0-dimensional ideals in shape position; cf. Sec. 4.2), we have $(n_1, n_2, \ldots, n_t) = (1, 2, \ldots, n)$. Thus, one can obtain the solutions of the polynomial system using

a backward substitution, i.e., by first solving a univariate polynomial $g_t$, next solving bivariate polynomial $g_{t-1}$, which becomes univariate after substituting the solutions of $g_t$, and so forth.

**Other notations.** The subset $k[x_1, \ldots, x_n]_{\leq d} \subset k[x_1, \ldots, x_n]$ denotes the set of all polynomials of total degree at most $d$. For a polynomial matrix $A \in k[x_1, \ldots, x_n]^{s \times s}$, its determinant is given by $\det(A) \in k[x_1, \ldots, x_n]$. The set $\mathbb{F}_p$ with a prime number $p$ denotes the finite field of order $p$. The set $\mathrm{ST}(n, k[x_1, \ldots, x_n])$ denotes the set of upper-triangular matrices with all-one diagonal entries (i.e., unimodular upper-triangular matrices) with entries in $k[x_1, \ldots, x_n]$. The total degree of $f \in k[x_1, \ldots, x_n]$ is denoted by $\deg(f)$.

# 4 New Algebraic Problems

Our goal is to realize Gröbner basis computation through a machine learning model. To this end, we need a large training set $\{(F_i, G_i)\}_{i=1}^m$ with finite polynomial set $F_i \subset k[x_1, \ldots, x_n]$ and Gröbner basis $G_i$ of $\langle F_i \rangle$. As the computation from $F_i$ to $G_i$ is computationally expensive in general, we instead resort to *backward generation* (i.e., solution-to-problem process); that is, we generate a Gröbner basis $G_i$ randomly and transform it to non-Gröbner set $F_i$.

What makes the learning of Gröbner basis computation hard is that, to our knowledge, neither (i) a random generation of Gröbner basis nor (ii) the backward transform from Gröbner basis to non-Gröbner set has been considered in computational algebra. Its primary interest has been instead posed on Gröbner basis computation (i.e., forward generation), and nothing motivates the random generation of Gröbner basis nor the backward transform. Interestingly, machine learning now sheds light on them. Formally, we address the following problems for dataset generation.

**Problem 4.1** (Random generation of Gröbner bases). *Find a collection $\mathcal{G} = \{G_i\}_{i=1}^m$ with the reduced Gröbner basis $G_i \subset k[x_1, \ldots, x_n]$ of $\langle G_i \rangle$, $i = 1, \ldots, m$. The collection should contain diverse bases, and we need an efficient algorithm for constructing them.*

**Problem 4.2** (Backward Gröbner problem). *Given a Gröbner basis $G \subset k[x_1, \ldots, x_n]$, find a collection $\mathcal{F} = \{F_i\}_{i=1}^\mu$ of polynomial sets that are not Gröbner bases but $\langle F_i \rangle = \langle G \rangle$ for $i = 1, \ldots, \mu$. The collection should contain diverse sets, and we need an efficient algorithm for constructing them.*

Problems 4.1 and 4.2 require the collections $\mathcal{G}, \mathcal{F}$ to contain diverse polynomial sets. Thus, the algorithms for these problems should not be deterministic but should have some controllable randomness. Several studies reported that the distribution of samples in a training set determines the generalization ability of models trained on it [19, 58]. However, the distribution of non-Gröbner sets and Gröbner bases is an unexplored and challenging object of study. It can be another challenging topic and goes beyond the scope of the present study.

## 4.1 Scope of this study

Non-Gröbner sets have various forms across applications. For example, in cryptography (particularly post-quantum cryptosystems), polynomials are restricted to dense degree-2 polynomials and generated by an encryption scheme [93]. On the other hand, in systems biology (particularly, reconstruction of gene regulatory networks), they are typically assumed to be sparse [59]. In statistics (particularly algebraic statistics), they are restricted to binomials, i.e., polynomials with two monomials [41, 79].

As the first study of Gröbner basis computation using Transformers, we do not focus on a particular application and instead address a generic case reflecting a motivation shared by various applications of computing Gröbner basis: solving polynomial systems or understanding ideals associated with polynomial systems having solutions. Particularly, we focus on 0-dimensional radical ideals, a special but fundamental class of ideals.

**Definition 4.3** (0-dimensional ideal). Let $F$ be a set of polynomials in $k[x_1, \ldots, x_n]$. An ideal $\langle F \rangle$ is called a *0-dimensional ideal* if all but a finite number of terms belong to $\mathrm{LT}(\langle F \rangle)$.

In fact, the number of terms not belong to $\mathrm{LT}(\langle f_1, \ldots, f_s \rangle)$ is an upper bound of the number of solutions of the system $f_1(x_1, \ldots, x_n) = \cdots = f_s(x_1, \ldots, x_n) = 0$. In particular, the finiteness of the number of terms not belong to $\mathrm{LT}(\langle f_1, \ldots, f_s \rangle)$ implies the finiteness of the number of solutions. This is the reason why we call such ideals "0-dimensional" ideals in Def. 4.3.

0-dimensional ideals are the fundamental ideals in the study of pure algebra. This is partly because of the ease of analysis. As Def. A.5 shows, 0-dimensional ideals relate to finite-dimensional vector spaces, and thus, analysis and algorithm design can be essentially addressed by matrices and linear algebra.

Also ideals in most practical scenarios are known to be 0-dimensional. For example, a multivariate public-key encrypted communication (a candidate of post-quantum cryptosystems) with a public polynomial system $F$ over a finite field $\mathbb{F}_p$ will be broken if one finds any root of the system $F \cup \{x_1^p - x_1, \ldots, x_n^p - x_n\})$. One should note that the ideal $\langle F \cup \{x_1^p - x_1, \ldots, x_n^p - x_n\}\rangle$ is 0-dimensional [84, Sec. 2.2]. Generically, 0-dimensional ideals defined from polynomial systems having solutions are radical[2] (i.e., non-radical ideals are in a zero-measure set in the Zariski topology). The proofs of the results in the following sections can be found in App. C. Hereinafter, the sampling of polynomials is done by a uniform sampling of coefficients from a prescribed range.

It is also worth noting that Transformers cannot be an efficient tool for *general* Gröbner basis computation, and thus, we should focus on a particular class of ideals and pursue in-distribution accuracy. This is evident from the facts that Gröbner basis computation is NP-hard and that machine learning models perform best on in-distribution samples and do not generalize perfectly. Fortunately, unlike standard machine learning tasks (e.g., image classification task), users can frame their problems beforehand (i.e., they know what types of polynomials they want to handle), and they can collect as many training samples as they want if an efficient algorithm exists. As mentioned above, the form of non-Gröbner sets varies across applications, and thus, we focus on the generic case and leave the specialization to future work.

## 4.2 Random generation of Gröbner bases

We address Prob. 4.1 using the fact that 0-dimensional radical ideals are generally *in shape position*.

**Definition 4.4** (Shape position). Ideal $I \subset k[x_1, \ldots, x_n]$ is called in *shape position* if some univariate polynomials $h, g_1, \ldots, g_{n-1} \in k[x_n]$ form the reduced $\prec_{\text{lex}}$-Gröbner basis of $I$ as follows.

$$G = \{h, x_1 - g_1, \ldots, x_{n-1} - g_{n-1}\}. \tag{4.1}$$

As can be seen, the $\prec_{\text{lex}}$-Gröbner basis consists of a univariate polynomial in $x_n$ and the difference of univariate polynomials in $x_n$ and a leading term $x_i$ for $i < n$. While not all ideals are in shape position, 0-dimensional radical ideals are almost always in shape position: if an $\langle f_1, \ldots, f_s \rangle \subset k[x_1, \ldots, x_n]$ is a 0-dimensional and radical ideal, a random coordinate change $(y_1, \ldots, y_n) = (x_1, \ldots, x_n)R$ with a regular (i.e., invertible) matrix $R \in k^n$ yields $\tilde{f}_1, \cdots, \tilde{f}_s \in k[y_1, \ldots, y_n]$, and the ideal $\langle y_1, \ldots, y_n \rangle$ generally has the reduced $\prec_{\text{lex}}$-Gröbner basis in the form of Eq. (4.1) (cf. Prop. A.14).

With this fact, an efficient sampling of Gröbner bases of 0-dimensional radical ideals can be realized by sampling $n$ polynomials in $k[x_n]$, i.e., $h, g_1, \ldots, g_{n-1}$ with $h \neq 0$. We have to make sure that the degree of $h$ is always greater than that of $g_1, \ldots, g_{n-1}$, which is necessary and sufficient for $G$ to be a reduced Gröbner basis. This approach involves efficiency and randomness, and thus resolving Prob. 4.1. Note that while our approach assumes term order $\prec_{\text{lex}}$, if necessary, one can use an efficient change-of-ordering algorithm, e.g., the FGLM algorithm [35]. The cost of the FGLM algorithm is $\mathcal{O}(n \cdot \deg(h)^3)$ based on the number of arithmetic operations over $k$. Besides the ideals in shape position, we also consider the Cauchy module in App. B, which defines another class of 0-dimensional ideals.

## 4.3 Backward Gröbner problem

To address Prob. 4.2, we consider the following problem.

**Problem 4.5.** *Let $I \subset k[x_1, \ldots, x_n]$ be a 0-dimensional ideal, and let $G = (g_1, \ldots, g_t)^\top \in k[x_1, \ldots, x_n]^t$ be its $\prec$-Gröbner basis with respect to term order $\prec$.[3] Find a polynomial matrix $A \in k[x_1, \ldots, x_n]^{s \times t}$ giving a non-Gröbner set $F = (f_1, \ldots, f_s)^\top = AG$ such that $\langle F \rangle = \langle G \rangle$.*

---

[2]See App. A for the definition.
[3]We surcharge notations to mean that the set $\{g_1, \ldots, g_t\}$ defined by the vector $G$ is a $\prec$-Gröbner basis.

Namely, we generate a set of polynomials $F = (f_1, \ldots, f_s)^\top$ from $G = (g_1, \ldots, g_t)^\top$ by $f_i = \sum_{j=1}^{t} a_{ij} g_j$ for $i = 1, \ldots, s$, where $a_{ij} \in k[x_1, \ldots, x_n]$ denotes the $(i, j)$-th entry of $A$. Note that $\langle F \rangle$ and $\langle G \rangle$ are generally not identical, and the design of $A$ such that $\langle F \rangle = \langle G \rangle$ is of our question.

A similar question was studied without the Gröbner condition in [17, 18]. They provided an algebraic necessary and sufficient condition for the polynomial system of $F$ to have a solution outside the variety defined by $G$. This condition is expressed explicitly by multivariate resultants. However, strong additional assumptions are required: $A, F, G$ are homogeneous, $G$ is a regular sequence, and in the end, $\langle F \rangle = \langle G \rangle$ is only satisfied up to saturation. Thus, they are not compatible with our setting and method for Prob. 4.1.

Our analysis gives the following results for the design $A$ to achieve $\langle F \rangle = \langle G \rangle$ for the 0-dimensional case (without radicality or shape position assumption).

**Theorem 4.6.** *Let $G = (g_1, \ldots, g_t)^\top$ be a Gröbner basis of a 0-dimensional ideal in $k[x_1, \ldots, x_n]$. Let $F = (f_1, \ldots, f_s)^\top = AG$ with $A \in k[x_1, \ldots, x_n]^{s \times t}$.*

1. *If $\langle F \rangle = \langle G \rangle$, it implies $s \geq n$.*

2. *If $A$ has a left-inverse in $k[x_1, \ldots, x_n]^{t \times s}$, $\langle F \rangle = \langle G \rangle$ holds.*

3. *The equality $\langle F \rangle = \langle G \rangle$ holds if and only if there exists a matrix $B \in k[x_1, \ldots, x_n]^{t \times s}$ such that each row of $BA - E_t$ is a syzygy[4] of $G$, where $E_t$ is the identity matrix of size $t$.*

The first statement of Thm. 4.6 argues that polynomial matrix $A$ should have at least $n$ rows. For an ideal in shape position, we have a $\prec_{\text{lex}}$-Gröbner basis $G$ of size $n$, and thus, $A$ is a square or tall matrix. The second statement shows a sufficient condition. The third statement provides a necessary and sufficient condition. Using the second statement, we design a simple random transform of a Gröbner basis to a non-Gröbner set without changing the ideal.

We now assume $\prec = \prec_{\text{lex}}$ and 0-dimensional ideals in shape position. Then, $G$ has exactly $n$ generators. When $s = n$, we have the following.

**Proposition 4.7.** *For any $A \in k[x_1, \ldots, x_n]^{n \times n}$ with $\det(A) \in k \setminus \{0\}$, we have $\langle F \rangle = \langle G \rangle$.*

As non-zero constant scaling does not change the ideal, we focus on $A$ with $\det(A) = \pm 1$ without loss of generality. Such $A$ can be constructed using the Bruhat decomposition:

$$A = U_1 P U_2, \tag{4.2}$$

where $U_1, U_2 \in \text{ST}(n, k[x_1, \ldots, x_n])$ are upper-triangular matrices with all-one diagonal entries (i.e., unimodular upper-triangular matrices) and $P \in \{0, 1\}^{n \times n}$ denotes a permutation matrix. Noting that $A^{-1}$ satisfies $A^{-1}A = E_n$, we have $\langle AG \rangle = \langle G \rangle$ from Thm. 4.6. Therefore, random sampling $(U_1, U_2, P)$ of unimodular upper-triangular matrices $U_1, U_2$ and a permutation matrix $P$ resolves the backward Gröbner problem for $s = n$.

We extend this idea to the case of $s > n$ using a rectangular unimodular upper-triangular matrix:

$$U_2 = \begin{pmatrix} U_2' \\ O_{s-n,n} \end{pmatrix} \in k[x_1, \ldots, x_n]^{s \times n}, \tag{4.3}$$

where $U_2' \in \text{ST}(n, k[x_1, \ldots, x_n])$ and $O_{s-n,n} \in k[x_1, \ldots, x_n]^{(s-n) \times n}$ is the zero matrix. The permutation matrix is now $P \in \{0, 1\}^{s \times s}$. Note that $U_2 G$ already gives a non-Gröbner set such that $\langle U_2 G \rangle = \langle G \rangle$; however, the polynomials in the last $s - n$ entries of $U_2 G$ are all zero by its construction. To avoid this, the permutation matrix $P$ shuffles the rows and also $U_1$ to exclude the zero polynomial from the final polynomial set.

To summarize, our strategy is to compute $F = U_1 P U_2 G$, which only requires a sampling of $\mathcal{O}(s^2)$ polynomials in $k[x_1, \ldots, x_n]$, and $\mathcal{O}(n^2 + s^2)$-times multiplications of polynomials. Note that even in the large polynomial systems in the MQ challenge, a post-quantum cryptography challenge, we have $n < 100$ and $s < 200$ [93].

---

[4]Refer to App. A for the definition.

## 4.4 Dataset generation algorithm

The combination of the discussion in the previous sections gives an efficient dataset generation algorithm (see Alg. 1 for a pseudocode).

**Theorem 4.8.** *Consider a polynomial ring $k[x_1, \ldots, x_n]$. Given the dataset size $m$, maximum degrees $d, d' > 0$, maximum size of non-Gröbner set $s_{\max} \geq n$, and term order $\prec$, Alg. 1 returns a collection $\mathcal{D} = \{(F_i, G_i)\}_{i=1}^m$ with the following properties: For all $i = 1, \ldots, m$,*

1. *$|G_i| = n$ and $|F_i| \leq s_{\max}$.*

2. *The set $G_i$ is the reduced $\prec$-Gröbner basis of $\langle F_i \rangle$. The set $F_i$ is not, unless $G_i, U_1, U_2', P$ are all sampled in a non-trivial Zariski closed subset.[5]*

3. *The ideal $\langle F_i \rangle = \langle G_i \rangle$ is a 0-dimensional ideal in shape position.*

*The time complexity is $\mathcal{O}(m(nS_{1,d} + s^2 S_{n,d'} + (n^2 + s^2)M_{n,2d'+d}))$ when $\prec = \prec_{\mathrm{lex}}$, where $S_{n,d}$ denotes the complexity of sampling an $n$-variate polynomial with total degree at most $d$, and $M_{n,d}$ denotes that of multiplying two $n$-variate polynomials with total degree at most $d$. If $\prec \neq \prec_{\mathrm{lex}}$, $\mathcal{O}(mnd^3)$ is additionally needed.*

The proposed dataset generation method is a backward approach, which first generates solutions and then transforms them into problems. In this case, we have control over the complexity of the Gröbner bases and can add some intrinsic structure if any prior information is available. For example, a multi-variate encryption scheme encapsulates secret key information in the solution of a polynomial system with a single solution in a base field [93]. The ideal associated with such a system is 0-dimensional and in shape position; namely, its Gröbner basis has the following form: $G = \langle x_n - a_n, x_1 - a_1, \ldots, x_{n-1} - a_{n-1} \rangle$, where $a_1, a_2, \ldots, a_n$ are constants [84]. The backward approach allows one to restrict the Gröbner bases in a dataset into such a class.

This is not the case with forward approaches. While they may include prior information into non-Gröbner sets, it is computationally expensive to obtain the corresponding Gröbner bases. It is also worth noting that a naive forward approach, which randomly generates non-Gröbner sets and computes their Gröbner bases, should be avoided even if it were computationally tractable because, for example, if the Gröbner basis of such $F$ is generally $\{1\}$ when $|F| > n$.

## 5 Hybrid Input Embedding

Transformers are an efficient learner of sequence-to-sequence functions. They receive and generate a sequence of *tokens*. In our context, for example, $\{x^2 - 10, y\}$ can be tokenized as [x, ^, 2, +, -10, <sep>, y], a sequence of tokens. The vocabulary set $\mathcal{V}$ is the collection of all possible tokens. Each token $s \in \mathcal{V}$ has a predesignated token ID $i(s) \in \mathbb{N}$, and the input embedding layer of Transformer associates them with the $i(s)$-th row of the embedding matrix $W_{\mathrm{E}} \in \mathbb{R}^{|\mathcal{V}| \times D}$, where $D$ is the embedding dimension. The embedding matrix is updated during training, and eventually, we have a nice vector representation for each token.

However, this approach requires a large embedding matrix to handle a wide range of number tokens.[6] The number tokens dominate the vocabulary set, and Transformers have to learn the relationship of the number tokens from scratch during the training. At the inference, if the given numbers are out of range, the model cannot work. In our case, this is inconvenient, particularly when the coefficient field is $k = \mathbb{Q}$. For example, let $F = \{-5/3y^3 - y - 1/2, 7/2xy^2 - 5/3x - 2\}$, which we obtained from a random sampling with some upper bounds on the degree, number of terms, and integers appearing in numerators and denominators. Even the simplicity of $F$, its reduced $\prec_{\mathrm{lex}}$-Gröbner basis has a polynomial of $g_1 = x + 569520/427411y^2 - 158760/427411y + 612912/427411$. If we tokenize a/b as [a, /, b], we need to prepare more than a million integer tokens. Noting that $g_1 \approx x + 1.33y^2 - 0.37y + 1.43$, it is more reasonable to handle coefficients as real values and also implement the inductive bias on the continuity of numbers in the embedding.

---

[5]This can happen with probability zero if $k$ is infinite and very low probability over a large finite field.

[6]We may tokenize a number by the digits (e.g., 123 by [1, 2, 3]), but this makes input sequences long and affects the quadratic memory cost of attention mechanism. See [19] for the effect of number tokenization.

Table 1: Runtime comparison (in seconds) of forward generation (F.) and backward generation (B.) of dataset $\mathcal{D}_n(\mathbb{Q})$ of size 1,000. The forward generation used either of the three algorithms provided in SageMath with the libSingular backend. We set a timeout limit to five seconds (added to the total runtime at every occurrence) for each Gröbner basis computation. The numbers with † and ‡ include the timeout for more than 13% and 25% of the runs, respectively (cf. Tab. 5 for the success rate).

| Method | $n = 2$ | $n = 3$ | $n = 4$ | $n = 5$ |
|---|---|---|---|---|
| F. (STD) | 4.20 | 216.3 | 740.1† | 1411.1‡ |
| F. (SLIMGB) | 4.29 | 183.4 | 697.5† | 1322.7‡ |
| F. (STDFGLM) | 7.22 | 8.29 | 21.0 | 164.3 |
| B. (ours) | 5.23 | 5.46 | 7.05 | 7.91 |

We propose a hybrid input embedding that accepts both discrete token IDs and continuous values. Let $\boldsymbol{s} = [s_1, \ldots, s_L]$ to be a sequence of tokens. Some of these tokens are in $\mathcal{V}$ and otherwise in $\mathbb{R}$. For those in $\mathcal{V}$, the standard input embedding based on the embedding matrix is applied. For the others, a small feed-forward network $f_E : \mathbb{R} \to \mathbb{R}^D$ is applied. A Transformer with the proposed embedding should equip a regression head for these continuous tokens. This allows us to handle any number as a single token without the explosion of the vocabulary set (i.e., embedding matrix). As feed-forward networks are a continuous function, they naturally implement the continuity of numbers; two close values $s_1, s_2 \in \mathbb{R}$ are expected to be embedded in similar vectors. The hybrid input embedding has two advantages. First, as claimed above, we are no longer suffering from the large embedding matrix for registering many number tokens and can naturally implement the continuity bias. Second, We do not have the "out-of-range" issue. Further, we can scale the coefficients of given polynomials globally so that they match our training coefficient range.[7] Refer to App. D for the details.

## 6 Experiments

We now present the efficiency of the proposed dataset generation method and the learnability of Gröbner basis computation.[8] All the experiments were conducted with 48-core CPUs, 768GB RAM, and NVIDIA RTX A6000ada GPUs. The training of a model takes less than a day on a single GPU. More information on the profile of generated datasets, the training setup, and additional experimental results are given in Apps. E and F.

### 6.1 Dataset generation

First, we demonstrate the efficiency of the proposed dataset generation framework. We constructed 16 datasets $\mathcal{D}_n(k)$ for $n \in \{2, 3, 4, 5\}$ and $k \in \{\mathbb{F}_7, \mathbb{F}_{31}, \mathbb{Q}, \mathbb{R}\}$ and measured the runtime of the forward generation and our backward generation. The dataset $\mathcal{D}_n(k)$ consists of 1,000 pairs of non-Gröbner set and Gröbner basis in $k[x_1, \ldots, x_n]$ of ideals in shape position. Each sample $(F, G) \in \mathcal{D}_n(k)$ was prepared using Alg. 1 with $(d, d', s_{\max}, \prec) = (5, 3, n + 2, \prec_{\text{lex}})$. The number of terms of univariate polynomials and $n$-variate polynomials is uniformly determined from $[1, 5]$ and $[1, 2]$, respectively. When $k = \mathbb{Q}$, the coefficient $a/b$ are restricted to those with $a, b \in \{-5, \ldots, 5\}$ for random polynomials and $a, b \in \{-100, \ldots, 100\}$ for polynomials in $F$. In the forward generation, one may first generate random polynomial sets and then compute their Gröbner bases. However, this leads to a dataset with a totally different complexity from that constructed by the backward generation, leading to an unfair runtime comparison between the two generation processes. As such, the forward generation instead computes Gröbner bases of the non-Gröbner sets given by the backward generation, leading to the identical dataset. We used SageMath [82] with the libSingular backend. As Tab. 1 shows, our backward generation is a few orders of magnitude faster than the forward generation. A sharp runtime growth is observed in the forward generation as the number of variables increases. Note that these numbers only show the runtime on 1,000 samples, while training typically requires millions of samples. Therefore, the forward generation is almost infeasible, and the proposed method resolves a bottleneck in the learning of Gröbner basis computation.

---

[7]In the follow-up survey, we found that a very similar idea was proposed in [37] in a broader context, which shows continuous embedding of number tokens perform better than the discrete embedding in various tasks.

[8]The code is available at `https://github.com/HiroshiKERA/transformer-groebner`.

Table 2: Accuracy [%] / support accuracy [%] of Gröbner basis computation by Transformer on $\mathcal{D}_n^-(k)$. In the support accuracy, two polynomials are considered identical if they consist of an identical set of terms (i.e., identical *support*), Transformers are trained on either discrete input embedding (*disc.*) and the hybrid embedding (*hyb.*). Note that the datasets for $n = 3, 4, 5$ are here constructed using $U_1, U_2'$ (cf. Alg. 1) with density $\sigma = 0.6, 0.3, 0.2$, respectively.

| Coeff. | | Shape position | | | | Cauchy module | |
|---|---|---|---|---|---|---|---|
| | | $n = 2$ | $n = 3$ | $n = 4$ | $n = 5$ | $n = 2$ | $n = 3$ |
| $\mathbb{Q}$ | *disc.* | 93.7 / 95.4 | 88.7 / 92.0 | 90.8 / 94.0 | 86.5 / 90.6 | 99.7 / 99.8 | 97.2 / 97.6 |
| | *hyb.* | 66.8 / 87.3 | 69.0 / 89.8 | 62.7 / 86.8 | 0.0 / 84.9 | 98.3 / 99.7 | 80.1 / 89.2 |
| $\mathbb{F}_7$ | *disc.* | 72.3 / 79.1 | 78.1 / 83.2 | 71.3 / 84.6 | 84.3 / 88.5 | 98.7 / 99.8 | 98.1 / 98.7 |
| | *hyb.* | 54.1 / 78.7 | 55.8 / 84.3 | 46.1 / 81.8 | 54.4 / 81.5 | 95.8 / 99.7 | 80.8 / 91.2 |
| $\mathbb{F}_{31}$ | *disc.* | 46.8 / 77.3 | 50.2 / 80.9 | 51.1 / 83.7 | 28.6 / 77.9 | 93.8 / 99.7 | 94.7 / 99.6 |
| | *hyb.* | 6.1 / 75.3 | 5.8 / 80.4 | 0.1 / 73.0 | 0.1 / 76.9 | 15.1 / 99.5 | 10.9 / 98.4 |
| $\mathbb{R}$ | *hyb.* | 57.2 / 85.0 | 61.0 / 88.0 | 61.7 / 87.5 | 45.6 / 82.9 | 28.3 / 100 | 4.3 / 100 |

## 6.2 Learnability of Gröbner basis computation

We now demonstrate that Transformers can learn to compute Gröbner bases. To examine the general Transformer's ability, we focus on a standard architecture (e.g., 6 encoder/decoder layers and 8 attention heads) and a standard training setup (e.g., the AdamW optimizer [65] with $(\beta_1, \beta_2) = (0.9, 0.999)$ and a linear decay of learning rate from $10^{-4}$). The batch size was set to 16, and models were trained for 8 epochs. We also tested the hybrid input embedding. Refer to App. E for the complete information. Each polynomial set in the datasets is converted into a sequence using the prefix representation and the separator tokens. Unlike natural language processing, our task does not allow the truncation of an input sequence because the first term of the first polynomial in $F$ certainly relates to the last term of the last polynomial. To make the input sequence length manageable for vanilla Transformers, we used simpler datasets $\mathcal{D}_n^-(k)$ using $U_1, U_2'$ in Alg. 1 of a moderate density $\sigma \in (0, 1]$. This makes the maximum sequence length less than 5,000. Specifically, we used $\sigma = 1.0, 0.6, 0.3, 0.2$ for $n = 2, 3, 4, 5$, respectively. The training set has one million samples, and the test set has one thousand samples. With hybrid input embedding, coefficients are predicted by regression, and we quantized them for $\mathbb{F}_p$ and otherwise regarded them correct when the mean squared error is less than 0.1.

Table 2 shows that trained Transformers successfully compute Gröbner bases with moderate/high accuracy. Several intriguing observations below are obtained. See App. F for more results. Particularly, App. F.3 presents several examples found in the datasets for which Transformer successfully computed Gröbner bases significantly faster than math algorithms. Table 2 also includes the results on Cauchy module datasets on which Transformers are trained and tested. The dataset generation starts with sampling the roots in $k^n$, and the other parts follow the generation of $\mathcal{D}_n^-(k)$. The results on $(\mathbb{Q}, n = 3)$ with standard embedding is not shown as it requires too many number tokens.

**The performance gap across the rings.** The accuracy shows that the learning is more successful on infinite field coefficients $k \in \{\mathbb{Q}, \mathbb{R}\}$ than finite field ones $k = \mathbb{F}_p$. This may be a counter-intuitive observation because there are more possible coefficients in $G$ and $F$ for $\mathbb{Q}$ than $\mathbb{F}_p$. Specifically, for $G$, the coefficient $a/b \in \mathbb{Q}$ is restricted to those with $a, b \in \{-5, \ldots, 5\}$ (i.e., roughly 50 choices), and $a, b \in \{-100, \ldots, 100\}$ (i.e., roughly 20,000 choices) for $F$. In contrast, there are only $p$ choices for $\mathbb{F}_p$. The performance even degrades for the larger order $p = 31$. Interestingly, the support accuracy shows that the terms forming the polynomial (i.e., the *support* of polynomial) are correctly identified well. Thus, Transformers have difficulty determining the coefficients in finite fields. Several studies have also reported that learning to solve a problem involving modular arithmetic may encounter some difficulties [21, 38, 74], but no critical workaround is known.

**Incorrect yet reasonable failures.** We observed that the predictions by a Transformer are mostly reasonable even when they are incorrect. For example, only several coefficients may be incorrect, and the support can be correct as suggested by the relatively high support accuracy in Tab. 2. In such a case, one can use a Gröbner basis computation algorithm that works efficiently given the leading terms of the target unknown Gröbner basis [83]. Refer to App. F.2 for extensive lists of examples.

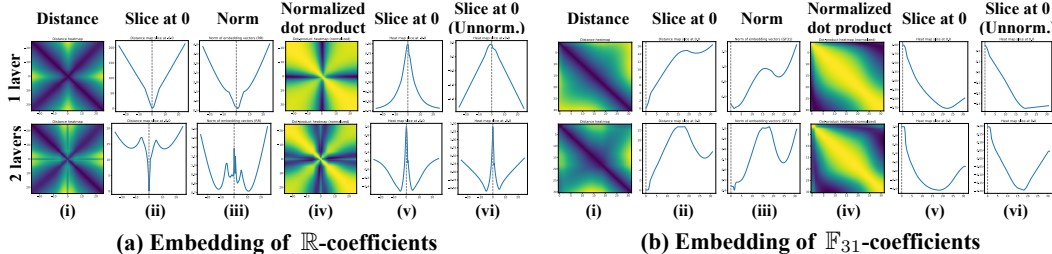

| Distance | Slice at 0 | Norm | Normalized dot product | Slice at 0 | Slice at 0 (Unnorm.) | Distance | Slice at 0 | Norm | Normalized dot product | Slice at 0 | Slice at 0 (Unnorm.) |

(a) Embedding of $\mathbb{R}$-coefficients     (b) Embedding of $\mathbb{F}_{31}$-coefficients

Figure 1: Visual analysis of embedding vectors of numbers given by the proposed embedding. Embedding $c \in \mathbb{R}$ to $f_{\mathrm{E}}(c) \in \mathbb{R}^D$ from $c_{\min}$ to $c_{\max}$ with $B$ bins to obtain $M \in \mathbb{R}^{B \times D}$, the fix figures show from the left, (i) the Euclidean distance matrix of $M$, (ii) its slice at 0, (iii) the norm of embedding vectors, (iv) the dot product $\tilde{M}\tilde{M}^\top$ with $\tilde{M}$ of the row-normalized $M$, (v) $f_{\mathrm{E}}(0)^\top \tilde{M}$ and (vi) $f_{\mathrm{E}}(c_0)^\top M$. (a) Trained on $\mathbb{R}[x_1, x_2]$; $(c_{\min}, c_{\max}) = (-100, 100)$. (b) Trained on $\mathbb{F}_{31}[x_1, x_2]$; $(c_{\min}, c_{\max}) = (0, 31)$. The embedding layer $f_{\mathrm{E}}$ has one/two hidden layers (top/bottom rows). As can be seen, the relationship between embedding vectors in terms of distance and dot product is aligned well in the infinite field and not in the finite field.

**Hybrid embedding.** Table 2 shows that determining coefficients by regression is less successful than classifications. For infinite field $k$, this may be because of the accumulation of coefficient errors during the auto-regressive generation. Thus, the current best practice would be to prepare many number tokens in the vocabulary set, or a sophisticated regression-by-classification approach may be helpful [76]. Note that the results for $k = \mathbb{F}_p$ are shown for reference as the finite field elements do not have ordering. Figure 1 shows a contrast between the embedding functions learned in infinite field and finite field. Particularly, the slice of distance matrix (ii) and that of the dot-product matrix (vi) show that these metrics align well with the difference between numbers in $\mathbb{R}$. However, we cannot observe convincing patterns in the embedding in $\mathbb{F}_{31}$. For the two-layer case in Fig. 1(a), we observe sharp changes around $\pm 5$ of the horizontal axis. This may be because of the gap in the coefficient range in the input and output space. The coefficients of $F$ ranges between $[-100, 100]$, while that of $G$ does between $[-5, 5]$. In Tab. 3, we show that the increase of hidden layers of $f_{\mathrm{E}}$ does not lead to improvement.

## 7 Conclusion

This study proposed the first learning approach to a fundamental algebraic task, the Gröbner basis computation. While various recent studies have reported the learnability of mathematical problems by Transformers, we addressed the first problem with nontriviality in the dataset generation. Ultimately, the learning approach may be useful to address large-scale problems that cannot be approached by Gröbner basis computation algorithms because of their computational complexity. Transformers can output predictions in moderate runtime. The outputs may be incorrect, but there is a chance of obtaining a hint of a solution, as shown in our experiments. We believe that our study reveals many interesting open questions to achieve Gröbner basis computation learning. Some are algebraic problems, and others are machine learning challenges, further discussed in Sec. H.

**Acknowledgement.** We would like to thank Masayuki Noro (Rikkyo University) for his fruitful comments on our dataset construction algorithm and Noriki Nishida (RIKEN Center for Advanced Intelligence Project) for his help in the implementation. Hiroshi Kera was supported by JST PRESTO Grant Number JPMJPR24K4, JST ACT-X Grant Number JPMJAX23C8, Mitsubishi Electric Information Technology R&D Center, and the Chiba University IAAR Research Support Program and the Program for Forming Japan's Peak Research Universities (J-PEAKS). Yuki Ishihara was supported by JSPS KAKENHI Grant Number JP22K13901 and Institute of Mathematics for Industry, Joint Usage/Research Center in Kyushu University (FY2023 Short-term Joint Research "Speeding up of symbolic computation and its application to solving industrial problems" (2023a006)). Yuta Kambe was supported by Mitsubishi Electric Research Associate Program.

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

# A    Basic Definitions in Algebra

**Definition A.1** (Ring, Field ([7], Chap. 1 §1))**.** A set $R$ with an additive operation $+$ and a multiplicative operation $*$ is called a (commutative) ring if it satisfies the following conditions:

1. $a + (b + c) = (a + b) + c$ for any $a, b, c \in R$,

2. there exists $0 \in R$ such that $a + 0 = 0 + a = a$ for any $a \in R$,

3. for any $a \in R$, there exists $-a$ such that $a + (-a) = (-a) + a = 0$,

4. $a + b = b + a$ for any $a, b \in R$,

5. $a * (b * c) = (a * b) * c$ for any $a, b, c \in R$,

6. there exists $1 \in R$ such that $a * 1 = 1 * a = a$ for any $a \in R$,

7. $a * (b + c) = a * b + a * c$ for any $a, b, c \in R$,

8. $(a + b) * c = a * c + b * c$ for any $a, b, c \in R$,

9. $a * b = b * a$ for any $a, b \in R$.

A commutative ring $R$ is called a field if it satisfies the following condition

10. for any $a \in R \setminus \{0\}$, there exists $a^{-1}$ such that $a * a^{-1} = a^{-1} * a = 1$.

**Definition A.2** (Polynomial Ring ([7], Chap. 1 §1))**.** In Definition A.1, $k[x_1, \ldots, x_n]$, the set of all $n$-variate polynomials with coefficients in $k$, satisfies all conditions (1)-(9). Thus, $k[x_1, \ldots, x_n]$ is called a polynomial ring.

**Definition A.3** (Quotient Ring ([7], Chap. 1 §1))**.** Let $R$ be a ring and $I$ an ideal of $R$. For each $f \in R$, we set $[f] = \{g \in R \mid f - g \in I\}$. Then, the set $\{[f] \mid f \in R\}$ is called the quotient ring of $R$ modulo $I$ and denoted by $R/I$. Indeed, $R/I$ is a ring with an additive operation $+$ and a multiplicative operation $*$, where $[f] + [g] = [f + g]$ and $[f] * [g] = [f * g]$ for $f, g \in R$ respectively.

**Definition A.4** (Generators)**.** For $F = \{f_1, \ldots, f_s\} \subset k[x_1, \ldots, x_n]$, the following set

$$\langle F \rangle = \left\{ \sum_{i=1}^{s} h_i f_i \mid h_1, \ldots, h_s \in k[x_1, \ldots, x_n] \right\}. \tag{A.1}$$

is an ideal and said to be *generated* by $F$, and $f_1, \ldots, f_s$ are called *generators*.

**Definition A.5** (0-dimensional ideal ([25], Chap. 5 §3, Thm. 6))**.** Let $F$ be a set of polynomials in $k[x_1, \ldots, x_n]$. An ideal $\langle F \rangle$ is called a *0-dimensional ideal* if the $k$-linear space $k[x_1, \ldots, x_n]/\langle F \rangle$ is finite-dimensional, where $k[x_1, \ldots, x_n]/\langle F \rangle$ is the quotient ring of $k[x_1, \ldots, x_n]$ modulo $\langle F \rangle$.

**Definition A.6** (Radical ideal ([7], Chap. 1 §1))**.** For an ideal $I$ of $k[x_1, \ldots, x_n]$, the set $\{f \in k[x_1, \ldots, x_n] \mid f^m \in I$ for a positive integer $m\}$ is called the radical of $I$ and denoted by $\sqrt{I}$. Also, $I$ is called a radical ideal if $I = \sqrt{I}$.

**Definition A.7** (Syzygy ([10], Chap. 3, §3))**.** Let $F = \{f_1, \ldots, f_s\} \subset k[x_1, \ldots, x_n]$. A *syzygy* of $F$ is an $s$-tuple of polynomials $(q_1, \ldots, q_s) \in k[x_1, \ldots, x_n]^s$ such that $q_1 f_1 + \cdots + q_s f_s = 0$.

**Definition A.8** (Term ([10], Chap. 2, §1))**.** For a polynomial $f = \sum_{\alpha_1, \ldots, \alpha_n} c_{\alpha_1, \ldots, \alpha_n} x_1^{\alpha_1} \cdots x_n^{\alpha_n}$ with $c_{\alpha_1, \ldots, \alpha_n} \in K$ and $\alpha_1, \ldots, \alpha_n \in \mathbb{Z}_{\geq 0}$, each $x_1^{\alpha_1} \cdots x_n^{\alpha_n}$ is called a term in $f$.

**Definition A.9** (Total Degree ([25], Chap. 1 §1, Def. 3))**.** For a term $x_1^{\alpha_1} \cdots x_n^{\alpha_n}$, its total degree is the sum of indices $\alpha_1 + \cdots + \alpha_n$. For a polynomial $f$, the total degree of $f$ is the maximal total degree of terms in $f$.

**Definition A.10** (Term order ([10], Definition 5.3))**.** A *term order* $\prec$ is a relation between terms such that

1. (comparability) for different terms $x_1^{\alpha_1} \cdots x_n^{\alpha_n}$ and $x_1^{\beta_1} \cdots x_n^{\beta_n}$, either $x_1^{\alpha_1} \cdots x_n^{\alpha_n} \prec x_1^{\beta_1} \cdots x_n^{\beta_n}$ or $x_1^{\beta_1} \cdots x_n^{\beta_n} \prec x_1^{\alpha_1} \cdots x_n^{\alpha_n}$ holds,

2. (order-preserving) for terms $x_1^{\alpha_1} \cdots x_n^{\alpha_n}$, $x_1^{\beta_1} \cdots x_n^{\beta_n}$ and $x_1^{\gamma_1} \cdots x_n^{\gamma_n} \neq 1$, if $x_1^{\alpha_1} \cdots x_n^{\alpha_n} \prec x_1^{\beta_1} \cdots x_n^{\beta_n}$ then $x_1^{\alpha_1 + \gamma_1} \cdots x_n^{\alpha_n + \gamma_n} \prec x_1^{\beta_1 + + \gamma_1} \cdots x_n^{\beta_n + \gamma_n}$ holds,

3. (minimality of 1) the term 1 is the smallest term i.e. $1 \prec x_1^{\alpha_1} \cdots x_n^{\alpha_n}$ for any term $x_1^{\alpha_1} \cdots x_n^{\alpha_n} \neq 1$.

**Example A.11.** The *lexicographic order* $\prec_{\mathrm{lex}}$ prioritizes terms with larger exponents for the variables of small indices, e.g.,

$$x_2 \succ_{\mathrm{lex}} x_3^2 \quad \text{and} \quad x_1 x_2 x_3^2 \prec_{\mathrm{lex}} x_1 x_2^2 x_3. \tag{A.2}$$

Two terms are first compared in terms of the exponent in $x_1$ (larger one is prioritized), and if a tie-break is needed, the next variable $x_2$ is considered, and so forth.

**Example A.12.** The *graded lexicographic order* $\prec_{\mathrm{grlex}}$ prioritizes terms with higher total degree.[9] For tie-break, the lexicographic order is used, e.g.,

$$1 \prec_{\mathrm{grlex}} x_n \quad \text{and} \quad x_2 \prec_{\mathrm{grlex}} x_3^2 \quad \text{and} \quad x_1 x_2 x_3^2 \prec_{\mathrm{grlex}} x_1 x_2^2 x_3. \tag{A.3}$$

Term orders prioritizing lower total degree terms as $\prec_{\mathrm{grlex}}$ are called graded term orders.

**Definition A.13** (Reduced Gröbner basis). A $\prec$-Gröbner basis $G = \{g_1, \ldots, g_t\}$ of $I$ is called the reduced Gröbner basis of $I$ if

    1. the leading coefficient of $g_i$ with respect to $\prec$ is 1 for all $i = 1, \ldots, t$,

    2. no terms of $g_i$ lies on $\langle \mathrm{LT}(G \setminus \{g_i\}) \rangle$ for any $i = 1, \ldots, t$.

**Proposition A.14** ([36], Prop. 1.6; [69], Lem. 4.4). *Let $I$ be a 0-dimensional radical ideal. If $k$ is of characteristic 0 or a finite field of large enough order, then a random linear coordinate change puts $I$ in shape position.*

# B Cauchy module

We here provide the definition of the Cauchy module, which defines another class of 0-dimensional ideals.

**Definition B.1** (Elementary symmetric polynomials). The elementary symmetric polynomials $s_1, \ldots, s_n$ in $n$ variables $x_1, \ldots, x_n$ are

$$s_k = \sum_{i_1 < \cdots < i_k} x_{i_1} \cdots x_{i_k}, \quad k = 1, \ldots, n. \tag{B.1}$$

**Definition B.2** (Cauchy module). Let $S_n$ be the symmetric group on a finite set of size $n$. Let $k$ be an algebraically closed field, and let $\alpha = (\alpha_1, \cdots, \alpha_n) \in k^n$ be a generic point (i.e., $\alpha_i \neq \alpha_j$ for $i \neq j$). Let the finite subset $A \subset k^n$

$$A = \{\sigma(\alpha) = (\alpha_{\sigma(1)}, \ldots, \alpha_{\sigma(n)}) \mid \sigma \in S_n\}. \tag{B.2}$$

Let $f_1(t)$ be a polynomial of $t$,

$$f_1(t) = t^2 - s_1(\alpha) t^{n-1} + s_2(\alpha) t^{n-2} - \cdots + (-1)^n s_n(\alpha) = \prod_{i=1}^{n} (t - \alpha_i). \tag{B.3}$$

Let us introduce indeterminates $z_1, \ldots, z_n$ and

$$f_2(z_2, z_1) = \frac{f_1(z_2) - f_1(z_1)}{z_2 - z_1} \tag{B.4}$$

$$f_3(z_3, z_2, z_1) = \frac{f_2(z_3, z_1) - f_2(z_2, z_1)}{z_3 - z_2} \tag{B.5}$$

$$\vdots \tag{B.6}$$

$$f_n(z_n, \ldots, z_1) = \frac{f_{n-1}(z_n, z_{n-2}, \ldots, z_1) - f_{n-1}(z_{n-1}, z_{n-2}, \ldots, z_1)}{z_n - z_{n-1}} \tag{B.7}$$

$$(= z_1 + \cdots + z_n - s_1(\alpha)) \tag{B.8}$$

The set of polynomials $C = \{f_1, \ldots, f_s\}$ is called the Cauchy module.

**Remark B.3.** The Cauchy module is the reduced $\prec_{\mathrm{lex}}$-Gröbner basis of $\langle C \rangle$ with $z_1 \prec_{\mathrm{lex}} \cdots \prec_{\mathrm{lex}} z_n$.

---

[9]The total degree of term $x_1^{\alpha_1} \cdots x_n^{\alpha_n}$ refers to $\sum_{i=1}^{n} \alpha_i$. The total degree of polynomial $f$ refers to the maximum total degree of the terms in $f$.

---

**Algorithm 1:** Dataset generation for learning to compute 0-dimensional Gröbner bases.

---
**Assumption:** polynomial ring $k[x_1, \ldots, x_n]$
**Input:** dataset size $m$, maximum degrees $d, d'$, maximum size of non-Gröbner set $s_{\max} \geq n$,
      and term order $\prec$.
**Output:** collection $\mathcal{D} = \{(F_i, G_i)\}_{i=1}^m$ of non-Gröbner set $F_i \in k[x_1, \ldots, x_n]^m$ and the
      reduced $\prec$-Gröbner basis $G_i \subset k[x_1, \ldots, x_n]$ of 0-dimensional ideal $\langle F \rangle = \langle G \rangle$.

---

**1** $\mathcal{D} \leftarrow \{\ \}$
**2 for** $i = 1, \ldots, m$ **do**
**3**    $G_i \leftarrow \{h\}$ with $h$, non-constant, monic/primitive, sampled from $k[x_n]_{\leq d}$.   ▷ Prob. 4.1
**4**    **for** $j = 1, \ldots, n-1$ **do**
**5**       |  $G_i \leftarrow G_i \cup \{g_j\}$ with $g_j$ sampled from $k[x_n]_{\leq \deg(h)-1}$.
**6**    **end**
**7**    $s \sim \mathbb{U}[n, s_{\max}]$   ▷ Prob. 4.2
**8**    Sample a unimodular upper-triangular matrix $U_1 \in \mathrm{ST}(s, k[x_1, \ldots, x_n]_{\leq d'})$.
**9**    Sample a unimodular upper-triangular matrix $U_2' \in \mathrm{ST}(n, k[x_1, \ldots, x_n]_{\leq d'})$.
**10**   Sample a permutation matrix $P \in \{0,1\}^{s \times s}$
**11**   $F_i \leftarrow U_1 P U_2 G_i$, where $U_2 = [{U_2'}^\top \ \ O_{n,s-n}]^\top \in k[x_1, \ldots, x_n]^{s \times n}$.
**12**   **if** $\prec \neq \prec_{\mathrm{lex}}$ **then**
**13**      |  $G_i \leftarrow \mathrm{FGLM}(G_i, \prec_{\mathrm{lex}}, \prec)$
**14**   **end**
**15**   $\mathcal{D} \leftarrow \mathcal{D} \cup \{(F_i, G_i)\}$   ▷ Reorder terms in $F_i$ if $\prec \neq \prec_{\mathrm{lex}}$.
**16 end**

---

## C   Dataset generation algorithm

**Theorem 4.6.** *Let $G = (g_1, \ldots, g_t)^\top$ be a Gröbner basis of a 0-dimensional ideal in $k[x_1, \ldots, x_n]$. Let $F = (f_1, \ldots, f_s)^\top = AG$ with $A \in k[x_1, \ldots, x_n]^{s \times t}$.*

    *1. If $\langle F \rangle = \langle G \rangle$, it implies $s \geq n$.*

    *2. If $A$ has a left-inverse in $k[x_1, \ldots, x_n]^{t \times s}$, $\langle F \rangle = \langle G \rangle$ holds.*

    *3. The equality $\langle F \rangle = \langle G \rangle$ holds if and only if there exists a matrix $B \in k[x_1, \ldots, x_n]^{t \times s}$ such that each row of $BA - E_t$ is a syzygy[10] of $G$, where $E_t$ is the identity matrix of size $t$.*

*Proof.*
(1) In general, if an ideal $I$ is generated by $s$ elements and $s < n$, then the Krull dimension of $k[x_1, \ldots, x_n]/I$ satisfies that $\dim k[x_1, \ldots, x_n]/I \geq n - s > 0$ (Krull's principal ideal theorem [31, §10]). Since the Krull dimension of $k[x_1, \ldots, x_n]/\langle G \rangle$ is 0, we have $s \geq n$.

(2) From $F = AG$, we have $\langle F \rangle \subset \langle G \rangle$. If $A$ has a left-inverse $B \in k[x_1, \ldots, x_n]^{t \times s}$, we have $BF = BAG = G$, indicating $\langle F \rangle \supset \langle G \rangle$. Therefore, we have $\langle F \rangle = \langle G \rangle$.

(3) If the equality $\langle F \rangle = \langle G \rangle$ holds, then there exists a $t \times s$ matrix $B \in k[x_1, \ldots, x_n]^{t \times s}$ such that $G = BF$. Since $F$ is defined as $F = AG$, we have $G = BF = BAG$ and $G = E_t G$ in $k[x_1, \ldots, x_n]^t$. Therefore we obtain $(BA - E_t)G = 0$. In particular, each row of $BA - E_t$ is a syzygy of $G$. Conversely, if there exists a $t \times s$ matrix $B \in k[x_1, \ldots, x_n]^{t \times s}$ such that each row of $BA - E_t$ is a syzygy of $G$, then we have $(BA - E_t)G = 0$ in $k[x_1, \ldots, x_n]^t$, therefore the equality $\langle F \rangle = \langle G \rangle$ holds since we have $G = E_t G = BAG = BF$.    □

**Proposition 4.7.** *For any $A \in k[x_1, \ldots, x_n]^{n \times n}$ with $\det(A) \in k \setminus \{0\}$, we have $\langle F \rangle = \langle G \rangle$.*

---

[10]Refer to App. A for the definition.

*Proof.* From the Cramer's rule, there exists $B \in k[x_1, \ldots, x_n]^{n \times n}$ such that $BA = \det(A)E_n$, where $E_n$ denotes the $n$-by-$n$ identity matrix. Indeed, the $i$-th row $B_i$ of $B$ satisfies for $i = 1, \ldots, n$,

$$B_i = \frac{1}{\det(A)} \left( \det\left( \tilde{A}_1^{(i)} \right), \ldots, \det\left( \tilde{A}_n^{(i)} \right) \right), \tag{C.1}$$

where $\tilde{A}_j^{(i)}$ is the matrix $A$ with the $j$-th column replaced by the $i$-th canonical basis $(0, ..., 1, ..., 0)^\top$. Since $\det(A)$ is a non-zero constant, $A$ has the left-inverse $B$ in $k[x_1, \ldots, x_n]^{n \times n}$. Thus $\langle F \rangle = \langle G \rangle$ from Thm. 4.6. $\square$

**Theorem 4.8.** *Consider a polynomial ring $k[x_1, \ldots, x_n]$. Given the dataset size $m$, maximum degrees $d, d' > 0$, maximum size of non-Gröbner set $s_{\max} \geq n$, and term order $\prec$, Alg. 1 returns a collection $\mathcal{D} = \{(F_i, G_i)\}_{i=1}^m$ with the following properties: For all $i = 1, \ldots, m$,*

1. *$|G_i| = n$ and $|F_i| \leq s_{\max}$.*

2. *The set $G_i$ is the reduced $\prec$-Gröbner basis of $\langle F_i \rangle$. The set $F_i$ is not, unless $G_i, U_1, U_2', P$ are all sampled in a non-trivial Zariski closed subset.[11]*

3. *The ideal $\langle F_i \rangle = \langle G_i \rangle$ is a 0-dimensional ideal in shape position.*

*The time complexity is $\mathcal{O}(m(nS_{1,d} + s^2 S_{n,d'} + (n^2 + s^2)M_{n,2d'+d}))$ when $\prec = \prec_{\text{lex}}$, where $S_{n,d}$ denotes the complexity of sampling an $n$-variate polynomial with total degree at most $d$, and $M_{n,d}$ denotes that of multiplying two $n$-variate polynomials with total degree at most $d$. If $\prec \neq \prec_{\text{lex}}$, $\mathcal{O}(mnd^3)$ is additionally needed.*

*Proof.* Outside of the Zariski subset part, statements 1–3 are trivial from Alg. 1 and the discussion in Sec.s 4.2 and 4.3. To obtain the desired Zariski subsets, we consider the vector space of polynomials of degree $d + 2d'$ or less. We remark that if $F_i$ is a $\prec$-Gröbner basis, its leading terms must belong to a finite amount of possibilities. For a polynomial to have a given term as its leading term, zero conditions on terms greater than this term are needed, defining a closed Zariski subset condition. By considering the finite union of all these conditions, we obtain the desired result.

To obtain one pair $(F, G)$, the random generation of $G$ needs $\mathcal{O}(nS_{1,d})$, and the backward transform from $G$ to $F$ needs $\mathcal{O}(s^2 S_{n,d'})$ to get $U_1, U_2$ and $(n^2 + s^2)M_{n,2d'+d}$ for the multiplication $F = U_1 P U_2 G$. Note that the maximum total degree of polynomials in $F$ is $2d' + d$. $\square$

# D    Hybrid Input Embedding

We here present the supplemental information of Transformers with hybrid input embedding. Let $\boldsymbol{s} = [s_1, \ldots, s_L]$ to be a sequence of tokens. Some of these tokens are in $\mathcal{V}$ and otherwise in $\mathbb{R}$. We call the former discrete tokens and the latter continuous tokens. For discrete tokens, the standard input embedding based on the embedding matrix is applied. For continuous tokens, a small feed-forward network $f_{\text{E}} : \mathbb{R} \rightarrow \mathbb{R}^D$ is applied. Unlike discrete tokens, continuous tokens are predicted by regression. For this sake, Transformers should equip a regression head, and they solve a classification task and a regression task simultaneously. In the classification task, continuous tokens are all replaced with a single coefficient token. In other words, the classification head predicts the support of the polynomials, while the regression head predicts the coefficients to be filled in the coefficient tokens. The auto-regressive generation process is naturally induced by a standard method.

In our experiments, we implemented $f_{\text{E}}$ by one-hidden layer ReLU Network, i.e.,

$$f_{\text{E}}(x) = W_2 \varphi(\boldsymbol{w}_1 x + \boldsymbol{b}_1) + \boldsymbol{b}_2, \tag{D.1}$$

where $\boldsymbol{w}_1, \boldsymbol{b_1} \in \mathbb{R}^D, W_2 \in \mathbb{R}^{D \times D}, \boldsymbol{b}_2 \in \mathbb{R}^D$ and $\varphi$ is the ReLU function applied entry-wise. We also tried $f_{\text{E}}$ with one more hidden layer. However, this only has a minor improvement on the $\mathbb{R}$ case; see Tab. 3.

# E    Training Setup

This section provides the supplemental information of our experiments presented in Sec. 6.

---

[11]This can happen with probability zero if $k$ is infinite and very low probability over a large finite field.

Table 3: Comparison of implementations of continuous input embedding $f_{\mathrm{E}}$ on $\mathcal{D}_2^-(k)$.

| Coefficient | $\mathbb{Q}$ | $\mathbb{F}_7$ | $\mathbb{F}_{31}$ | $\mathbb{R}$ |
|---|---|---|---|---|
| one hidden layer | 66.8 / 87.3 | 54.1 / 78.7 | 6.1 / 75.3 | 57.2 / 85.0 |
| two hidden layers | 67.2 / 87.7 | 54.3 / 78.8 | 5.6 / 75.7 | 56.2 / 83.6 |

### E.1 Gröbner basis computation algorithms

In Tab. 1, we tested three algorithms provided in SageMath with the libSingular backend for forward generation.

**STD** (`libsingular:std`): The standard Buchberger algorithm.

**SLIMGB** (`libsingular:slimgb`): A variant of the Faugère's F4 algorithm. Refer to [15].

**STDFGLM** (`libsingular:stdfglm`): Fast computation using STD with the graded reverse lexicographic order followed by the FGLM for the change of term orders. Only for 0-dimensional cases.

### E.2 Training setup

**Dataset.** Both training and test samples are generated using our method. It involves sampling of random polynomials. The degree and the support size (the number of terms) of them as well as coefficients are restricted by user-defined bounds (see Sec. 6.1). Let $d_{\max}, \mu_{\max}$ be the maximum degree and the maximum support size. A random polynomial is obtained as the sum of $\mu \sim \mathbb{U}[1, \mu_{\max}]$ monomials uniformly and randomly sampled from $k[x_1, \ldots, x_n]_{\leq d_{\max}}$. When the samples are fed to a Transformer, polynomials are tokenized into an infix representation. For example, $\{x^2 - 1/2y, y\} \subset \mathbb{Q}[x, y]$ is tokenized to [C1, E2, E0, +, C-1, /, C2, E0, E1, <sep>, C1, E0, E1].

**Training.** We used a Transformer model [85] with a standard architecture: 6 encoder/decoder layers, 8 attention heads, token embedding dimension of 512 dimensions, and feed-forward networks with 2048 inner dimensions. The absolute positional embedding is learned from scratch. The dropout rate was set to 0.1. We used the AdamW optimizer [65] with $(\beta_1, \beta_2) = (0.9, 0.999)$ with no weight decay. The learning rate was initially set to $10^{-4}$ and then linearly decayed over training steps. All training samples are visited in a single epoch, and the total number of epochs was set to 8. The batch size was set to 16. At the inference time, output sequences are generated using a beam search with width 1. For the hybrid input embedding, we used a ReLU network with one hidden layer (cf. Sec. D). A model with this embedding predicts coefficients as continuous values, and the mean-squared loss with weight 0.01 is additionally used for the training. Note that while the exponents are also numbers, we treat them as discrete tokens because they are always discrete and their range is moderate.

## F Additional Experimental Results.

We provide the additional experimental results with Transformers with the standard input embedding.

### F.1 Dataset generation

The runtime comparison for datasets with and without density control is given in Tab. 4, and the success rate (i.e., not encountering the timeout) is given in Tab. 5. The generation of density-controlled datasets $\mathcal{D}_n^-(k)$ (1,000 samples) requires less runtime, and the proposed method is not always the fastest. However, it is important to remember that the forward method is still not feasible because of the difficulty in sampling overdetermined non-Gröbner sets (i.e., $F$s). Generally, such $F$ only leads to a trivial ideal with Gröbner basis $\{1\}$. The profile of datasets is given in Tab. 6.

Table 4: Runtime comparison (in seconds) of forward generation (F.) and backward generation (B.) of dataset $\mathcal{D}_n(k)$ of size 1,000. The forward generation used either of the three algorithms provided in SageMath with the libSingular backend. For $\mathcal{D}_n^-(k)$, the proposed method is not necessarily the fastest. Note that the runtime of the forward methods does not include the sampling of $F$, and $F$ is given from the datasets constructed by the backward method. The sampling step roughly consists of 30% of the runtime in the backward method. It is also worth noting that sampling of overdetermined $F$ generally leads to a trivial ideal with the Gröbner basis $\{1\}$.

| Method | $\mathcal{D}_n(\mathbb{Q})$ | | | | $\mathcal{D}_n^-(\mathbb{Q})$ | | | |
|---|---|---|---|---|---|---|---|---|
| | $n=2$ | $n=3$ | $n=4$ | $n=5$ | $n=2$ $\sigma=1.0$ | $n=3$ $\sigma=0.6$ | $n=4$ $\sigma=0.3$ | $n=5$ $\sigma=0.2$ |
| F. (STD) | **4.20** | 216.3 | 740.1 | 1411.1 | **4.20** | 104.3 | 101.0 | 117.4 |
| F. (SLIMGB) | 4.29 | 183.4 | 697.5 | 1322.7 | 4.29 | 77.1 | 98.9 | 134.5 |
| F. (STDFGLM) | 7.22 | 8.29 | 21.0 | 164.3 | 7.22 | 12.1 | 9.75 | 14.9 |
| B. (ours) | 5.23 | **5.46** | **7.05** | **7.91** | 5.23 | **11.2** | **7.85** | **13.7** |

| Method | $\mathcal{D}_n(\mathbb{F}_7)$ | | | | $\mathcal{D}_n^-(\mathbb{F}_7)$ | | | |
|---|---|---|---|---|---|---|---|---|
| | $n=2$ | $n=3$ | $n=4$ | $n=5$ | $n=2$ $\sigma=1.0$ | $n=3$ $\sigma=0.6$ | $n=4$ $\sigma=0.3$ | $n=5$ $\sigma=0.2$ |
| F. (STD) | 4.93 | **4.57** | 818.9 | 2123.3 | 4.93 | **4.30** | 48.8 | 91.5 |
| F. (SLIMGB) | **4.92** | 5.57 | 561.0 | 1981.2 | **4.92** | 4.65 | 32.9 | 81.8 |
| F. (STDFGLM) | 8.02 | 6.33 | **9.20** | 62.6 | 8.02 | 7.50 | **7.25** | **7.46** |
| B. (ours) | 6.79 | 8.36 | 10.5 | **14.2** | 6.79 | 8.72 | 10.5 | 14.5 |

| Method | $\mathcal{D}_n(\mathbb{F}_{31})$ | | | | $\mathcal{D}_n^-(\mathbb{F}_{31})$ | | | |
|---|---|---|---|---|---|---|---|---|
| | $n=2$ | $n=3$ | $n=4$ | $n=5$ | $n=2$ $\sigma=1.0$ | $n=3$ $\sigma=0.6$ | $n=4$ $\sigma=0.3$ | $n=5$ $\sigma=0.2$ |
| F. (STD) | 5.08 | **5.04** | 777.6 | 2110.4 | 5.08 | **4.39** | 20.6 | 114.0 |
| F. (SLIMGB) | **5.07** | 6.91 | 664.2 | 2026.0 | **5.07** | 4.98 | 22.2 | 103.2 |
| F. (STDFGLM) | 8.10 | 6.73 | **9.14** | 80.2 | 8.10 | 6.95 | **7.23** | **8.58** |
| B. (ours) | 7.40 | 8.37 | 10.5 | **14.7** | 7.40 | 18.0 | 9.91 | 15.3 |

## F.2 Success and failure cases

Tables 7–18 show examples of success cases. One can see that Transformers can solve many non-trivial instances. Tables 19 and 21 show examples of failure cases. One can see that, interestingly, the incorrect predictions appear reasonable. Examples are all taken according to their order in each dataset (i.e., no cherry-picking).

## F.3 Superiority of Transformer in several cases.

Approaching Gröbner basis computation using a Transformer has a potential advantage in the runtime because the computational cost has less dependency on the problem difficulty than mathematical algorithms do. However, currently, mathematical algorithms run faster than Transformers because of our naive input scheme. Nevertheless, we observed several examples in our $\mathcal{D}_n^-(k)$ datasets for which Transformers generate the solutions efficiently, while mathematical algorithms take significantly longer time or encounter a timeout.

Particularly, we examined the examples in $\mathcal{D}_n^-(k)$ where several forward methods encounter a timeout with the five-second budget, see Tab. 5. We fed these examples to Transformer and the three forward algorithms again, but now with a 100-second budget. For such examples, as shown in Tab. 22, Transformers completed the computation in less than a second, while the two forward algorithms, STD and SLIMGB, often used longer computation time or encountered a timeout.

It is worth noting that STD and SLIMGB are general-purpose algorithms, while STDFGLM is specially designed for the zero-dimension ideals. To summarize, Transformers successfully computed Gröbner bases with much less runtime than general-purpose algorithms for several examples. This shows a

Table 5: Success rate [%] of forward generation with the five-second timeout limit.

| Method | $\mathcal{D}_n(\mathbb{Q})$ | | | | $\mathcal{D}_n^-(\mathbb{Q})$ | | | |
|---|---|---|---|---|---|---|---|---|
| | $n=2$ | $n=3$ | $n=4$ | $n=5$ | $n=2$ $\sigma=1.0$ | $n=3$ $\sigma=0.6$ | $n=4$ $\sigma=0.3$ | $n=5$ $\sigma=0.2$ |
| F. (STD) | 100.0 | 96.0 | 85.5 | 72.2 | 100.0 | 98.3 | 98.2 | 97.9 |
| F. (SLIMGB) | 100.0 | 96.6 | 86.4 | 74.4 | 100.0 | 98.7 | 98.3 | 97.6 |
| F. (STDFGLM) | 100.0 | 100.0 | 99.9 | 98.4 | 100.0 | 100.0 | 100.0 | 100.0 |

| Method | $\mathcal{D}_n(\mathbb{F}_7)$ | | | | $\mathcal{D}_n^-(\mathbb{F}_7)$ | | | |
|---|---|---|---|---|---|---|---|---|
| | $n=2$ | $n=3$ | $n=4$ | $n=5$ | $n=2$ $\sigma=1.0$ | $n=3$ $\sigma=0.6$ | $n=4$ $\sigma=0.3$ | $n=5$ $\sigma=0.2$ |
| F. (STD) | 100.0 | 100.0 | 84.8 | 58.7 | 100.0 | 100.0 | 99.3 | 98.4 |
| F. (SLIMGB) | 100.0 | 100.0 | 91.6 | 62.9 | 100.0 | 100.0 | 99.7 | 98.7 |
| F. (STDFGLM) | 100.0 | 100.0 | 100.0 | 100.0 | 100.0 | 100.0 | 100.0 | 100.0 |

| Method | $\mathcal{D}_n(\mathbb{F}_{31})$ | | | | $\mathcal{D}_n^-(\mathbb{F}_{31})$ | | | |
|---|---|---|---|---|---|---|---|---|
| | $n=2$ | $n=3$ | $n=4$ | $n=5$ | $n=2$ $\sigma=1.0$ | $n=3$ $\sigma=0.6$ | $n=4$ $\sigma=0.3$ | $n=5$ $\sigma=0.2$ |
| F. (STD) | 100.0 | 100.0 | 86.0 | 59.2 | 100.0 | 100.0 | 99.7 | 98.0 |
| F. (SLIMGB) | 100.0 | 100.0 | 89.6 | 62.3 | 100.0 | 100.0 | 99.8 | 98.5 |
| F. (STDFGLM) | 100.0 | 100.0 | 100.0 | 100.0 | 100.0 | 100.0 | 100.0 | 100.0 |

Table 6: Dataset profiles. The standard deviation is shown in the superscript.

| Metric | $\mathcal{D}_n(\mathbb{Q})$ | | | | $\mathcal{D}_n^-(\mathbb{Q})$ | | | |
|---|---|---|---|---|---|---|---|---|
| | $n=2$ | $n=3$ | $n=4$ | $n=5$ | $n=2$ $\sigma=1.0$ | $n=3$ $\sigma=0.6$ | $n=4$ $\sigma=0.3$ | $n=5$ $\sigma=0.2$ |
| Size of $F$ | $2.57^{(\pm0.71)}$ | $3.46^{(\pm0.66)}$ | $4.40^{(\pm0.62)}$ | $5.37^{(\pm0.60)}$ | $2.57^{(\pm0.71)}$ | $3.71^{(\pm0.77)}$ | $4.86^{(\pm0.80)}$ | $5.90^{(\pm0.81)}$ |
| Max degree in $F$ | $7.31^{(\pm1.91)}$ | $8.54^{(\pm1.44)}$ | $9.02^{(\pm1.26)}$ | $9.17^{(\pm1.22)}$ | $7.31^{(\pm1.91)}$ | $8.20^{(\pm1.62)}$ | $8.34^{(\pm1.53)}$ | $8.46^{(\pm1.46)}$ |
| Min degree in $F$ | $4.09^{(\pm1.93)}$ | $4.45^{(\pm1.92)}$ | $4.75^{(\pm1.89)}$ | $4.96^{(\pm1.85)}$ | $4.09^{(\pm1.93)}$ | $3.96^{(\pm2.00)}$ | $3.54^{(\pm2.09)}$ | $3.38^{(\pm2.08)}$ |
| # of terms in $F$ | $15.46^{(\pm7.67)}$ | $23.86^{(\pm7.97)}$ | $33.18^{(\pm8.25)}$ | $42.70^{(\pm8.74)}$ | $15.46^{(\pm7.67)}$ | $24.36^{(\pm9.15)}$ | $32.36^{(\pm10.39)}$ | $40.32^{(\pm11.38)}$ |
| Gröbner ratio | $0.001^{(\pm0.026)}$ | $0^{(\pm0)}$ | $0^{(\pm0)}$ | $0^{(\pm0)}$ | $0.001^{(\pm0.026)}$ | $0^{(\pm0.012)}$ | $0^{(\pm0)}$ | $0^{(\pm0)}$ |
| Size of $G$ | $2.00^{(\pm0)}$ | $3.00^{(\pm0)}$ | $4.00^{(\pm0)}$ | $5.00^{(\pm0)}$ | $2.00^{(\pm0)}$ | $3.00^{(\pm0)}$ | $4.00^{(\pm0)}$ | $5.00^{(\pm0)}$ |
| Max degree in $G$ | $4.00^{(\pm1.32)}$ | $4.00^{(\pm1.32)}$ | $4.00^{(\pm1.32)}$ | $4.00^{(\pm1.32)}$ | $4.00^{(\pm1.32)}$ | $4.00^{(\pm1.32)}$ | $4.00^{(\pm1.32)}$ | $4.00^{(\pm1.32)}$ |
| Min degree in $G$ | $2.47^{(\pm1.23)}$ | $2.07^{(\pm1.14)}$ | $1.79^{(\pm1.02)}$ | $1.60^{(\pm0.90)}$ | $2.47^{(\pm1.23)}$ | $2.07^{(\pm1.14)}$ | $1.79^{(\pm1.02)}$ | $1.60^{(\pm0.90)}$ |
| # of terms in $G$ | $6.46^{(\pm2.33)}$ | $8.93^{(\pm3.25)}$ | $11.40^{(\pm4.13)}$ | $13.86^{(\pm4.99)}$ | $6.46^{(\pm2.33)}$ | $8.93^{(\pm3.24)}$ | $11.39^{(\pm4.13)}$ | $13.87^{(\pm4.99)}$ |
| Gröbner ratio | $1^{(\pm0)}$ | $1^{(\pm0)}$ | $1^{(\pm0)}$ | $1^{(\pm0)}$ | $1^{(\pm0)}$ | $1^{(\pm0)}$ | $1^{(\pm0)}$ | $1^{(\pm0)}$ |

| Metric | $\mathcal{D}_n(\mathbb{F}_7)$ | | | | $\mathcal{D}_n^-(\mathbb{F}_7)$ | | | |
|---|---|---|---|---|---|---|---|---|
| | $n=2$ | $n=3$ | $n=4$ | $n=5$ | $n=2$ $\sigma=1.0$ | $n=3$ $\sigma=0.6$ | $n=4$ $\sigma=0.3$ | $n=5$ $\sigma=0.2$ |
| Size of $F$ | $3.00^{(\pm0.82)}$ | $4.00^{(\pm0.82)}$ | $5.00^{(\pm0.82)}$ | $6.00^{(\pm0.82)}$ | $3.00^{(\pm0.82)}$ | $4.00^{(\pm0.82)}$ | $5.00^{(\pm0.82)}$ | $6.00^{(\pm0.82)}$ |
| Max degree in $F$ | $7.91^{(\pm2.04)}$ | $8.45^{(\pm1.67)}$ | $8.43^{(\pm1.55)}$ | $8.51^{(\pm1.47)}$ | $7.91^{(\pm2.04)}$ | $8.45^{(\pm1.67)}$ | $8.43^{(\pm1.55)}$ | $8.51^{(\pm1.47)}$ |
| Min degree in $F$ | $4.37^{(\pm2.06)}$ | $4.15^{(\pm2.07)}$ | $3.64^{(\pm2.13)}$ | $3.44^{(\pm2.11)}$ | $4.37^{(\pm2.06)}$ | $4.15^{(\pm2.07)}$ | $3.64^{(\pm2.13)}$ | $3.44^{(\pm2.11)}$ |
| # of terms in $F$ | $19.88^{(\pm9.62)}$ | $27.56^{(\pm10.42)}$ | $34.02^{(\pm11.07)}$ | $41.50^{(\pm11.90)}$ | $19.88^{(\pm9.62)}$ | $27.56^{(\pm10.42)}$ | $34.02^{(\pm11.07)}$ | $41.50^{(\pm11.90)}$ |
| Gröbner ratio | $0.002^{(\pm0.045)}$ | $0^{(\pm0.011)}$ | $0^{(\pm0)}$ | $0^{(\pm0)}$ | $0.002^{(\pm0.045)}$ | $0^{(\pm0.011)}$ | $0^{(\pm0)}$ | $0^{(\pm0)}$ |
| Size of $G$ | $2.00^{(\pm0)}$ | $3.00^{(\pm0)}$ | $4.00^{(\pm0)}$ | $5.00^{(\pm0)}$ | $2.00^{(\pm0)}$ | $3.00^{(\pm0)}$ | $4.00^{(\pm0)}$ | $5.00^{(\pm0)}$ |
| Max degree in $G$ | $3.94^{(\pm1.34)}$ | $3.93^{(\pm1.34)}$ | $3.93^{(\pm1.34)}$ | $3.94^{(\pm1.34)}$ | $3.94^{(\pm1.34)}$ | $3.93^{(\pm1.34)}$ | $3.93^{(\pm1.34)}$ | $3.94^{(\pm1.34)}$ |
| Min degree in $G$ | $2.39^{(\pm1.22)}$ | $1.98^{(\pm1.11)}$ | $1.72^{(\pm0.97)}$ | $1.53^{(\pm0.84)}$ | $2.39^{(\pm1.22)}$ | $1.98^{(\pm1.11)}$ | $1.72^{(\pm0.97)}$ | $1.53^{(\pm0.84)}$ |
| # of terms in $G$ | $6.32^{(\pm2.33)}$ | $8.70^{(\pm3.23)}$ | $11.08^{(\pm4.10)}$ | $13.47^{(\pm4.94)}$ | $6.32^{(\pm2.33)}$ | $8.70^{(\pm3.23)}$ | $11.08^{(\pm4.10)}$ | $13.47^{(\pm4.94)}$ |
| Gröbner ratio | $1^{(\pm0)}$ | $1^{(\pm0)}$ | $1^{(\pm0)}$ | $1^{(\pm0)}$ | $1^{(\pm0)}$ | $1^{(\pm0)}$ | $1^{(\pm0)}$ | $1^{(\pm0)}$ |

| Metric | $\mathcal{D}_n(\mathbb{F}_{31})$ | | | | $\mathcal{D}_n^-(\mathbb{F}_{31})$ | | | |
|---|---|---|---|---|---|---|---|---|
| | $n=2$ | $n=3$ | $n=4$ | $n=5$ | $n=2$ $\sigma=1.0$ | $n=3$ $\sigma=0.6$ | $n=4$ $\sigma=0.3$ | $n=5$ $\sigma=0.2$ |
| Size of $F$ | $3.00^{(\pm0.82)}$ | $4.00^{(\pm0.82)}$ | $5.00^{(\pm0.82)}$ | $6.00^{(\pm0.82)}$ | $3.00^{(\pm0.82)}$ | $4.00^{(\pm0.82)}$ | $5.00^{(\pm0.82)}$ | $6.00^{(\pm0.82)}$ |
| Max degree in $F$ | $8.11^{(\pm2.02)}$ | $8.65^{(\pm1.65)}$ | $8.62^{(\pm1.54)}$ | $8.69^{(\pm1.46)}$ | $8.11^{(\pm2.02)}$ | $8.65^{(\pm1.65)}$ | $8.62^{(\pm1.54)}$ | $8.69^{(\pm1.46)}$ |
| Min degree in $F$ | $4.55^{(\pm2.06)}$ | $4.33^{(\pm2.08)}$ | $3.81^{(\pm2.16)}$ | $3.61^{(\pm2.15)}$ | $4.55^{(\pm2.06)}$ | $4.33^{(\pm2.08)}$ | $3.81^{(\pm2.16)}$ | $3.61^{(\pm2.15)}$ |
| # of terms in $F$ | $20.46^{(\pm9.74)}$ | $28.36^{(\pm10.52)}$ | $35.00^{(\pm11.19)}$ | $42.69^{(\pm12.00)}$ | $20.46^{(\pm9.74)}$ | $28.36^{(\pm10.52)}$ | $35.00^{(\pm11.19)}$ | $42.69^{(\pm12.00)}$ |
| Gröbner ratio | $0^{(\pm0.017)}$ | $0^{(\pm0.009)}$ | $0^{(\pm0.001)}$ | $0^{(\pm0)}$ | $0^{(\pm0.017)}$ | $0^{(\pm0.009)}$ | $0^{(\pm0.001)}$ | $0^{(\pm0)}$ |
| Size of $G$ | $2.00^{(\pm0)}$ | $3.00^{(\pm0)}$ | $4.00^{(\pm0)}$ | $5.00^{(\pm0)}$ | $2.00^{(\pm0)}$ | $3.00^{(\pm0)}$ | $4.00^{(\pm0)}$ | $5.00^{(\pm0)}$ |
| Max degree in $G$ | $4.07^{(\pm1.31)}$ | $4.07^{(\pm1.31)}$ | $4.06^{(\pm1.31)}$ | $4.07^{(\pm1.31)}$ | $4.07^{(\pm1.31)}$ | $4.07^{(\pm1.31)}$ | $4.06^{(\pm1.31)}$ | $4.07^{(\pm1.31)}$ |
| Min degree in $G$ | $2.56^{(\pm1.24)}$ | $2.16^{(\pm1.18)}$ | $1.88^{(\pm1.07)}$ | $1.68^{(\pm0.95)}$ | $2.56^{(\pm1.24)}$ | $2.16^{(\pm1.18)}$ | $1.88^{(\pm1.07)}$ | $1.68^{(\pm0.95)}$ |
| # of terms in $G$ | $6.63^{(\pm2.33)}$ | $9.18^{(\pm3.26)}$ | $11.74^{(\pm4.15)}$ | $14.30^{(\pm5.03)}$ | $6.63^{(\pm2.33)}$ | $9.18^{(\pm3.26)}$ | $11.74^{(\pm4.15)}$ | $14.30^{(\pm5.03)}$ |
| Gröbner ratio | $1^{(\pm0)}$ | $1^{(\pm0)}$ | $1^{(\pm0)}$ | $1^{(\pm0)}$ | $1^{(\pm0)}$ | $1^{(\pm0)}$ | $1^{(\pm0)}$ | $1^{(\pm0)}$ |

potential advantage of using a Transformer in Gröbner basis computation, particularly for large-scale problems.

### F.4 Generalization to out-distribution samples

Handling out-distribution samples is beyond the scope of the current work. Several studies of using a Transformer for math problems (e.g., integer multiplication [30] and linear algebra [19]) addressed out-distribution generalization by controlling the training sample distribution. This is because these problems are of moderate difficulty, and naive training sample generation methods exist. However, this is not the case for our task, and we thus focused on the problem of efficient generation of training samples in this paper. Nevertheless, we consider that presenting a limitation of our work by experiments is helpful for future work. Thus, we here present that the Transformers trained on our datasets certainly fail on out-distribution samples through several cases.

**Out-distribution samples.** We generate additional datasets $\mathcal{D}_n^{\mathrm{u}}(k)$ for $k \in \{\mathbb{Q}, \mathbb{F}_7, \mathbb{F}_{31}\}$. These datasets are generated as $\mathcal{D}_n(k)$ with a slight difference in sampling of random polynomials. Originally, a (degree-bounded) random polynomial is obtained using monomials randomly sampled from $k[x_1, \ldots, x_n]_{\leq d_{\max}}$. Since there are more high-degree terms than low-degree ones, random polynomials are more likely to be high-degree. In $\mathcal{D}_n^{\mathrm{u}}(k)$, we instead uniformly sample the degree-bound $d$ from $\mathbb{U}[1, d_{\max}]$, and then conduct the sampling of monomials. As Tab. 23 shows, this change in the sampling strategy causes some distribution shifts. Table 24 shows the prediction accuracy and support accuracy on the new datasets. As can be seen, the accuracy drops when the base field is a finite field $\mathbb{F}_p$. When it is $\mathbb{Q}$, the accuracy drop is moderate.

**Katsura-n.** Katsura-n is a typical benchmarking example for Gröbner basis computation algorithms ("-n" denotes the number of variables). Table 25 shows a list of examples for different $n \in \{2, 3, 4\}$ and $k \in \{\mathbb{Q}, \mathbb{F}_7, \mathbb{F}_{31}\}$. While the Gröbner bases in Katsura-n have a form of Eq. (4.1), one can readily find its qualitative difference in the non-Gröbner sets from those in our $\mathcal{D}_n(k)$ datasets (cf. Tables 7–18). For example, non-Gröbner sets in Katsura-n consist of low-degree sparse polynomials, whereas those in $\mathcal{D}_n(k)$ are not because of the generation processes (i.e., the product and sum of polynomials through the multiplication by polynomial matrices $U_1, U_2$). Therefore, Katsura-n examples are greatly out-distributed samples for our Transformers. We fed these samples to the trained Transformers, but only obtained the output sequences that cannot be transformed back to polynomials because of their invalid prefix representation.

As Gröbner basis computation is an NP-hard problem, it may not be a good idea to peruse a general solver via learning. Instead, we should ultimately aim at a solver for large-scale but specialized cases. Thus, the generation of application-specific training samples and the pursuit of in-distribution accuracy will be a future direction. From this perspective, existing mathematical benchmark examples may not be practical because they are mostly artificial, empirically found difficult, and/or designed for easily generating variations in the number of variables $n$.[12] They are useful for math algorithms, i.e., the algorithms proved to work for all the cases (i.e., 100% in/out-distribution samples), but not for our current work because they are out-distribution samples.

## G  Buchberger–Möller Algorithm for Prob. 4.1

Here, we discuss another approach for Prob. 4.1 using the Buchberger–Möller (BM) algorithm [2, 68]. Although we did not adopt this approach, we include this for completeness as many variants have been recently developed and applied extensively in machine learning and other data-centric applications.

Given a set of points $\mathbb{X} \subset k^n$ and a graded term order, the BM algorithm computes a Gröbner basis of its vanishing ideal $I(\mathbb{X}) = \{g \in k[x_1, \ldots, x_n] \mid g(p) = 0, \forall p \in \mathbb{X}\}$. While several variants follow in computational algebra [1, 32, 40, 48, 49, 54, 62], interestingly, it is also recently tailored for machine learning [42, 51–53, 56, 63, 90, 91]. Various applications have followed such as machine learning [77, 92], signal processing [86–88], nonlinear dynamics [46, 47, 50], and more [5, 43, 55]. Such broad applications derived from the distinguishing design of the BM algorithm: unlike most computer-algebraic algorithms, it takes a set of points (i.e., dataset) as input, not a set of polynomials.

---

[12]For example, refer to `https://www-sop.inria.fr/coprin/logiciels/ALIAS/Benches/node1.html`.

Therefore, to address Prob. 4.1, one may consider using the BM algorithm or its variants, e.g., by running the BM algorithm $m$ times while sampling diverse sets of points. An important caveat is that Gröbner bases that can be given by the BM algorithm may be more restrictive than those considered in the main text (i.e., the Gröbner bases of ideals in shape position). For example, the former generates the largest ideals that have given $k$-rational points for their roots, whereas this is not the case for the latter. Another drawback of using the BM algorithm is its large computational cost. The time complexity of the BM algorithm is $\mathcal{O}(n \cdot |\mathbb{X}|^3)$. Furthermore, we need $\mathcal{O}(n^d)$ points to obtain a Gröbner basis that includes a polynomial of degree $d$ in the average case. Therefore, the BM algorithm does not fit our settings that a large number of Gröbner bases are needed (i.e., $m \approx 10^6$). Accelerating the BM algorithm by reusing the results of runs instead of independently running the algorithm many times can be interesting for future work.

## H    Open Questions

**Random generation of Gröbner bases**    To our knowledge, no prior studies addressed this problem. Our study focuses on generic 0-dimensional ideals. These ideals are generally in shape position, and a simple sampling strategy is available. However, some applications may be interested in other classes of ideals (e.g., positive dimensional or binomial ones) or a particular subset of 0-dimensional ideals (e.g., those associated with a single solution in the coefficient field). The former case is an open problem. The latter case may be addressed by the Buchberger–Möller algorithms [68] (cf. App. G).

**Backward Gröbner problem.**    Machine learning models perform better for in-distribution samples than out-distribution samples. Thus, it is essential to design a training sample distribution that is close to one's use case. As noted in Sec. 4.1, polynomial systems (either Gröbner basis or not) take a domain-specific form. Backward generation gives us control over Gröbner bases but not for non-Gröbner sets. Hence, we need a well-tailored backward generation method specialized to an application, as the specialized Gröbner basis computation algorithms in computational algebra. This paper addressed a generic case. Prop. 4.7 states that any matrix $A \in \mathrm{SL}_n(k[x_1, \ldots, x_n])$ satisfies Prob. 4.5. This raises two sets of open questions: (i) *are there matrices outside* $\mathrm{SL}_n(k[x_1, \ldots, x_n])$ *satisfying Prob. 4.5? Can we sample them?* and (ii) *is it possible to efficiently sample matrices of* $\mathrm{SL}_n(k[x_1, \ldots, x_n])$? To efficiently generate our dataset, we have restricted ourselves to sampling matrices having a Bruhat decomposition (see Eq. (4.2)), which is a strict subset of $\mathrm{SL}_n(k[x_1, \ldots, x_n])$. Sampling matrices in $\mathrm{SL}_n(k[x_1, \ldots, x_n])$ remains an open question. Thanks to Suslin's stability theorem and its algorithmic proofs [64, 72, 81], $\mathrm{SL}_n(k[x_1, \ldots, x_n])$ is generated by elementary matrices and a decomposition into a product of elementary matrices can be computed algorithmically. One may hope to use sampling of elementary matrices to sample matrices of $\mathrm{SL}_n(k[x_1, \ldots, x_n])$. It is unclear whether this can be efficient as many elementary matrices are needed [64].

**Distribution analysis.**    Several studies have reported that careful design of a training set leads to a better generalization of Transformers [19, 21]. Algebraically, analyzing the distribution of Gröbner bases and the non-Gröbner sets is challenging, particularly when some additional structures (e.g., sparsity) are injected. Thus, the first step may be to investigate the generic case (i.e., dense polynomials). In this case, Thm. 4.6(3) is helpful to design an algorithm that is certified to be able to yield all possible $G$ and $F$ almost uniformly. While dataset generation algorithms should run efficiently for practicality, a solid analysis may be of independent interest in computational algebra.

**Long mathematical expressions.**    As in most of the prior studies, we used a Transformer of a standard architecture to see the necessity of a specialized model. From the accuracy aspect, such a vanilla model may be sufficient for $\mathbb{Q}[x_1, \ldots, x_n]$ but not for $\mathbb{F}_p[x_1, \ldots, x_n]$. From the efficiency perspective, the quadratic cost of the attention mechanism prevents us from scaling up the problem size. For large $n$, both non-GB sets and Gröbner bases generally consist of many dense polynomials, leading to long input and output sequences. Unlike natural language processing tasks, the input sequence cannot be split as all the symbols are related in mathematical tasks. It is worth studying several attention mechanisms that work with sub-quadratic memory cost [11, 23, 28, 57, 80] can be introduced with a small degradation of performance even for mathematical sequences, which have a different nature from natural language (e.g., a single modification of the input sequence can completely change the solution of the problem).

Table 7: Success examples from the $\mathcal{D}_2^-(\mathbb{Q})$ test set.

| ID | $F$ | $G$ |
|---|---|---|
| 0 | $f_1 = x_0 + 5/2x_1^4 - 2x_1^3 + 5x_1^2 - 2/3x_1 + 1/5$ 
 $f_2 = 3/5x_0^2x_1^2 + 1/5x_0^2x_1 + 3/2x_0x_1^6 - 7/10x_0x_1^5 + 13/5x_0x_1^4 + 3/5x_0x_1^3 - 1/75x_0x_1^2 + 1/25x_0x_1 + x_1^5 - 5/4x_1^3 + 5x_1$ | $g_1 = x_0 + 5/2x_1^4 - 2x_1^3 + 5x_1^2 - 2/3x_1 + 1/5$ 
 $g_2 = x_1^5 - 5/4x_1^3 + 5x_1$ |
| 1 | $f_1 = x_0x_1^3 + x_1^5 + 3x_1^4 - x_1 + 2/3$ 
 $f_2 = 1/3x_0^2x_1^2 + 4x_0x_1^7 + 8x_0x_1^6 - 2x_0x_1^5 - 11/3x_0x_1^3 + 8/3x_0x_1^2 - 2x_1^7 - 6x_1^6 + 2x_1^3 - 4/3x_1^2$ 
 $f_3 = 4/3x_0^4x_1^2 + 16x_0^3x_1^7 + 32x_0^3x_1^6 - 44/3x_0^3x_1^3 + 32/3x_0^3x_1^2 + 3/5x_0^3x_1 + 13/4x_0^2x_1^5 + 5/2x_0^2x_1^4 - 33/20x_0^2x_1^2 - 5/4x_0^2x_1 + 5/6x_0^2 - x_0x_1^7 + 3/5x_0x_1^6 + 6/5x_0x_1^5 - 5/4x_0x_1^3 - 19/15x_0x_1^2 + 2/5x_0x_1$ 
 $f_4 = -3/25x_0^5x_1 - 1/4x_0^4x_1^5 - 1/2x_0^4x_1^4 + 7/25x_0^4x_1^2 + 1/4x_0^4x_1 - 1/6x_0^4 - 3/25x_0^3x_1^6 - 6/25x_0^3x_1^5 + 13/20x_0^3x_1^3 - 22/25x_0^3x_1^2 - 2/25x_0^3x_1 - 12x_0^2x_1^7 - 24x_0^2x_1^6 + 1/4x_0^2x_1^5 + 3/4x_0^2x_1^4 + 11x_0^2x_1^3 - 8x_0^2x_1^2 - 1/4x_0^2x_1 + 1/6x_0^2 + x_0 + x_1$ | $g_1 = x_0 + x_1$ 
 $g_2 = x_1^5 + 2x_1^4 - x_1 + 2/3$ |
| 2 | $f_1 = 3/4x_0^2x_1^3 - x_0x_1^5 + 2x_0x_1^2 - 1/5x_0x_1 - 2x_1^3 + 1/5x_1^2$ 
 $f_2 = -5/3x_0^3 - 3/2x_0^2x_1^4 + 5/3x_0^2x_1 + 2x_0x_1^6 - 4x_0x_1^3 + 2/5x_0x_1^2 + 4x_1^4 + 3/5x_1^3$ 
 $f_3 = -5/6x_0^5 + 5/6x_0^4x_1 - 5/2x_0^3x_1^3 + 13/4x_0^2x_1^4 + 1/2x_0^2x_1^3 - x_0x_1^6 + 2x_0x_1^3 - 1/5x_0x_1^2 + x_0 + 3/2x_1^6 - 2x_1^4 + 1/5x_1^3 - x_1$ | $g_1 = x_0 - x_1$ 

 $g_2 = x_1^3$ |
| 3 | $f_1 = -5x_0^3x_1 + 4x_0^2x_1 + x_1$ 
 $f_2 = -25/3x_0^4x_1^3 + 20/3x_0^3x_1^3 - 4x_0^2x_1^2 - 5/3x_0^2x_1 + 5/3x_0x_1^3 + 4/3x_0x_1$ 
 $f_3 = -5/2x_0^4x_1^3 + 2x_0^3x_1^3 - 2x_0^2x_1^5 - 5/6x_0^2x_1^4 + 2/3x_0x_1^4 + 1/2x_0x_1^3 + x_0 - 4/5$ | $g_1 = x_0 - 4/5$ 
 $g_2 = x_1$ |
| 4 | $f_1 = x_0 + 5/3x_1^4 + x_1^2 + 5/4x_1 - 5/2$ 
 $f_2 = x_0^3 + 5/3x_0^2x_1^4 + x_0^2x_1^2 + 5/4x_0^2x_1 - 1/2x_0^2 + 5x_0x_1^6 - 25x_0x_1^5 + 10/3x_0x_1^4 + 2x_0x_1^2 + 2x_0x_1 - 5x_0 + 5/6x_1^5 + 1/2x_1^3 + 5/8x_1^2 - 5/4x_1$ 
 $f_3 = x_0^5 + 5/3x_0^4x_1^4 + x_0^4x_1^2 + 5/4x_0^4x_1 - 1/2x_0^4 + 5x_0^3x_1^6 - 25x_0^3x_1^5 + 10/3x_0^3x_1^4 + 2x_0^3x_1^2 + 5/2x_0^3x_1 - 5x_0^3 + 5/3x_0^2x_1^5 + x_0^2x_1^3 + 5/4x_0^2x_1^2 - 5/2x_0^2x_1 + x_0x_1 + 8/3x_1^5 - 5x_1^4 + x_1^3 + 5/4x_1^2 - 5/2x_1 - 1/5$ | $g_1 = x_0 + 5/3x_1^4 + x_1^2 + 5/4x_1 - 5/2$ 
 $g_2 = x_1^5 - 5x_1^4 - 1/5$ |
| 5 | $f_1 = x_0 + 2/5x_1$ 
 $f_2 = -1/3x_0^2x_1^2 - 3/4x_0^2 - 2/15x_0x_1^3 - 1/20x_0x_1 + 11/10x_1^2 - 4$ | $g_1 = x_0 + 2/5x_1$ 
 $g_2 = x_1^2 - 4$ |
| 6 | $f_1 = 3/5x_0^2x_1^2 - x_0x_1^6 + 3/25x_0x_1^4 - 12/25x_0x_1^3 + 3/4x_0 + x_1^5 - 29/20x_1^4 + 19/20x_1^2 - 3/5x_1 - 1/2$ 
 $f_2 = 3/10x_0^2x_1^3 - 1/2x_0x_1^7 + 3/50x_0x_1^5 - 6/25x_0x_1^4 + 3/8x_0x_1 + x_0 + 1/2x_1^6 - 29/40x_1^5 - 5/3x_1^4 + 19/40x_1^3 - 1/10x_1^2 - 21/20x_1$ | $g_1 = x_0 - 5/3x_1^4 + 1/5x_1^2 - 4/5x_1$ 


 $g_2 = x_1^5 - 1/5x_1^4 + 4/5x_1^2 - 1/2$ |
| 7 | $f_1 = x_0 + 5/4x_1^3 - 5/4$ 
 $f_2 = -x_0^3 - 5/4x_0^2x_1^3 + 4x_0^2x_1 + 5/4x_0^2 + 5x_0x_1^4 - 5x_0x_1 + x_1^5 + 1/2x_1^3 + 2/5x_1$ | $g_1 = x_0 + 5/4x_1^3 - 5/4$ 
 $g_2 = x_1^5 + 1/2x_1^3 + 2/5x_1$ |

Table 8: Success examples from the $\mathcal{D}_3^-(\mathbb{Q})$ test set.

| ID | $F$ | $G$ |
|---|---|---|
| 0 | $f_1 = 4x_0x_1^2x_2 + 4/5x_0x_1x_2 + 4/5x_0x_2^2 - x_1^2x_2^2 - 4x_1^2x_2 + x_2^2 - 1$ | $g_1 = x_0 - 1/4x_2 - 1$ |
| | $f_2 = 20x_0x_1^3x_2^2 + 4x_0x_1^2x_2^2 + 4x_0x_1x_2^3 + x_0 - 5x_1^3x_2^3 - 20x_1^3x_2^2 + 5x_1x_2^3 - 5x_1x_2 - 1/4x_2 - 1$ | $g_2 = x_1 + x_2$ |
| | $f_3 = 4/5x_0^3x_1^2x_2^2 + 4/25x_0^3x_1x_2^2 + 4/25x_0^3x_2^3 - 1/5x_0^2x_1^2x_2^3 - 4/5x_0^2x_1^2x_2^2 + 1/5x_0^2x_2^3 - 3/5x_0^2x_2 + 16/3x_0x_1^3x_2^3 + 16/15x_0x_1^2x_2^3 + 16/15x_0x_1x_2^4 + 1/10x_0x_2^2 + 2/5x_0x_2 - 4/3x_1^3x_2^4 - 16/3x_1^3x_2^3 + 4/3x_1x_2^4 - 4/3x_1x_2^2 + x_1 + x_2$ | $g_3 = x_2^2 - 1$ |
| | $f_4 = 1/2x_0^2x_1x_2 + 1/2x_0^2x_2^2 + 20x_0x_1^3x_2^2 + 4x_0x_1^2x_2^2 + 4x_0x_1x_2^3 + 3/5x_0x_1x_2 + 4/3x_0x_1 + 3/5x_0x_2^2 - 5x_1^3x_2^3 - 20x_1^3x_2^2 - 5/4x_1^3x_2 - 5/4x_1^2x_2^2 + 5x_1x_2^3 - 16/3x_1x_2 - 4/3x_1$ | |
| | $f_5 = -1/3x_0^4x_1x_2 - 1/3x_0^4x_2^2 - 2/5x_0^3x_1x_2 - 2/5x_0^3x_2^2 + 16x_0^2x_1^4x_2 + 16/5x_0^2x_1^3x_2 + 16/5x_0^2x_1^2x_2^2 - 4/3x_0^2x_1x_2 - 4x_0x_1^4x_2^2 - 16x_0x_1^4x_2 + 4x_0x_1^2x_2^2 - 4x_0x_1^2 + 5x_0x_1x_2^3 + 1/3x_0x_1x_2^2 - 11/3x_0x_1x_2 + 2/15x_0x_2^3 + 1/5x_0x_2 - x_1^2 - x_1x_2 - 1/12x_2^4 - 1/3x_2^3$ | |
| 2 | $f_1 = x_1 - x_2^2 - 4x_2$ | $g_1 = x_0 + 1/5x_2^2 + 4/3x_2 + 5/3$ |
| | $f_2 = -2/3x_0^2x_1x_2 + 2/3x_0^2x_2^3 + 8/3x_0^2x_2^2 - 1/5x_0x_1x_2^2 - 2/3x_0x_1 - 2/5x_0x_2^4 - 8/5x_0x_2^3 + 2/3x_0x_2^2 + 8/3x_0x_2 - 1/5x_1^2x_2 - 3/25x_1x_2^4 - 3/5x_1x_2^3 - 1/5x_1x_2^2$ | $g_2 = x_1 - x_2^2 - 4x_2$ |
| | $f_3 = -1/2x_0^2x_1^3x_2 + 1/2x_0^2x_1^2x_2^3 + 2x_0^2x_1^2x_2^2 - 2/3x_0^2x_1x_2 + 2/3x_0^2x_2^2 + 8/3x_0^2x_2^2 - 3/20x_0x_1^3x_2^2 - 3/10x_0x_1^2x_2^4 - 6/5x_0x_1^2x_2^3 - 1/5x_0x_1x_2^2 - 2/5x_0x_2^4 - 8/5x_0x_2^3 + x_0 - 9/100x_1^3x_2^4 - 3/5x_1^3x_2^3 - 3/4x_1^3x_2^2 - 3/25x_1x_2^4 + 1/5x_1x_2^3 - 9/5x_1x_2^2 - x_2^5 - 16/5x_2^4 + 16/5x_2^3 + 1/5x_2^2 + 4/3x_2 + 5/3$ | $g_3 = x_2^3 + 3/5x_2^2 - 3/2$ |
| | $f_4 = -1/2x_0^3x_1^2x_2 + 1/2x_0^3x_1x_2^3 + 2x_0^3x_1x_2^2 - x_0^3x_1 - 3/20x_0^2x_1^2x_2^2 - 3/10x_0^2x_1x_2^4 - 6/5x_0^2x_1x_2^3 - 1/5x_0^2x_1x_2^2 - 4/3x_0^2x_1x_2 - 5/3x_0^2x_1 - 9/100x_0x_1^2x_2^4 - 3/5x_0x_1^2x_2^3 - 3/4x_0x_1^2x_2^2 - 1/4x_0x_1^2 + 1/2x_0x_1x_2^2 + 3/2x_0x_1x_2 - 1/2x_0x_2^4 - 7/2x_0x_2^3 - 6x_0x_2^2 - 1/20x_1^2x_2^2 - 1/3x_1^2x_2 - 5/12x_1^2 - x_1x_2 + 2x_2^3 + 23/5x_2^2 - 3/2$ | |
| | $f_5 = 4/9x_0^3x_1^3x_2 + 3x_0^3x_1^3 - 4/9x_0^3x_1^2x_2^3 - 16/9x_0^3x_1^2x_2^2 + 11/15x_0^3x_1^3x_2^2 + 4x_0^3x_1^3x_2 + 5x_0^2x_1^3 + 4/15x_0^2x_1^2x_2^4 + 16/15x_0^2x_1^2x_2^3 - 1/2x_0^2x_1^2 + 2/25x_0x_1^3x_2^4 + 8/15x_0x_1^3x_2^3 - 5/6x_0x_1^3x_2^2 + 3/2x_0x_1^2x_2^4 + 6x_0x_1^2x_2^3 - 1/10x_0x_1^2x_2^2 - 2/3x_0x_1^2x_2 + 2/3x_0x_1^2 - 3/2x_0x_1x_2^2 - 23/3x_0x_1x_2 + 5/3x_0x_2^3 + 20/3x_0x_2^2 - x_0x_2 + 3x_1^3x_2 - 6x_1^2x_2^3 - 69/5x_1^2x_2^2 + 9/2x_1^2 + 1/3x_1x_2^4 + 17/10x_1x_2^3 - 3/4x_1x_2 - 3/2x_2^5 - 6x_2^4 + 1/20x_2^3 - 1/3x_2^2 - 5/3x_2$ | |

Table 9: Success examples from the $\mathcal{D}_4^-(\mathbb{Q})$ test set.

| ID | $F$ | $G$ |
|---|---|---|
| 0 | $f_1 = x_0 + 3x_3^4 - x_3^3 - 3/2x_3$ 
 $f_2 = -3/5x_0^2x_3^2 - 9/5x_0x_3^6 + 3/5x_0x_3^5 + 9/10x_0x_3^3 + x_1 + x_3$ 
 $f_3 = -1/2x_0^2x_1^2 - 3/2x_0x_1^2x_3^4 + 1/2x_0x_1^2x_3^3 + 3/4x_0x_1^2x_3 - x_1^4 - x_1^3x_3 - 4/5x_1x_2^2x_3 - 4/5x_2^2x_3^2 + x_3^5 - x_3^4 + 4/3x_3^2$ 
 $f_4 = x_0^2x_1 + x_0^2x_3 - 1/4x_0x_1x_3^2 - 3/2x_1^3 - 3/2x_1^2x_3 - 3/4x_1x_3^6 + 1/4x_1x_3^5 + 3/8x_1x_3^3 + 1/5x_2x_3^5 - 1/5x_2x_3^4 + 4/15x_2x_3^2 + x_2 - 1/5x_3^4 + 1/4x_3 + 1$ | $g_1 = x_0 + 3x_3^4 - x_3^3 - 3/2x_3$ 
 $g_2 = x_1 + x_3$ 

 $g_3 = x_2 - 1/5x_3^4 + 1/4x_3 + 1$ 

 $g_4 = x_3^5 - x_3^4 + 4/3x_3^2$ |
| 1 | $f_1 = x_1 - 2/5x_3 - 1/2$ 
 $f_2 = x_0x_1^2 - 2/5x_0x_1x_3 - 1/2x_0x_1 - 2/3x_1^3x_3 - 1/3x_1^2x_2x_3 + 4/15x_1^2x_3^2 + 1/3x_1^2x_3 + 2/15x_1x_2x_3^2 + 1/6x_1x_2x_3 + x_3^5 + 1/2x_3^4 - 5/2$ 
 $f_3 = -4/3x_0x_1^3 + 8/15x_0x_1^2x_3 + 2/3x_0x_1^2 + x_0 + 5/2x_1^2x_2 - x_1x_2x_3 - 5/4x_1x_2 - 4/3x_1x_3^5 - 2/3x_1x_3^4 + 1/4x_1x_3 + 10/3x_1 - 1/10x_3^2 + 3/8x_3$ 
 $f_4 = 4x_0^3x_3 + 2x_0^2x_3^2 + x_0x_1^2x_2^2 - 2/5x_0x_1x_2^2x_3 - 1/2x_0x_1x_2^2 + 1/3x_1^3x_2 - 2/15x_1^2x_2x_3 - 1/6x_1^2x_2 - 1/5x_1^2x_3^2 + 2/25x_1x_3^3 + 1/10x_1x_3^2 + x_2^2x_3^5 + 1/2x_2^2x_3^4 - 5/2x_2^2 + x_2 + 4/3x_3^4 + x_3^3 + 4/5x_3^2$ | $g_1 = x_0 + 1/2x_3$ 
 $g_2 = x_1 - 2/5x_3 - 1/2$ 


 $g_3 = x_2 + 4/3x_3^4 + x_3^3 + 4/5x_3^2$ 


 $g_4 = x_3^5 + 1/2x_3^4 - 5/2$ |
| 2 | $f_1 = x_2 + x_3$ 
 $f_2 = -3/2x_0^2x_2 - 3/2x_0^2x_3 - 5/4x_0^2 - 5/3x_0x_1x_3 + 2/3x_0x_2^3 + 2/3x_0x_2^2x_3 - 5/12x_0x_3 + 5/6x_0 - 5/9x_1x_3^2 + 10/9x_1x_3 + x_1 + 3/2x_3$ 
 $f_3 = -15/8x_0^3x_3^2 - 5/2x_0^2x_1x_3^3 - 5/8x_0^2x_3^3 + 5/4x_0^2x_3^2 - 5/6x_0x_1x_3^4 + 5/3x_0x_1x_3^3 + 3/2x_0x_1x_3^2 + 9/4x_0x_3^3 + 1/2x_1^2x_2 + 1/2x_1^2x_3 + 1/2x_2 + 1/2x_3$ 
 $f_4 = -5/8x_0^2x_2^2 - 5/6x_0x_1x_2^2x_3 - 5/24x_0x_2^2x_3 - 31/12x_0x_2^2 - 3x_0x_2x_3 - 5/18x_1x_2^2x_3^2 + 5/9x_1x_2^2x_3 + 1/2x_1x_2^2 + 3/4x_2^2x_3 + 1/2x_2^2 + 1/2x_2x_3 + x_3^2$ 
 $f_5 = 5x_0^2x_3^3 - 3/4x_0x_1x_2^2 - 3/4x_0x_1x_2x_3 + 20/3x_0x_1x_3^4 + 9/4x_0x_2^3x_3 + 9/4x_0x_2^2x_3^2 + 5/3x_0x_3^4 - 10/3x_0x_3^3 + x_0 + 20/9x_1x_3^5 - 40/9x_1x_3^4 - 4x_1x_3^3 + 1/4x_1x_3^2 - 3/4x_2x_3^3 + 2x_2x_3^2 - 6x_3^4 + 2x_3^3 + 1/3x_3 - 2/3$ | $g_1 = x_0 + 1/3x_3 - 2/3$ 
 $g_2 = x_1 + 3/2x_3$ 



 $g_3 = x_2 + x_3$ 



 $g_4 = x_3^2$ |
| 3 | $f_1 = x_0 - 1/3x_3$ 
 $f_2 = -1/3x_0x_1^2 + 1/2x_0x_3^3 + 1/9x_1^2x_3 - 1/6x_3^4 + x_3^2 + 3/5x_3$ 
 $f_3 = -x_0^2x_2x_3^2 - 3/5x_0^2x_2x_3 - x_0x_1^2x_2 + 5x_0 + 1/3x_1^2x_2x_3 + x_1 - 7/6x_3$ 
 $f_4 = 1/2x_0x_1x_2 + 4/5x_0x_3^3 + x_0x_3^3 + 3/5x_0x_3^2 + 4/5x_1x_2^2 - 1/6x_1x_2x_3 - 4/15x_2^3x_3 + 2/5x_2^2x_3 + x_2 + x_3$ 
 $f_5 = -5x_0^2x_2x_3 - 3/10x_0x_1^2x_2^2 - 2x_0x_1x_2x_3 - 3/2x_0x_2^2x_3 + 5/12x_0x_2x_3^2 - 3/4x_0x_2x_3 + 1/2x_0x_3^4 + 3/10x_0x_3^3 + 1/10x_1^2x_2^2x_3 - 3/5x_1x_2^2 + x_1x_2x_3^2 - 3/5x_1x_2x_3 + 1/2x_2^2x_3^2 + 1/6x_2x_3^3 - 5x_3^5 - 3x_3^4$ | $g_1 = x_0 - 1/3x_3$ 
 $g_2 = x_1 + 1/2x_3$ 

 $g_3 = x_2 + x_3$ 

 $g_4 = x_3^2 + 3/5x_3$ |

Table 10: Success examples from the $\mathcal{D}_5^-(\mathbb{Q})$ test set.

| ID | F | G |
|---|---|---|
| 0 | $f_1 = x_0 - x_4^4 - x_4^3 - 2x_4^2 + 1/4$ | $g_1 = x_0 - x_4^4 - x_4^3 - 2x_4^2 + 1/4$ |
| | $f_2 = 5/4x_0x_1^3 + 1/2x_0x_1^2 - 5/4x_1^3x_4^4 -$ | $g_2 = x_1 - 5x_4$ |
| | $5/4x_1^3x_4^3 - 5/2x_1^3x_4^2 + 5/16x_1^3 - 1/2x_1^2x_4^4 -$ | |
| | $1/2x_1^2x_4^3 - x_1^2x_4^2 + 1/8x_1^2 - 1/3x_1x_2x_3^3 +$ | |
| | $3x_1x_3x_4 + 5/3x_2x_3^2x_4 + x_2 - 15x_3x_4^2 -$ | |
| | $5x_4^4 + x_4^3 + 2/5x_4^2 + 2x_4 - 5/2$ | |
| | $f_3 = x_0^3x_4 - x_0^2x_4^5 - x_0^2x_4^4 - 2x_0^2x_4^3 +$ | $g_3 = x_2 - 5x_4^4 + x_4^3 + 2/5x_4^2 + 2x_4 - 5/2$ |
| | $1/4x_0^2x_4 + x_0x_1^2x_4 + 2x_0x_1x_4^2 +$ | |
| | $x_0x_3^3 - x_1^2x_4^5 - x_1^2x_4^4 - 2x_1^2x_4^3 +$ | |
| | $1/4x_1^2x_4 - 4/3x_1x_3^3x_2^2x_4 + 1/15x_1x_2^2x_3^3 +$ | |
| | $12x_1x_2^2x_3x_4^2 - 3/5x_1x_2x_3^2x_4 - 2x_1x_4^6 -$ | |
| | $2x_1x_4^5 - 4x_1x_4^4 + 1/2x_1x_4^2 + 20/3x_2^3x_3^2x_4^2 +$ | |
| | $4x_2^3x_4 - 1/3x_2^2x_3^3x_4 - 60x_2^2x_3x_4^3 -$ | |
| | $1/5x_2^2x_3 - 20x_2^2x_4^5 + 4x_2^2x_4^4 + 8/5x_2^2x_4^3 +$ | |
| | $8x_2^2x_4^2 - 10x_2^2x_4 + 3x_2x_3^2x_4^2 + x_2x_3x_4^4 -$ | |
| | $1/5x_2x_3x_4^3 - 2/25x_2x_3x_4^2 - 2/5x_2x_3x_4 +$ | |
| | $1/2x_2x_3 - x_3^3x_4^4 - x_3^3x_4^3 - 2x_3^3x_4^2 + 1/4x_3^3 +$ | |
| | $x_4^5 + 3/2x_4^4 - 1/2x_4^3 - 5/3x_4^2 - 1$ | |
| | $f_4 = 1/2x_0^2x_1x_2 + 1/2x_0x_1^2x_3^2x_4 +$ | $g_4 = x_3 + 5x_4^4 + 1/5x_4^2 + 1/2x_4 + 2/5$ |
| | $4/3x_0x_1^2x_3x_4^2 - 5/3x_0x_1^2x_3 -$ | |
| | $5/6x_0x_1x_2x_3^3 - 1/2x_0x_1x_2x_4^4 -$ | |
| | $1/2x_0x_1x_2x_4^3 - x_0x_1x_2x_4^2 + 1/8x_0x_1x_2 +$ | |
| | $15/2x_0x_1x_3^3x_4 + 25/6x_0x_2x_3^4x_4 +$ | |
| | $5/2x_0x_2x_3^2 + 1/2x_0x_3^5 + 4/3x_0x_3^4x_4 -$ | |
| | $75/2x_0x_3^3x_4^2 - 25/2x_0x_3^2x_4^4 + 5/2x_0x_3^2x_4^3 +$ | |
| | $x_0x_3^2x_4^2 + 5x_0x_3^2x_4 - 25/4x_0x_3^2 -$ | |
| | $1/2x_1^2x_3^2x_4^5 - 1/2x_1^2x_3^2x_4^4 - x_1^2x_3^2x_4^3 +$ | |
| | $1/8x_1^2x_3^2x_4 - 4/3x_1^2x_3x_4^6 - 4/3x_1^2x_3x_4^5 -$ | |
| | $x_1^2x_3x_4^4 + 5/3x_1^2x_3x_4^3 + 11/3x_1^2x_3x_4^2 -$ | |
| | $5/12x_1^2x_3 - 1/2x_3^5x_4^4 - 1/2x_3^5x_4^3 - x_3^5x_4^2 +$ | |
| | $1/8x_3^5 - 4/3x_3^4x_4^5 - 4/3x_3^4x_4^4 - 8/3x_3^4x_4^3 +$ | |
| | $1/3x_3^4x_4 + 1/2x_3^2x_4^5 + 3/4x_3^2x_4^4 - 1/4x_3^2x_4^3 -$ | |
| | $5/6x_3^2x_4^2 - 1/2x_3^2 + 4/3x_3x_4^6 + 2x_3x_4^5 -$ | |
| | $2/3x_3x_4^4 - 20/9x_3x_4^3 - 4/3x_3x_4 + x_3 +$ | |
| | $5x_4^4 + 1/5x_4^2 + 1/2x_4 + 2/5$ | |
| | $f_5 = -x_0^2x_3^2 + 3/2x_0x_1^2x_2x_3^2x_4 +$ | $g_5 = x_4^5 + 3/2x_4^4 - 1/2x_4^3 - 5/3x_4^2 - 1$ |
| | $3/2x_0x_2x_3^5 + x_0x_3^2x_4^4 + x_0x_3^2x_4^3 + 2x_0x_3^2x_4^2 -$ | |
| | $1/4x_0x_3^2 - 3/2x_1^2x_2x_3^2x_4^5 - 3/2x_1^2x_2x_3^2x_4^4 -$ | |
| | $3x_1^2x_2x_3^2x_4^3 + 3/8x_1^2x_2x_3^2x_4 + 1/3x_1x_2^2x_3^3 -$ | |
| | $3x_1x_2x_3^3x_4 + x_1 - 5/3x_2^2x_3^4x_4 - x_2^2x_3^3 -$ | |
| | $3/2x_2x_3^5x_4^4 - 3/2x_2x_3^5x_4^3 - 3x_2x_3^5x_4^2 +$ | |
| | $3/8x_2x_3^5 + 15x_2x_3^3x_4^2 + 3/2x_2x_3^2x_4^5 +$ | |
| | $29/4x_2x_3^2x_4^4 - 7/4x_2x_3^2x_4^3 - 29/10x_2x_3^2x_4^2 -$ | |
| | $2x_2x_3^2x_4 + x_2x_3^2 - 3x_3^2x_4^2 - 15x_3x_4^6 -$ | |
| | $3/5x_3x_4^4 - 3/2x_3x_4^3 - 6/5x_3x_4^2 - 5x_4$ | |

Table 11: Success examples from the $\mathcal{D}_2^-(\mathbb{F}_7)$ test set.

| ID | $F$ | $G$ |
|---|---|---|
| 0 | $f_1 = x_0 + 3x_1$
$f_2 = x_0^3 + x_0^2 x_1 + 3x_0 x_1^2 + x_1^4 - x_1^3 - 3$ | $g_1 = x_0 + 3x_1$
$g_2 = x_1^4 - 3$ |
| 1 | $f_1 = -3x_0 x_1^3 + 2x_1^3$
$f_2 = -x_0^2 x_1 + 2x_0 x_1^4 + x_0 x_1^3 + 2x_0 x_1 + x_1^4 - 3x_1^3 + x_1$
$f_3 = x_0^3 x_1^3 - 2x_0^2 x_1^3 + x_0 x_1^4 - x_0 x_1^3 + x_0 - 3x_1^4 - 2$ | $g_1 = x_0 - 2$
$g_2 = x_1$ |
| 3 | $f_1 = 2x_0^3 - x_0^2 x_1^4 + 3x_0^2 x_1^2 + x_0^2 - 2x_1^6 + 3x_1^3 + 2x_1^2 + 3x_1$
$f_2 = -3x_0^4 x_1^2 + 3x_0^4 x_1 - 2x_0^3 x_1^6 + 2x_0^3 x_1^5 - x_0^3 x_1^4 + x_0^3 x_1^3 + 2x_0^3 x_1^2 - 2x_0^3 x_1 + 3x_0 x_1^8 - 3x_0 x_1^7 - x_0 x_1^5 - 2x_0 x_1^4 + 2x_0 x_1^3 + x_0 x_1^2 + x_0 + 3x_1^4 - 2x_1^2 - 3$
$f_3 = -2x_0^5 + x_0^4 x_1^4 - 3x_0^4 x_1^2 - x_0^4 + 2x_0^2 x_1^6 - 3x_0^2 x_1^3 - 2x_0^2 x_1^2 - 3x_0^2 x_1 + x_0^2 + 3x_0 x_1^4 - 3x_0 - x_1^6 + x_1^5 + 3x_1^4 + 3x_1^2 - x_1 + 2$ | $g_1 = x_0 + 3x_1^4 - 2x_1^2 - 3$

$g_2 = x_1^5 + 2x_1^2 - x_1 + 2$ |
| 4 | $f_1 = -2x_0^2 x_1^2 - 2x_0 x_1^3 - 2x_0 x_1^2 + 2x_0 x_1 + x_1^4 + 2x_1^2 + 2x_1$
$f_2 = x_0^3 x_1^2 + x_0^2 x_1^3 + x_0^2 x_1^2 - x_0^2 x_1 + x_0^2 + 3x_0 x_1^4 - 2x_0 x_1^2 + x_0 x_1 + x_0 + x_1^2 + x_1$
$f_3 = -x_0^4 x_1^2 - x_0^3 x_1^3 + x_0^3 x_1 - 3x_0^2 x_1^4 + 2x_0^2 x_1^3 + 2x_0^2 x_1^2 + x_0^2 x_1 + x_0 x_1^5 + 2x_0 x_1^4 + x_0 + 3x_1^6 - x_1^4 - x_1^3 + x_1 + 1$
$f_4 = -3x_0^4 x_1 + 2x_0^3 x_1^3 + x_0^3 x_1^2 - 3x_0^3 x_1 + x_0^3 - x_0^2 x_1^4 + 3x_0^2 x_1^3 - 2x_0^2 x_1^2 + x_0^2 x_1 + x_0^2 - 3x_0 x_1^5 + x_0 x_1^3 + 3x_0 x_1^2 - 2x_0 + 2x_1^3 + 3x_1^2 - 2x_1 - 2$ | $g_1 = x_0 + x_1 + 1$

$g_2 = x_1^2$ |
| 5 | $f_1 = -x_0^2 x_1 + 3x_0 x_1 + x_1^3$
$f_2 = 3x_0^2 x_1^4 - 3x_0^2 x_1^3 - 2x_0 x_1^4 + 2x_0 x_1^3 + x_0 - 3x_1^6 + 3x_1^5 - 3$ | $g_1 = x_0 - 3$
$g_2 = x_1^3$ |
| 7 | $f_1 = 3x_0^2 x_1 - x_0 x_1^4 - x_0 x_1^2 - x_1^7 - 3x_1^5 + 2x_1^3 - x_1$
$f_2 = 2x_0^2 x_1^3 - 3x_0 x_1^6 - 3x_0 x_1^4 + x_0 - 3x_1^9 - 2x_1^7 - x_1^5 - x_1^3 + 2x_1$
$f_3 = -2x_0^3 x_1^2 + x_0^3 x_1 + 3x_0^2 x_1^5 + 2x_0^2 x_1^4 + 3x_0^2 x_1^3 + 2x_0^2 x_1^2 + 3x_0 x_1^8 + 2x_0 x_1^6 + x_0 x_1^4 - 2x_0 x_1^2 - x_0 x_1 - 3x_1^5 - x_1^4 - 3x_1^3 - 2x_1^2 - 2$
$f_4 = x_0^3 x_1^2 + 2x_0^3 x_1 + x_0^3 + 2x_0^2 x_1^5 + x_0^2 x_1^4 - 3x_0^2 x_1^3 - 3x_0^2 x_1^2 + 2x_0^2 - 2x_0 x_1^7 + x_0 x_1^5 - 2x_0 x_1^4 + 3x_0 x_1^3 + 2x_0 x_1 + 3x_0 + x_1^5 - 2x_1^4 + x_1^3 + 3x_1^2 + 3$ | $g_1 = x_0 + 2x_1^3 + 2x_1$

$g_2 = x_1^4 - 2$ |
| 10 | $f_1 = x_0 - 1$
$f_2 = -3x_0^2 x_1^2 + 3x_0^2 x_1 + 2x_0^2 - 2x_0 x_1^3 + 3x_0 x_1^2 - 3x_0 x_1 - 2x_0$
$f_3 = -x_0^4 x_1^3 - 3x_0^3 x_1^4 + x_0^3 x_1^3 + x_0^3 x_1^2 + x_0^3 + 3x_0^2 x_1^3 - x_0^2 x_1^2 - x_0^2 + 3x_0 x_1^2 - 3x_1^2 + x_1$ | $g_1 = x_0 - 1$
$g_2 = x_1$ |
| 11 | $f_1 = x_0 + x_1^4 + 2x_1^2$
$f_2 = 3x_0^2 x_1 + 3x_0 x_1^5 - x_0 x_1^3 - 2x_0 x_1^2 - 2x_1^6 + x_1^5 + 3x_1^4 - x_1^3 - x_1^2 - x_1$
$f_3 = -x_0^3 x_1^2 + 2x_0^2 x_1^6 - 3x_0^2 x_1^5 + 2x_0^2 x_1^4 + 3x_0^2 x_1 - 2x_0^2 - 2x_0 x_1^4 + 3x_0 x_1^2 - 2x_0 x_1 + 3x_0 + x_1^6 - 2x_1^5 + 2x_1^4 + 2x_1^3 - 2x_1^2$
$f_4 = 2x_0^3 x_1^8 - 2x_0^3 x_1^6 - 2x_0^3 x_1^5 - 2x_0^3 x_1^4 - 3x_0^3 x_1^2 - 3x_0^2 x_1^6 + x_0^2 x_1^3 + 3x_0^2 x_1^2 + 2x_0^2 + 2x_0 x_1^8 - 2x_0 x_1^7 + x_0 x_1^6 + 2x_0 x_1^5 - 2x_0 x_1^4 + x_0 x_1^2$ | $g_1 = x_0 + x_1^4 + 2x_1^2$
$g_2 = x_1^5 - x_1^3 - x_1^2 - x_1$ |

Table 12: Success examples from the $\mathcal{D}_3^-\left(\mathbb{F}_7\right)$ test set.

| ID | $F$ | $G$ |
|---|---|---|
| 0 | $f_1 = -2x_0^2x_1^2 - x_0^2x_1x_2^2 - x_0^2x_1 - 2x_0x_1^2 + x_1^2x_2^2$ | $g_1 = x_0 + 3x_2^2$ |
| | $f_2 = -2x_0^3x_1 + 3x_0^2x_1^4 - 2x_0^2x_1^3x_2^2 - 2x_0^2x_1^3 + x_0^2x_1x_2^2 + 3x_0x_1^4 - 3x_0x_1 + 2x_1^4x_2^2 - 2x_1x_2^2 + x_1 - 3x_2^2 - 3$ | $g_2 = x_1 - 3x_2^2 - 3$ |
| | $f_3 = 2x_0^3x_1^2x_2^2 + x_0^3x_1x_2^4 - 3x_0^3x_1x_2^3 + x_0^3x_1x_2^2 + 2x_0^2x_1^2x_2^2 - 2x_0^2x_1x_2^5 - x_0x_1^2x_2^4 - x_0x_1x_2^3 + x_0 - 3x_1x_2^5 - 2x_1x_2^3 - x_2^5 - x_2^3 + 3x_2^2$ | $g_3 = x_2^5 - 3x_2^4 - 2x_2^3 - 3x_2^2 - 1$ |
| | $f_4 = 3x_0^4x_1^2 + 3x_0^4x_1x_2^2 + 3x_0^4x_1 - 2x_0^3x_1^2x_2^2 + 3x_0^3x_1^2 + 3x_0^3x_1 + 2x_0^2x_1^5 + x_0^2x_1^4x_2^2 + x_0^2x_1^4 + 2x_0^2x_1^2x_2^2 - x_0^2x_1^2 + 2x_0^2x_1x_2^2 + 2x_0^2x_1 + 2x_0x_1^5 - 3x_0x_1^2x_2^2 - 3x_0x_1^2 - x_1^5x_2^2 - 2x_1^3x_2 - x_1^2x_2^3 - 3x_1^2x_2^2 - x_1^2x_2 - 2x_1^2 - x_1x_2^4 + 2x_1x_2^2 - x_1 + x_2^5 - 3x_2^4 - 2x_2^3 - 3x_2^2 - 1$ | |
| 1 | $f_1 = x_0 + 3x_2$ | $g_1 = x_0 + 3x_2$ |
| | $f_2 = 3x_0^2x_2^2 + 2x_0x_2^3 + x_1 + x_2 + 3$ | $g_2 = x_1 + x_2 + 3$ |
| | $f_3 = -3x_0^3x_1^3 - x_0^3x_1 - 2x_0^2x_1^3x_2 + 2x_0^2x_1^2 + x_0^2x_1x_2 - 2x_0^2x_1 - 2x_0^2x_2 + x_0^2 - x_0x_1^5x_2 - x_0x_1^2x_2 - 2x_0x_1x_2^2 + x_0x_1x_2 + x_0x_2^2 + 3x_0x_2 - 3x_1^5x_2^2 + 3x_1^3x_2 + 3x_1^2x_2^2 + 2x_1^2x_2 + x_2^2$ | $g_3 = x_2^2$ |
| | $f_4 = -3x_0^4x_1^3 + 3x_0^3x_1^3x_2 + 2x_0^3x_1^3 + x_0^2x_1^3x_2^2 + 2x_0^2x_1^3x_2 + 3x_0^2x_1^2x_2^2 + 2x_0^2x_1^2x_2 + 3x_0^2x_1x_2^2 - x_0^2x_1 - x_0^2x_2 - 3x_0^2 - 3x_0x_1^5x_2 - 3x_0x_1^4x_2^2 + 3x_0x_1^2x_2^2 + 3x_0x_1^2x_2 + 2x_0x_1^2 - x_0x_1x_2^2 + 2x_0x_1x_2 - x_0x_1 + x_0x_2^4 - x_1^4x_2^2 + 2x_1^2x_2^2 - x_1^2x_2 - 3x_1x_2^3 - x_1x_2^2 - 3x_1x_2$ | |
| 2 | $f_1 = 2x_0x_2 + 2x_1^2x_2^2 + 2x_1x_2^2 - 2x_1 - 3x_2^2 + 3$ | $g_1 = x_0 - 2$ |
| | $f_2 = -3x_0^2x_2 - 3x_0x_1^2x_2^2 + 3x_0x_1x_2 + 3x_0x_1 + x_0x_2^2 - x_0 + 3x_1^3x_2^3 + 3x_1^2x_2^3 - 3x_1^2x_2 - x_1x_2^3 + 2x_1x_2 + x_1 + 2$ | $g_2 = x_1 + 2$ |
| | $f_3 = x_0^3x_1 - 2x_0^2x_1 - x_0x_1x_2^3 + 2x_0x_2^3 - x_0x_2 + 2x_1^2x_2^4 - x_1^2x_2^2 + 2x_1x_2^4 + 2x_1x_2^3 - x_1x_2^2 + x_1 - 3x_2^4 - 2x_2^2 + x_2 + 2$ | $g_3 = x_2$ |
| | $f_4 = -2x_0^4x_1 + 2x_0^3x_1^3 - 3x_0^3x_1 + 3x_0^2x_1^3 + 2x_0x_1x_2^2 + 2x_0x_2^4 + 2x_0x_2^3 - 2x_0x_2 + 2x_1^2x_2^5 + 2x_1^2x_2 + 2x_1x_2^5 - 2x_1x_2^3 + 3x_1x_2^2 + 3x_1x_2 - 3x_2^5 + 3x_2^3 - x_2$ | |
| | $f_5 = 2x_0^5x_1 + 3x_0^4x_1 + x_0^3x_1x_2^2 - x_0^2x_1^2x_2 + 2x_0^2x_2 - x_0x_1^2x_2^2 + 2x_0x_1^2x_2 + 2x_0x_1^2 + 2x_0x_1x_2 - 3x_0x_1 + 3x_0x_2^3 + x_0 + 2x_1^3x_2^2 - 3x_1^2x_2^2 + 2x_1^2x_2 - 2x_1^2 - 3x_1x_2^2 - 3x_1x_2 + 3x_1 - 2$ | |

Table 13: Success examples from the $\mathcal{D}_4^- (\mathbb{F}_7)$ test set.

| ID | $F$ | $G$ |
|---|---|---|
| 2 | $f_1 = x_2 - 3x_3^3 + 2x_3^2 + 1$ | $g_1 = x_0 - x_3$ |
| | $f_2 = x_1 x_2 - 3x_1 x_3^3 + 2x_1 x_3^2 + 2x_1 - 3x_3^3 + x_3^2 - 2x_3 + 2$ | $g_2 = x_1 - 3x_3^3 + x_3^2 - 2x_3 + 2$ |
| | $f_3 = -2x_0^2 x_1^2 - x_0^2 x_1 x_3^3 - 2x_0^2 x_1 x_3^2 - 3x_0^2 x_1 x_3 + 3x_0^2 x_1 - x_0 x_1 x_2^2 - x_0 x_1 x_2 x_3 + 3x_0 x_2^2 x_3^3 - x_0 x_2^2 x_3^2 + 2x_0 x_2^2 x_3 - 2x_0 x_2^2 + 3x_0 x_2 x_3^4 - x_0 x_2 x_3^3 + 2x_0 x_2 x_3^2 - 2x_0 x_2 x_3 - x_1^2 x_2 + 3x_1^2 x_3^3 - 2x_1^2 x_3^2 - x_1^2$ | $g_3 = x_2 - 3x_3^3 + 2x_3^2 + 1$ |
| | $f_4 = 2x_0^3 x_1^3 x_3 + x_0^3 x_1^2 x_3^4 + 2x_0^3 x_1^2 x_3^3 + 3x_0^3 x_1^2 x_3^2 - 3x_0^3 x_1^2 x_3 + x_0^2 x_1^2 x_2 x_3^2 - 3x_0^2 x_1^2 x_3^3 - 3x_0^2 x_1 x_2 x_3^5 + x_0^2 x_1 x_2 x_3^4 - 2x_0^2 x_1 x_2 x_3^3 + 2x_0^2 x_1 x_2 x_3^2 + 3x_0^2 x_1 x_2 + 2x_0^2 x_1 x_3^6 - 3x_0^2 x_1 x_3^5 - x_0^2 x_1 x_3^4 - x_0^2 x_1 x_3^3 - x_0^2 x_1 x_3^2 + 3x_0^2 x_1 + 3x_0 x_1^2 x_2 - 2x_0 x_1^2 x_3^3 - x_0 x_1^2 x_3^2 + x_0 x_1^2 - x_0 x_1 x_2^2 + 2x_0 x_1 x_2 x_3^4 - x_0 x_1 x_3^3 - 2x_0 x_1 x_3^2 - 3x_0 x_1 x_3 + 3x_0 x_1 + 3x_0 x_2^2 x_3^3 - x_0 x_2^2 x_3^2 + 2x_0 x_2^2 x_3 - 2x_0 x_2^2 + x_0 x_2 x_3^7 + 2x_0 x_2 x_3^6 + 3x_0 x_2 x_3^5 - 3x_0 x_2 x_3^4 + 3x_1^3 x_2 - 2x_1^2 x_2 x_3^3 + 3x_1^2 x_2 x_3^2 + x_1^2 x_2 x_3 - x_1^2 x_2 + x_3^4 - 3x_3 - 1$ | $g_4 = x_3^4 - 3x_3 - 1$ |
| | $f_5 = -x_0^3 x_1^3 x_3 + 3x_0^3 x_1^2 x_3^4 - x_0^3 x_1^2 x_3^3 + 2x_0^3 x_1^2 x_3^2 - 2x_0^3 x_1^2 x_3 + 3x_0^2 x_1^2 x_2 x_3^2 + 3x_0^2 x_1^2 x_3 - 3x_0^2 x_1^2 - 2x_0^2 x_1 x_2 x_3^5 + 3x_0^2 x_1 x_2 x_3^4 + x_0^2 x_1 x_2 x_3^3 - x_0^2 x_1 x_2 x_3^2 - 2x_0^2 x_1 x_3^4 - 2x_0^2 x_1 x_3^3 - 2x_0^2 x_1 x_3^2 - 2x_0^2 x_1 x_3 + x_0^2 x_1 + 2x_0 x_1^3 - 2x_0 x_1 x_2 x_3^2 - x_0 x_2 x_3^5 - 2x_0 x_2 x_3^4 - 3x_0 x_2 x_3^3 + 3x_0 x_2 x_3^2 - 3x_0 x_2 - 2x_0 x_3^4 + 2x_0 x_3^3 + x_0 x_3^2 - x_0 x_3 - x_0 - 2x_1^3 x_3 + x_1 x_2^2 x_3 - 2x_2^3 - 3x_2^2 x_3^4 + x_2^2 x_3^2 + 2x_2^2 x_3 - 2x_2^2$ | |
| | $f_6 = -3x_0^2 x_1^3 x_2 - 2x_0^2 x_1^3 + 2x_0^2 x_1^2 x_2 x_3^3 - 3x_0^2 x_1^2 x_2 x_3^2 - x_0^2 x_1^2 x_2 x_3 + x_0^2 x_1^2 x_2 - x_0^2 x_1^2 x_3^3 - 2x_0^2 x_1^2 x_3^2 - 3x_0^2 x_1^2 x_3 + 2x_0^2 x_1^2 + 3x_0^2 x_1 x_3^3 - x_0^2 x_1 x_3^2 + 2x_0^2 x_1 x_3 - 2x_0^2 x_1 + 3x_0^2 x_2 - 2x_0^2 x_3^3 - x_0^2 x_3^2 + 3x_0^2 + x_0 x_1^5 x_3 + 2x_0 x_1^2 x_2^2 x_3 - x_0 x_1^2 x_2 x_3 + x_0 x_1 x_2^2 x_3^4 + 2x_0 x_1 x_2^2 x_3^3 + 3x_0 x_1 x_2^2 x_3^2 - 3x_0 x_1 x_2^2 x_3 + 3x_0 x_1 x_2 x_3^4 - x_0 x_1 x_2 x_3^3 + 2x_0 x_1 x_2 x_3^2 - 2x_0 x_1 x_2 x_3 - 3x_0 x_3^4 + 2x_0 x_3 - 3x_0 - x_1^5 x_3^2 - x_1^2 x_2^2 + 3x_1 x_2^2 x_3^3 - x_1 x_2^2 x_3^2 + 2x_1 x_2^2 x_3 - 2x_1 x_2^2 - x_1 x_3^3 + 3x_3^6 - x_3^5 + 2x_3^4 - 2x_3^3 - x_3$ | |
| 3 | $f_1 = 3x_0 x_1 x_3 - x_1 x_3^2 + x_1 x_3 + x_3^2$ | $g_1 = x_0 + 2x_3 - 2$ |
| | $f_2 = -2x_0^3 x_1 x_3 + 3x_0^2 x_1 x_3^2 - 3x_0^2 x_1 x_3 + 2x_0^2 x_1 - 3x_0^2 x_3^2 + 3x_0^2 x_3 - 3x_0 x_1^2 x_2 x_3 + x_1^2 x_2 x_3^2 - x_1^2 x_2 x_3 + 2x_1^2 x_3^2 - x_1 x_2 x_3^2 + 3x_1 x_3^3 + x_2 - 2x_3 - 3$ | $g_2 = x_1 - 2x_3$ |
| | $f_3 = -2x_0^2 x_1^4 - 3x_0^2 x_1^3 x_3 + 3x_0^2 x_1 x_2 x_3 + x_0 x_1 x_2^2 x_3 - x_0 x_1 x_2 x_3^2 + x_0 x_1 x_2 x_3 + x_0 x_2 x_3^2 + x_0 - 2x_1^5 x_3^2 - 3x_1^4 x_3^3 - x_1^3 x_2 + 2x_1^3 x_3 + 3x_1^3 + 2x_1 x_2^2 x_3^3 - 2x_1 x_2^2 x_3 - 2x_2^2 x_3^2 + 2x_3 - 2$ | $g_3 = x_2 - 2x_3 - 3$ |
| | $f_4 = -x_0^2 x_1^2 x_2 x_3 + 2x_0^2 x_1 x_2 x_3^2 + x_0^2 x_1 x_2 - 3x_0^2 x_1 x_3^3 + x_0^2 x_1 - x_0^2 x_3^4 + 2x_0 x_1^3 x_2 x_3 + 2x_0 x_1 x_2 x_3 - 2x_0 x_1 x_2 + 2x_0 x_1 x_3 - x_0 x_1 - x_1^3 x_2 x_3^3 - 3x_1^3 x_2 x_3^2 + 3x_1^3 x_2 x_3 + 2x_1^2 x_2 x_3^4 + 3x_1^2 x_2 x_3^2 - 3x_1^2 x_3^5 + 3x_1 x_2^2 x_3 + x_1 x_2 x_3^2 - 2x_1 x_2 x_3 - x_1 x_3^6 + 2x_1 x_3 - x_1 + 2x_2 x_3^3 + 3x_3^4 + x_3^3 - 2x_3$ | $g_4 = x_3^2$ |

Table 14: Success examples from the $\mathcal{D}_5^-\,(\mathbb{F}_7)$ test set.

| ID | $F$ | $G$ |
|---|---|---|
| 0 | $f_1 = -x_0x_1^2 + 2x_1^2x_4^2 + x_1^2x_4 + 3x_1^2 + x_1 - 3x_4$ | $g_1 = x_0 - 2x_4^2 - x_4 - 3$ |
| | $f_2 = -x_0x_1^2x_3^3 + 2x_1^2x_3^3x_4^2 + x_1^2x_3^3x_4 + 3x_1^2x_3^3 + 3x_1x_2^2x_3 + x_1x_3^3 - 2x_2^2x_3x_4 - 3x_3^3x_4$ | $g_2 = x_1 - 3x_4$ |
| | $f_3 = -3x_0^2x_1^2x_3 - 3x_0^2x_1x_2^2x_3^2 + 2x_0^2x_2^2x_3^2x_4 - x_0x_1^2x_3x_4^2 + 3x_0x_1^2x_3x_4 + 2x_0x_1^2x_3 + 3x_0x_1x_3 - 2x_0x_3x_4 + x_0 + x_1x_2^3x_3x_4^2 - 3x_2^3x_3x_4^3 - 2x_4^2 - x_4 - 3$ | $g_3 = x_2 - x_4$ |
| | $f_4 = -2x_0^2x_1x_2^2x_3 - x_0^2x_2^2x_3x_4 + x_0^2x_3 + 3x_0^2x_4 - 3x_0x_1^3x_2x_4 - 3x_0x_1^2x_2^3x_3 + 2x_0x_1x_2^3x_3x_4 + 3x_0x_2^3x_4 - x_1^3x_2x_4^3 + 3x_1^3x_2x_4^2 + 2x_1^3x_2x_4 + 3x_1^2x_2x_4 - 2x_1x_2x_4^2 + x_2^3x_4^3 - 3x_2^3x_4^2 - 2x_2^3x_4 + x_4^3 - x_4^2 + 3x_4$ | $g_4 = x_3 + 3x_4$ |
| | $f_5 = 2x_0^3x_1x_2x_3 - x_0^3x_1x_2x_4 + x_0^2x_3^3 + 3x_0^2x_2^3x_4 + 3x_0x_1^2x_3x_4 + 2x_0x_1x_2x_3 + 2x_0x_1x_2x_4^3 - 2x_0x_1x_2x_4^2 - x_0x_1x_2x_4 + x_1^2x_2^3x_3^2 - 3x_1^2x_2^2x_3x_4 + x_1^2x_3x_4^3 - 3x_1^2x_3x_4^2 - 2x_1^2x_3x_4 - 3x_1x_2^3x_3^2x_4 + 2x_1x_2^2x_3x_4^2 + 3x_1x_2x_3x_4^2 - 2x_1x_2x_3x_4 + x_1x_2x_3 - 3x_1x_3x_4 + x_3^2x_4^3 - x_3^2x_4^2 + 3x_3^2x_4 + 2x_3x_4^2 + x_3 + 3x_4$ | $g_5 = x_4^3 - x_4^2 + 3x_4$ |
| | $f_6 = -x_0^2x_1^2x_3 - 2x_0^2x_1^2x_4^2 - 3x_0^2x_1^2x_4 - 3x_0x_1^2x_2 + 3x_0x_1^2x_3^2 - 3x_0x_1^2x_4^4 + 2x_0x_1^2x_4^3 - x_0x_1^2x_4^2 - x_0x_1x_2x_4 + 2x_0x_1x_4^2 + 3x_0x_2x_4^2 - x_0x_4^3 + x_1^2x_2^3x_3 + x_1^2x_3^2x_4^2 - 3x_1^2x_2^3x_4 - 2x_1^2x_3^2 - x_1^2x_3 - x_1^2x_4^3 + x_1^2x_4^2 + x_1^2x_4 - 3x_1x_2^3x_3x_4 - 3x_1x_3^2 - 2x_2x_4^4 - x_2x_4^3 - 3x_2x_4^2 + x_2 + 2x_3^2x_4 - 3x_4^5 + 2x_4^4 - x_4^3 - x_4$ | |
| | $f_7 = -2x_0^4x_2x_3 + x_0^4x_2x_4 + 2x_0^3x_1^2x_3 - x_0^3x_1x_3x_4 - 3x_0^3x_1x_4^2 + 3x_0^2x_1^2x_3x_4^2 - 2x_0^2x_1^2x_3x_4 + x_0^2x_1^2x_3 - 2x_0^2x_1x_3 - 2x_0^2x_2x_4^3 + 2x_0^2x_2x_4^2 + x_0^2x_2x_4 - x_0^2x_3x_4 - 2x_0x_1^3x_2x_3^2 - 3x_0x_1^2x_2x_3^2x_4 - x_0x_1x_2x_3^2x_4^2 + x_0x_1x_2x_3 + 3x_0x_1x_2x_4 - 2x_0x_1x_3^2 - x_0x_1x_4^4 + x_0x_1x_3^3 - 3x_0x_1x_4^2 - x_0x_1x_4 - 2x_0x_3x_4^3 + 2x_0x_3x_4^2 + x_0x_3x_4 - 3x_1^3x_2^2x_3x_4 + 2x_1^2x_2^2x_3x_4^2 + 3x_1x_2x_3^2 - 3x_1x_3^2x_4^2 - x_1x_3^2x_4 - x_1x_3^2 + 2x_1x_4^3 + x_1x_4^2 + 3x_1x_4$ | |
| 3 | $f_1 = x_4^2$ | $g_1 = x_0 - 2x_4$ |
| | $f_2 = x_0x_2x_4^3 + x_0 - 2x_4$ | $g_2 = x_1 + x_4 - 3$ |
| | $f_3 = -3x_0^2x_3^2 + 3x_0^2 - x_0x_1x_2x_4 - x_0x_3^2x_4 + x_0x_4 + 2x_1x_2x_4^2 + x_3 - x_4^5 - x_4 + 1$ | $g_3 = x_2 + x_4$ |
| | $f_4 = -2x_0^3x_2^2x_3^2 + 2x_0^3x_2^2 - 3x_0^2x_1^2x_3^3 + 3x_0^2x_1^2x_3 - 3x_0^2x_2^2x_3^2x_4 + 3x_0^2x_2^2x_4 - x_0x_1^2x_3^3x_4 + x_0x_1^2x_3x_4 + 3x_0x_2^2x_3 - 3x_0x_2^2x_4 + 3x_0x_2^2 + 3x_0x_2x_4^2 + x_1^2x_3^2 - x_1^2x_3x_4 + x_1^2x_3 - 3x_1x_2^2x_4^2 + x_1 + x_2x_4^3 + x_4 - 3$ | $g_4 = x_3 - x_4 + 1$ |
| | $f_5 = -2x_0^3x_2x_3^2 + 2x_0^3x_2 - x_0^3 - 3x_0^2x_2x_3^2x_4 + 3x_0^2x_2x_4 + 2x_0^2x_4^2 + 2x_0^2x_4 + 3x_0x_1x_3^2 + 3x_0x_2x_3 - 3x_0x_2x_4 + 3x_0x_2 + 3x_0x_3^2x_4 - 2x_0x_3^2 + 3x_0x_4^3 - 3x_1x_4^2 - 2x_2x_4^3 + x_2 - 3x_4^3 + 2x_4^2 + x_4$ | $g_5 = x_4^2$ |

Table 15: Success examples from the $\mathcal{D}_2^-\left(\mathbb{F}_{31}\right)$ test set.

| ID | $F$ | $G$ |
|---|---|---|
| 2 | $f_1 = x_0 + 8x_1$ 
 $f_2 = -8x_0^3x_1 - 2x_0^2x_1^2 - 12x_0x_1^5 - x_0x_1^3 + 12x_0x_1^2 - x_1^4 - x_1^3 + x_1$ 
 $f_3 = -7x_0^4x_1^3 + 6x_0^3x_1^4 + 6x_0^3x_1^2 - 15x_0^3x_1 + 5x_0^3 + 5x_0^2x_1^7 - 9x_0^2x_1^5 - 5x_0^2x_1^4 - 14x_0^2x_1^3 + 4x_0^2x_1^2 + 9x_0^2x_1 - 10x_0^2 + 9x_0x_1^6 + 3x_0x_1^5 - 10x_0x_1^4 - 12x_0x_1^3 + 13x_0x_1 - 7x_1^4 + x_1^3 + 7x_1^2 - 8x_1$ | $g_1 = x_0 + 8x_1$ 
 $g_2 = x_1^3 - 8x_1$ |
| 4 | $f_1 = 6x_0^3x_1 + 11x_0^2x_1^4 + 6x_0^2x_1^2 - 2x_0x_1^3 - 11x_1^6 - 2x_1^4$ 
 $f_2 = 3x_0^3x_1^4 - 10x_0^2x_1^7 + 3x_0^2x_1^5 - x_0x_1^6 + 3x_0x_1^5 - 11x_0x_1^3 + 15x_0x_1^2 + 10x_1^9 - x_1^7 - 14x_1^6 + 5x_1^5 - 11x_1^4 + 15x_1^3$ 
 $f_3 = 13x_0^4x_1 - 2x_0^3x_1^4 + 13x_0^3x_1^2 - 9x_0^2x_1^5 + 8x_0^2x_1^3 - 14x_0^2x_1^2 - 11x_0^2x_1 - 6x_0x_1^6 - 15x_0x_1^5 + 12x_0x_1^4 + 15x_0x_1^3 - 11x_0x_1^2 + 6x_1^7 - 11x_1^6 - 13x_1^5 - x_1^4$ 
 $f_4 = -7x_0^4x_1^2 - 10x_0^4x_1 + 13x_0^3x_1^5 + 6x_0^3x_1^4 - 7x_0^3x_1^3 - 10x_0^3x_1^2 - 3x_0^3x_1 - 2x_0^2x_1^7 - 3x_0^2x_1^5 - 5x_0^2x_1^4 + 14x_0^2x_1^3 - 3x_0^2x_1^2 - x_0x_1^8 - 6x_0x_1^7 + 12x_0x_1^6 + 13x_0x_1^5 + 3x_0x_1^4 + x_0 - 10x_1^3 + x_1$ | $g_1 = x_0 - 10x_1^3 + x_1$ 

 $g_2 = x_1^4$ |
| 6 | $f_1 = x_0 - 15x_1$ 
 $f_2 = 4x_0^3 + 2x_0^2x_1 - 10x_0x_1^2 - 5x_1^3 + x_1^2$ 
 $f_3 = 3x_0^3x_1^3 - 14x_0^2x_1^4 + 9x_0^2x_1^3 + 3x_0^2x_1^2 - 12x_0^2x_1 - 7x_0x_1^3 - 6x_0x_1^2 + 4x_0 - 12x_1^4 - 6x_1^3 + 2x_1$ | $g_1 = x_0 - 15x_1$ 
 $g_2 = x_1^2$ |
| 7 | $f_1 = -6x_0^3 - 10x_0^2 + 4x_0x_1^3 + 5x_1^2$ 
 $f_2 = -x_0^3x_1^3 - 12x_0^2x_1^3 + 11x_0x_1^6 + x_0 + 6x_1^5 + 12$ 
 $f_3 = -9x_0^5 - 15x_0^4 + 6x_0^3x_1^3 + 13x_0^2x_1^2 - 11x_0^2x_1 + 4x_0x_1^2 + 3x_0x_1 + 9x_1$ | $g_1 = x_0 + 12$ 
 $g_2 = x_1$ |
| 14 | $f_1 = 3x_0^3 - x_0^2x_1 - 12x_0^2 + x_1^2$ 
 $f_2 = 14x_0^3x_1 - 15x_0^2x_1^2 + 6x_0^2x_1 + x_0 + 15x_1^3 + 10x_1 - 4$ | $g_1 = x_0 + 10x_1 - 4$ 
 $g_2 = x_1^2$ |
| 15 | $f_1 = 4x_0^2 + 7x_0x_1^7 + 4x_0x_1^6 + 2x_0x_1^5 + 8x_0x_1^4 - x_0x_1^3 - 3x_0x_1^2 - 11x_0x_1 + 9x_0$ 
 $f_2 = 6x_0^4x_1 - 5x_0^3x_1^8 + 6x_0^3x_1^7 + 3x_0^3x_1^6 + 12x_0^3x_1^5 + 14x_0^3x_1^4 + 11x_0^3x_1^3 - x_0^3x_1^2 + 14x_0^3x_1 - 9x_0^2x_1^2 + 5x_0^2x_1 + 13x_0x_1^9 + 3x_0x_1^8 - 14x_0x_1^7 + 6x_0x_1^6 + 7x_0x_1^5 - 10x_0x_1^4 + 15x_0x_1^3 + 3x_0x_1^2 + x_1^5 + 5x_1^4 - 9x_1^3 - x_1^2 - 14x_1$ 
 $f_3 = -15x_0^4x_1^3 - 12x_0^3x_1^4 + 5x_0^3x_1^3 - 7x_0^3x_1^2 + 11x_0^2x_1^9 - 7x_0^2x_1^8 + 12x_0^2x_1^7 - 14x_0^2x_1^6 - 6x_0^2x_1^5 - 14x_0^2x_1^4 - 4x_0^2x_1^3 - 8x_0^2x_1^2 + 10x_0^2 + 3x_0x_1^7 + 15x_0x_1^6 - 4x_0x_1^5 - 12x_0x_1^4 - x_0x_1^3 + 8x_0x_1^2 - 12x_0x_1 + 8x_0 + 10$ | $g_1 = x_0 + 10$ 

 $g_2 = x_1^5 + 5x_1^4 - 9x_1^3 - x_1^2 - 14x_1$ |
| 16 | $f_1 = x_0 + 14x_1 + 5$ 
 $f_2 = -11x_0^3 - 8x_0^2x_1^3 + x_0^2x_1 + 13x_0^2 - 2x_0x_1^2 - 9x_0x_1 - x_0 + 7x_1^4 + 3x_1^3 - 10x_1^2$ 
 $f_3 = -x_0^4x_1^3 - 7x_0^4 - 8x_0^3x_1^2 - 5x_0^3x_1 - 4x_0^3 - 3x_0^2x_1^4 - 9x_0^2x_1^2 + 8x_0^2 - 13x_0x_1^3 - 3x_0x_1^2 + 2x_0x_1 + 9x_0 - x_1^4 - 14x_1^3 - 4x_1^2 + 8x_1$ | $g_1 = x_0 + 14x_1 + 5$ 
 $g_2 = x_1^2$ |

Table 16: Success examples from the $\mathcal{D}_3^-(\mathbb{F}_{31})$ test set.

| ID | $F$ | $G$ |
|---|---|---|
| 4 | $f_1 = 11x_0^2x_2 + 8x_0x_1 - x_0x_2^5 + 6x_0x_2^4 + 15x_0x_2^2 + 10x_0x_2 + 12x_1^2x_2^2 + 15x_1^2x_2 - 15x_1x_2^6 + 3x_1x_2^5 - 10x_1x_2^4 + 10x_1x_2^3 - 8x_1x_2^2 + 15x_1x_2 - 4x_1 + x_2^5 + 9x_2^3 + 6x_2^2 - 12x_2$ | $g_1 = x_0 + 14x_2^4 + 9x_2^3 + 7x_2 + 15$ |
| | $f_2 = 5x_0^3x_1x_2 - 4x_0^3x_1 - 2x_0^2x_1^2 + 8x_0^2x_1x_2^5 - 11x_0^2x_1x_2^4 - 5x_0^2x_1x_2^3 + 4x_0^2x_1x_2^2 - 15x_0^2x_1x_2 + 2x_0^2x_1 - 3x_0x_1^3x_2^2 + 4x_0x_1^3x_2 - 4x_0x_1^2x_2^6 + 7x_0x_1^2x_2^5 - 13x_0x_1^2x_2^4 + 13x_0x_1^2x_2^3 + 2x_0x_1^2x_2^2 + 4x_0x_1^2x_2 + x_0x_1^2 - 8x_0x_1x_2^5 - 10x_0x_1x_2^3 + 14x_0x_1x_2^2 + 3x_0x_1x_2 - 3x_0x_2 - 3x_1x_2^5 + 4x_1x_2^3 + 13x_1x_2^2 + 5x_1x_2 - 11x_2^5 + 4x_2^4 + 10x_2^2 - 14x_2$ | $g_2 = x_1 - 9x_2^4 - 4x_2^3 - 11$ |
| | $f_3 = 12x_0^5x_1 + 13x_0^4x_1x_2^4 + 15x_0^4x_1x_2^3 - 9x_0^4x_1x_2 + 7x_0^4x_1 - 4x_0^3x_1x_2^4 - 7x_0^3x_1x_2^3 - 2x_0^3x_1x_2 + 9x_0^3x_1 - x_0^3x_2^3 + 9x_0^3x_2 + 9x_0^2x_1x_2^5 - 12x_0^2x_1x_2^3 + 11x_0^2x_1x_2^2 - 15x_0^2x_1x_2 - 14x_0^2x_2^7 - 9x_0^2x_2^6 + 2x_0^2x_2^5 + 12x_0^2x_2^4 - 15x_0^2x_2^3 + x_0^2x_2^2 + 13x_0^2x_2 + 13x_0x_1^2x_2^4 - 7x_0x_1^2x_2^3 + 7x_0x_1x_2^8 + 11x_0x_1x_2^7 + 15x_0x_1x_2^6 - 13x_0x_1x_2^5 + 12x_0x_1x_2^4 + 11x_0x_1x_2^3 - 13x_0x_1x_2^2 + 7x_0x_1x_2 + 14x_0x_2^7 - x_0x_2^5 + 9x_0x_2^4 - 13x_0x_2^3 + 14x_0x_2^2 - x_0x_2 + x_1 - 9x_2^4 - 4x_2^3 - 11$ | $g_3 = x_2^5 + 9x_2^3 + 6x_2^2 - 12x_2$ |
| | $f_4 = 14x_0^3x_1^3x_2 + 10x_0^2x_1^3x_2^5 + 2x_0^2x_1^3x_2^4 + 5x_0^2x_1^3x_2^2 - 7x_0^2x_1^3x_2 - 3x_0^2x_1^2x_2^2 - 5x_0x_1^3x_2 - 11x_0x_1^2x_2^6 + 4x_0x_1^2x_2^5 + 10x_0x_1^2x_2^3 + 12x_0x_1^2x_2^2 + x_0 + 8x_1^4x_2^3 + 10x_1^4x_2^2 - 10x_1^3x_2^7 - 3x_1^3x_2^6 + 14x_1^3x_2^5 + 3x_1^3x_2^4 + 6x_1^3x_2^3 + 8x_1^3x_2^2 + 13x_1^3x_2 + x_1^3 + 3x_1^2x_2^6 - 14x_1^2x_2^4 - 4x_1^2x_2^3 + 10x_1^2x_2^2 - 7x_1^2x_2 - 11x_1^2 + 14x_2^4 + 9x_2^3 + 7x_2 + 15$ | |
| 5 | $f_1 = 7x_0x_1 - 8x_0 - 6x_1^2x_2$ | $g_1 = x_0 + 10$ |
| | $f_2 = -5x_0^2x_1 - 12x_0^2 + 12x_0x_1^2x_2^2 - 9x_0x_1^2x_2 + 4x_0x_1x_2^2 + x_0 + 3x_1^3x_2^3 + 10$ | $g_2 = x_1 - 10$ |
| | $f_3 = -12x_0x_1x_2^2 - x_0x_1x_2 - 4x_0x_2^2 - 3x_1^2x_2^3 - 10x_1x_2 + x_1 - 10$ | $g_3 = x_2$ |
| | $f_4 = -15x_0^2x_1 + 10x_0x_1^2x_2 + 12x_0x_1^2 + 11x_0x_1x_2^2 - 9x_0x_1x_2 + 5x_0x_1 - 13x_1^3x_2^2 - 3x_1^2x_2^2 - 4x_1^2 - 15x_1x_2^2 + 11x_1x_2 + x_2$ | |
| 9 | $f_1 = 0$ | $g_1 = x_0 + 12x_2^2 + 13x_2 - 2$ |
| | $f_2 = 12x_0x_1x_2^2 - 14x_0x_1x_2 + 11x_1x_2^4 - 12x_1x_2^3 + 11x_1x_2^2 - 3x_1x_2$ | $g_2 = x_1 + 5x_2^2 + 14x_2 + 9$ |
| | $f_3 = -9x_0^2x_1x_2^3 - 5x_0^2x_1x_2^2 + 15x_0x_1x_2^5 + 9x_0x_1x_2^4 + 15x_0x_1x_2^3 + 10x_0x_1x_2^2 + x_0 + 12x_2^2 + 13x_2 - 2$ | $g_3 = x_2^3$ |
| | $f_4 = 12x_0^3x_1 - 2x_0^3x_2 - 14x_0^2x_1^2x_2^2 + 6x_0^2x_1^2x_2 - 11x_0^2x_1x_2^2 + x_0^2x_1x_2 + 7x_0^2x_1 + 7x_0^2x_2^3 + 5x_0^2x_2^2 + 4x_0^2x_2 + 13x_0x_1^2x_2^4 + 14x_0x_1^2x_2^3 + 13x_0x_1^2x_2^2 - 12x_0x_1^2x_2 + x_2^3$ | |
| | $f_5 = 8x_0^2x_1^2x_2^3 + x_0^2x_1^2x_2^2 + 7x_0^2x_1^2 - 10x_0^2x_2^3 - 3x_0x_1^2x_2^5 - 8x_0x_1^2x_2^4 - 3x_0x_1^2x_2^3 - 11x_0x_1^2x_2^2 - 2x_0x_1^2x_2 - 14x_0x_1^2 - 13x_0x_1x_2^3 + 10x_0x_1x_2^2 + 11x_0x_2^3 + x_1x_2^5 + 13x_1x_2^4 + x_1x_2^3 + 11x_1x_2^2 + x_1 + 8x_2^5 - 12x_2^4 + 9x_2^3 + 5x_2^2 + 14x_2 + 9$ | |

| ID | $F$ | $G$ |
|---|---|---|
| 0 | $f_1 = -12x_0x_3$ 
 $f_2 = -5x_0^3x_3 + 3x_0x_3 + 6x_1^2x_3^2 - x_1x_3^2 + x_2 - 7x_3 + 7$ 
 $f_3 = -4x_0^2x_2x_3 - 5x_0x_1x_2^2x_3 + 7x_0x_2^2x_3 - 10x_1^3x_2^2x_3^2 - 5x_1^2x_2^2x_3^2 - 12x_1x_2^3 + 8x_1x_2^2x_3^2 - 9x_1x_2^2x_3 + 9x_1x_2^2 + x_1 - 8x_2^3 - 6x_2^2x_3 + 6x_2^2 + 5$ 
 $f_4 = 15x_0^3 - 4x_0^2 + 5x_0x_1^2x_2x_3 - 14x_0x_1x_3 - 10x_0x_2x_3^2 + 10x_1^4x_2x_3^2 - 12x_1^3x_2x_3^2 + 12x_1^2x_2^2 + 9x_1^2x_2x_3 - 9x_1^2x_2 - 2x_1x_2 + 12x_1x_3 - 10x_2 + x_3$ 
 $f_5 = 6x_0^5x_1 - 14x_0^4x_1 + 13x_0^3x_1^2x_3 + 11x_0^2x_1^2x_3 - 12x_0^2x_1x_3 + 13x_0x_1x_3^2 + 5x_0x_2^2x_3^2 - 2x_0x_3^2 + x_0 + 8x_1^4 - 5x_1^3x_3^3 - 8x_1^3x_3 + 9x_1^3 + 2x_1^2x_3^3 - 9x_1^2x_3 - 6x_1x_2x_3 + 11x_1x_3^3 + 11x_1x_3^2 - 11x_1x_3 - 11x_2x_3 + 15x_3^2 - 15x_3 + 8$ | $g_1 = x_0 + 8$ 
 $g_2 = x_1 + 5$ 

 $g_3 = x_2 + 7$ 




 $g_4 = x_3$ |
| 6 | $f_1 = 4x_0x_1x_3 + 11x_0x_3^4 + 3x_0x_3^3 + 10x_0x_3^2 + 10x_0x_3 + x_3^4$ 
 $f_2 = -4x_0^2x_1^2x_3 - 11x_0^2x_1x_3^4 - 3x_0^2x_1x_3^3 - 10x_0^2x_1x_3^2 - 10x_0^2x_1x_3 - x_0x_1x_3^4 + x_0 + 9x_3^3 - 15$ 
 $f_3 = -11x_0^3x_1x_3 - 7x_0^3x_3^4 + 15x_0^3x_3^3 - 12x_0^3x_3^2 - 12x_0^3x_3 - 9x_0^2x_1x_2^2x_3 + 11x_0^2x_1x_2 + 14x_0^2x_2^2x_3^4 + x_0^2x_2^2x_3^3 - 7x_0^2x_2^2x_3^2 - 7x_0^2x_2^2x_3 + 5x_0^2x_3^4 + 6x_0x_1x_2x_3^3 - 10x_0x_1x_2 - 10x_0x_2^2x_3^4 - 13x_0x_2^2x_3 + x_1 + 7x_2^2x_3^4 + 9x_2^2x_3 - 5x_3^3 - 7x_3^2 - 13x_3 - 13$ 
 $f_4 = -4x_0^2x_1x_2 - 11x_0^2x_2x_3^3 - 3x_0^2x_2x_3^2 - 10x_0^2x_2x_3 - 10x_0^2x_2 + 6x_0x_1^3 - 14x_0x_1^2x_3^2 + 8x_0x_1x_3^5 + 5x_0x_1x_3^4 - 4x_0x_1x_3^3 - 4x_0x_1x_3^2 - 8x_1^3x_3^3 + 3x_1^3 + 12x_1x_3^5$ 
 $f_5 = 2x_0^3x_3 - 13x_0^2x_3^4 + x_0^2x_3 + 5x_0x_1x_2x_3^3 - 15x_0x_1x_3^4 + 6x_0x_2x_3^6 - 4x_0x_2x_3^5 - 3x_0x_2x_3^4 - 3x_0x_2x_3^3 + 13x_0x_3^7 + 12x_0x_3^6 + 9x_0x_3^5 + 9x_0x_3^4 + 8x_1^3x_2 - 9x_1^2x_2x_3^3 + 6x_1^2x_2x_3^2 - 11x_1^2x_2x_3 - 11x_1^2x_2 + 9x_2x_3^6 + x_2 + 4x_3^7 - 8x_3^3 + 3x_3 - 9$ 
 $f_6 = x_0^2x_1^2x_2x_3 + 5x_0^2x_1x_2^2x_3 - 5x_0^2x_1x_2x_3^4 - 7x_0^2x_1x_2x_3^3 - 13x_0^2x_1x_2x_3^2 - 13x_0^2x_1x_2x_3 + 6x_0^2x_2^2x_3^4 - 4x_0^2x_2^2x_3^3 - 3x_0^2x_2^2x_3^2 - 3x_0^2x_2^2x_3 - 12x_0^2x_2^2 + 3x_0^2x_2x_3^3 - 5x_0^2x_2x_3 + 15x_0^2x_2 + 5x_0x_1x_2^3x_3 + 10x_0x_1x_2 + 13x_0x_1x_3^3 - 13x_0x_1x_3^2 - 15x_0x_1x_3 + 3x_0x_1 + 6x_0x_2^3x_3^4 - 4x_0x_2^3x_3^3 - 3x_0x_2^3x_3^2 - 3x_0x_2^3x_3 + 9x_0x_2^2x_3^4 - 7x_0x_2^2 + 3x_0x_3^5 - 2x_0x_3^4 + 14x_0x_3^3 + 14x_0x_3^2 + 11x_1^3x_2 + 5x_1^3x_3^3 + 2x_1^3x_3 - 6x_1^3 - 15x_1x_2^2x_3 - 2x_1x_3^4 + 15x_1x_3^3 - 6x_1x_3^2 - 7x_1x_3 - 12x_2^3 + 13x_2^2x_3^4 + 14x_2^2x_3^3 + 9x_2^2x_3^2 + 4x_2^2x_3 - 4x_2^2 + 6x_3^6 - 13x_3^5 - 15x_3^4 + 7x_3^3 - 15x_3^2$ | $g_1 = x_0 + 9x_3^3 - 15$ 

 $g_2 = x_1 - 5x_3^3 - 7x_3^2 - 13x_3 - 13$ 


 $g_3 = x_2 - 8x_3^3 + 3x_3 - 9$ 






 $g_4 = x_3^4$ |

Table 18: Success examples from the $\mathcal{D}_5^-\left(\mathbb{F}_{31}\right)$ test set.

| ID | $F$ | $G$ |
|---|---|---|
| 0 | $f_1 = 13x_0x_1^3 + 4x_1^3x_4 + 9x_1^3 + x_4^2$ | $g_1 = x_0 - 14x_4 + 15$ |
| | $f_2 = 12x_0x_1^3x_2^3 + 5x_0x_1^3x_3^2x_4 - 13x_1^3x_2^3x_4 - 6x_1^3x_3^3 - 8x_1^3x_3^2x_4^2 + 13x_1^3x_3^2x_4 - 11x_2^3x_4^2 - 2x_3^2x_4^3 + x_3 + 8x_4 - 4$ | $g_2 = x_1 + 8x_4$ |
| | $f_3 = -6x_0x_1^3x_2 - 11x_0x_2x_3^2 + 5x_0x_2x_3x_4 + 13x_0x_2x_3 - 9x_1^3x_2x_4 + 3x_1^3x_2 + 11x_1x_2x_3^2 - 5x_1x_2x_3x_4 - 13x_1x_2x_3 + x_1 - 10x_2x_4^2 + 8x_4$ | $g_3 = x_2 - 14x_4$ |
| | $f_4 = -6x_0^2x_1^3x_3x_4 - 9x_0x_1^3x_3x_4^2 + 3x_0x_1^3x_3x_4 + 15x_0x_1x_3 + 6x_0x_3^2 - 10x_0x_3x_4^3 + 13x_0x_3x_4 + 7x_0x_3 + x_0 - x_1x_2 + 5x_1x_3^2 + 9x_1x_3x_4 + 11x_1x_3 - 8x_2x_4 - 14x_4 + 15$ | $g_4 = x_3 + 8x_4 - 4$ |
| | $f_5 = 10x_0^3x_2 - 7x_0^2x_1^3x_3x_4 + 15x_0^2x_2x_4 - 5x_0^2x_2 - 3x_0x_1^3x_2x_4 + 5x_0x_1^3x_3x_4^2 - 12x_0x_1^3x_3x_4 + x_0x_2x_3^2 + 9x_0x_3x_4^3 + 7x_0x_3 - 6x_0x_4 + 3x_0 + 11x_1^3x_2x_4^2 - 14x_1^3x_2x_4 - 3x_1^2 + 7x_1x_4 + 12x_2^3x_3 + 3x_2^3x_4 + 14x_2^3 - 14x_2x_3^2x_4 + 15x_2x_3^2 - 5x_2x_4^3 + x_2 - 14x_4$ | $g_5 = x_4^2$ |
| 8 | $f_1 = 7x_1^2x_3 - 12x_1^2 - 12x_1x_3^2 - 6x_1x_3 - 3x_2^3x_4 + 8x_2^2x_4$ | $g_1 = x_0 - 12$ |
| | $f_2 = 13x_1^3x_2x_3^2 - 9x_1^3x_2x_3 - 9x_1^2x_2x_3^3 + 11x_1^2x_2x_3^2 + 8x_1^2x_2x_3x_4^2 + 4x_1^2x_2x_4^2 - 10x_1x_2^4x_3x_4 + 6x_1x_2^3x_3x_4 + 4x_1x_2x_3^2x_4^2 + 2x_1x_2x_3x_4^2 + x_1 + x_2^4x_4^3 - 13x_2^3x_3^3 + 11$ | $g_2 = x_1 + 11$ |
| | $f_3 = -4x_0^2x_1^2 - 13x_0^2x_1 + 8x_2^2x_3^4 + 4x_2^2x_3^3 + 4x_1x_3^5 + 2x_1x_3^4 + 12x_1x_3x_4^2 + x_2^3x_3^3x_4 - 13x_2^2x_3^3x_4 + 8x_3x_4^2 + x_3 - 15$ | $g_3 = x_2 - 13$ |
| | $f_4 = -10x_0x_1^2 + 5x_0x_3^2 - 13x_0x_3 + 7x_1^2x_3^3 + 2x_1^2x_3^2x_4^2 - 12x_1^2x_3^2 + x_1^2x_3x_4^2 - 4x_1^2 - 12x_1x_3^4 + x_1x_3^3x_4^2 - 6x_1x_3^3 - 15x_1x_3^2x_4^2 + 14x_1x_3^2 - 12x_1x_3x_4 - 6x_1x_4 - 3x_2^3x_3^2x_4 + 8x_2^3x_3x_4^3 + 8x_2^2x_3^2x_4 - 11x_2^2x_3x_4^3 + x_2 - x_3^2 - 13$ | $g_4 = x_3 - 15$ |
| | $f_5 = 3x_0x_1^2x_2x_4 + 5x_0x_1^2x_3 - 4x_0x_1^2x_4^3 - 13x_0x_1^2 + 7x_0x_2x_3^2 - 12x_0x_2x_3 + 8x_1^4x_2x_3 + 4x_1^4x_2 + 4x_1^3x_2x_3^2 + 2x_1^3x_2x_3 + x_1^2x_2^4x_4 - 13x_1^2x_2^3x_4 - 5x_1^2x_2x_4 - 14x_1^2x_4^3 - 13x_1x_4^3 + 9x_2^2x_4 - 12x_2x_3^3 + 7x_2x_4 + 13x_4^3 + x_4$ | $g_5 = x_4$ |
| | $f_6 = 11x_0^2x_1x_3 - 3x_0^2x_3 + 8x_0x_1^5 + 10x_0x_1^2x_4 - 10x_0x_1x_4 - 3x_1^5 - 7x_1^3x_2 - 2x_1^3 + 7x_1^2x_3^3 - 12x_1^2x_3^2 - 12x_1x_3^3x_3^2 - 6x_1x_3^3 + 15x_1x_2x_3x_4 - 3x_2^6x_4 + 8x_2^5x_4 + x_2^2x_3x_4 - 15x_2^2x_4$ | |
| | $f_7 = 10x_0^2x_1x_2x_3 - 14x_0^2x_2x_3 - 2x_0x_1^3x_3x_4 + 2x_0x_1^2x_3^2 + x_0x_1^2x_3 + 3x_0x_1x_2x_4 + x_0x_1x_3^3 - 15x_0x_1x_3^2 - 11x_0x_1x_4 + 8x_0x_2^3x_3x_4 - 11x_0x_2^2x_3x_4 + x_0 + 11x_1^4x_3^2 - 10x_1^4x_3 - 10x_1^3x_3^3 - 5x_1^3x_3^2 - 7x_1^3x_3x_4 + 13x_1^2x_2^3x_3x_4 - 14x_1^2x_2^2x_3x_4 + 14x_1^2x_3x_4 + 11x_1x_2^2x_3x_4 + 2x_1x_2^2x_3 + x_1x_2^2 - 6x_1x_2x_3x_4 + 13x_1x_2x_4^2 - 9x_1x_2x_4 + 15x_1x_3x_4 - 12x_2x_4^2 - 12$ | |

Table 19: Failure examples from the $\mathcal{D}_n^-(\mathbb{Q})$ test sets for $n = 2, 3, 4, 5$ (ground truth v.s. prediction).

| ID | $G$ (Ground Truth) | $G'$ (Transformer) |
|---|---|---|
| 15 | $g_1 = x_0 - 1/4x_1^4 - 5/4x_1^3 + 2/3x_1 - 5/4$
$g_2 = x_1^5 - 2/3x_1^4 - 4/3x_1^3 + 2$ | $g_1' = x_0 + 1/4x_1^4 + 2/3x_1 - 5/4$
$g_2' = x_1^5 - 2/3x_1^4 - x_1^3 - x_1^2 + 2$ |
| 26 | $g_1 = x_0 - 1/2x_1^4 - 1/5x_1^3 - 1/2x_1^2 - x_1 + 2/3$
$g_2 = x_1^5 - 3/5x_1 - 4/3$ | $g_1' = x_0 - 1/2x_1^4 - 1/5x_1^3 + 1/2x_1^2 - x_1 + 2/3$
$g_2' = x_1^5 - 1/2x_1^4 - 3/5x_1 - 4/3$ |
| 36 | $g_1 = x_0 - 2x_1$
$g_2 = x_1^5 + 5/2x_1^4 - 1/4x_1^3 + 1/2x_1^2 + 1/3$ | $g_1' = x_0 - 2x_1$
$g_2' = x_1^5 + 5/2x_1^4 - 5/4x_1^3 + 1/2x_1^2 + 1/3$ |
| 12 | $g_1 = x_0 + 4x_2^4 - 1/5x_2^2 + 5/4x_2 + 1/2$
$g_2 = x_1 - 5/2x_2^4 + 1/5x_2^3 - 2/3x_2^2 - 1/3x_2 + 3/2$
$g_3 = x_2^5 - 1/4x_2^3 - 3/2x_2^2 - 1/5x_2 - 5/3$ | $g_1' = x_0 + 4x_2^4 - 1/5x_2^2 + 5/4x_2 + 1/2$
$g_2' = x_1 - 5/2x_2^4 + 1/5x_2^3 - 2/3x_2^2 - 1/3x_2 + 3/2$
$g_3' = x_2^5 - 2/3x_2^3 - 3/2x_2^2 - 1/5x_2 - 5/3$ |
| 15 | $g_1 = x_0 + 2/3x_2^4 + x_2^3 + 4x_2^2 - 5/3x_2 + 4$
$g_2 = x_1 + 2x_2^3 + 5/3x_2 + 2$
$g_3 = x_2^5 + 4x_2^4 - x_2^3 + x_2^2$ | $g_1' = x_0 + 2/3x_2^4 + x_2^3 + 4x_2^2 - 5/3x_2 + 4$
$g_2' = x_1 + 5/3x_2 + 2$
$g_3' = x_2^5 + 4x_2^4 - x_2^3 + x_2^2$ |
| 16 | $g_1 = x_0 + 2x_2$
$g_2 = x_1 - 1/5$
$g_3 = x_2^3 - 2x_2^2 + 1/5x_2 + 1$ | $g_1' = x_0 + 2x_2$
$g_2' = x_1 - 1/5$
$g_3' = x_2^3 + 1/5x_2 + 1$ |
| 6 | $g_1 = x_0 - 2/3x_3^3$
$g_2 = x_1 - 4x_3^2 + 1/4x_3$
$g_3 = x_2 - x_3^4 + x_3^2 - 3/5$
$g_4 = x_3^5 - 5x_3^4 + 4x_3^2 + 4x_3$ | $g_1' = x_0 - 2/3x_3^3$
$g_2' = x_1 + 1/4x_3$
$g_3' = x_2 - x_3^4 + x_3^2 - 3/5$
$g_4' = x_3^5 - 5x_3^4 + 4x_3^2 + 4x_3$ |
| 7 | $g_1 = x_0 - 3$
$g_2 = x_1 - 1/5x_3^2$
$g_3 = x_2 + 2/5x_3^4 + 5/2x_3^2 - 2/3x_3 - 3/5$
$g_4 = x_3^5 - 4/5x_3^4 - x_3 - 1$ | $g_1' = x_0 - 3$
$g_2' = x_1 - 1/5x_3^2$
$g_3' = x_2 + 2/5x_3^4 + 5/2x_3^2 - 2/3x_3 - 3/5$
$g_4' = x_3^5 - 4/5x_3^4 - 3x_3^3 - x_3 - 1$ |
| 16 | $g_1 = x_0 + 1/4x_3^2 - 2$
$g_2 = x_1 + 2x_3^3$
$g_3 = x_2 - 2x_3^3 + 3/2$
$g_4 = x_3^5 - x_3^4 - 5x_3^3 - 1/2$ | $g_1' = x_0 + 1/4x_3^2 - 2$
$g_2' = x_1 + 2x_3^3$
$g_3' = x_2 - 2x_3^3 + 3/2$
$g_4' = x_3^5 - x_3^4 - 5x_3^3 - 1/4x_3^2 + 3/2$ |
| 1 | $g_1 = x_0 + 5x_4^4 - 3/5x_4^3 + 4x_4^2 + 2$
$g_2 = x_1 - 2/5x_4^4 + 1/3x_4^3 + x_4^2 - 1/2x_4 + 3/5$
$g_3 = x_2 - 3/5x_4$
$g_4 = x_3 + 4/5x_4^4 + x_4^3 - 3/5x_4$
$g_5 = x_4^5 - 5x_4^4 + 3/2x_4^3 - x_4 - 5$ | $g_1' = x_0 + 5x_4^4 - 3/5x_4^3 + 4x_4^2 + 2$
$g_2' = x_1 - 2/5x_4^4 + 1/3x_4^3 + x_4^2 - 1/2x_4 + 3/5$
$g_3' = x_2 - 2/5x_4$
$g_4' = x_3 + 4/5x_4^4 + x_4^3 - 3/5x_4$
$g_5' = x_4^5 + 3/2x_4^3 - x_4 - 5$ |
| 7 | $g_1 = x_0 - x_4^2 + 4x_4 - 5/3$
$g_2 = x_1 - x_4^2 + 1$
$g_3 = x_2 - 1$
$g_4 = x_3 + 2x_4^2 + 4/3x_4$
$g_5 = x_4^3$ | $g_1' = x_0 - x_4^2 + 4x_4 - 5/3$
$g_2' = x_1 - x_4^2 + 1$
$g_3' = x_2 - 1$
$g_4' = x_3 + 2x_4^2 + 4/3x_4$
$g_5' = x_4^3 + 2/3x_4^2$ |
| 8 | $g_1 = x_0 + x_4^2 - 5$
$g_2 = x_1 + 1/3x_4^4 + 4/3x_4^3 + 5x_4 + 3/4$
$g_3 = x_2 + 1/2x_4^2 - x_4$
$g_4 = x_3 - 5/4x_4^4 - x_4^3 + x_4^2$
$g_5 = x_4^5 - 4/3x_4^3 + x_4$ | $g_1' = x_0 + x_4^2 - 5$
$g_2' = x_1 + 1/3x_4^4 + 4/3x_4^3 + 5x_4 + 3/4$
$g_3' = x_2 - x_4$
$g_4' = x_3 - 5/4x_4^4 - x_4^3 + x_4^2$
$g_5' = x_4^5 - 4/3x_4^3 + x_4$ |

Table 20: Failure examples from the $\mathcal{D}_n^-(\mathbb{F}_7)$ test sets for $n = 2, 3, 4, 5$ (ground truth v.s. prediction).

| ID | $G$ (Ground Truth) | $G'$ (Transformer) |
|---|---|---|
| 2 | $g_1 = x_0 - 3x_1^2 + 1$ 
 $g_2 = x_1^3 - 2$ | $g_1' = x_0 - 3x_1^2 + 1$ 
 $g_2' = x_1^3 - 2x_1^2 - 2$ |
| 6 | $g_1 = x_0 - x_1^4 + 3x_1^3 + 3x_1$ 
 $g_2 = x_1^5 - 2x_1^4 - x_1^3$ | $g_1' = x_0 - x_1^4 + 3x_1^3 + 3x_1$ 
 $g_2' = x_1^5 - x_1^4 - x_1^3$ |
| 8 | $g_1 = x_0 + 3x_1^4 + 2x_1^3 + 2x_1$ 
 $g_2 = x_1^5 - 3x_1^3 + 2x_1^2$ | $g_1' = x_0 + 3x_1^4 + 2x_1^3 + 2x_1$ 
 $g_2' = x_1^5 - 3x_1^3 - x_1^2$ |
| 8 | $g_1 = x_0 - x_2^2$ 
 $g_2 = x_1 - 1$ 
 $g_3 = x_2^5 - x_2^4 + 2x_2^3 - 3x_2$ | $g_1' = x_0 - x_2^2$ 
 $g_2' = x_1 - 1$ 
 $g_3' = x_2^5 - x_2^4 + 2x_2^3 - x_2^2$ |
| 12 | $g_1 = x_0 + 1$ 
 $g_2 = x_1 - 3x_2^2 - 2$ 
 $g_3 = x_2^3 - 1$ | $g_1' = x_0 + 1$ 
 $g_2' = x_1 - 3x_2^2 + x_2 + 3$ 
 $g_3' = x_2^3 + 1$ |
| 18 | $g_1 = x_0 - 3x_2^4 - 2x_2^3 - x_2^2 + 1$ 
 $g_2 = x_1 - x_2^4 + 3x_2 - 2$ 
 $g_3 = x_2^5 - 3x_2^4 + 3x_2^3 + x_2^2$ | $g_1' = x_0 - 3x_2^4 - 2x_2^3 - x_2^2 + 1$ 
 $g_2' = x_1 - x_2^4 + 3x_2 - 2$ 
 $g_3' = x_2^5 - 3x_2^4 + x_2^2$ |
| 0 | $g_1 = x_0 + 3x_3^3 - 3x_3^2$ 
 $g_2 = x_1 + 2x_3^3 + x_3^2 - 3x_3$ 
 $g_3 = x_2 + 2x_3^3 - 2$ 
 $g_4 = x_3^4 - 3x_3$ | $g_1' = x_0 + 3x_3^3 - 3x_3^2$ 
 $g_2' = x_1 + 2x_3^3 + x_3^2 - 3x_3$ 
 $g_3' = x_2 + 2x_3^3 - 2$ 
 $g_4' = x_3^4 - x_3$ |
| 4 | $g_1 = x_0 - 3$ 
 $g_2 = x_1 + 3x_3^4 + x_3^3 - 3x_3^2 + 3x_3 - 3$ 
 $g_3 = x_2 + 3x_3^4 - 2$ 
 $g_4 = x_3^5 + 2x_3^3 - 3x_3^2 - 2x_3$ | $g_1' = x_0 - 3$ 
 $g_2' = x_1 + 3x_3^4 + x_3^3 + 3x_3^2 + 3x_3 - 3$ 
 $g_3' = x_2 + 3x_3^4 - 2$ 
 $g_4' = x_3^5 + 2x_3^3 - 3x_3^2 - 2x_3$ |
| 5 | $g_1 = x_0 + 1$ 
 $g_2 = x_1 + 1$ 
 $g_3 = x_2 - 3$ 
 $g_4 = x_3 - 3$ | $g_1' = x_0 - 1$ 
 $g_2' = x_1 + 1$ 
 $g_3' = x_2 - 3$ 
 $g_4' = x_3 - 3$ |
| 1 | $g_1 = x_0 - 3x_4 - 1$ 
 $g_2 = x_1 + 1$ 
 $g_3 = x_2 - 3$ 
 $g_4 = x_3 + 3$ 
 $g_5 = x_4^4$ | $g_1' = x_0 - 3x_4 - 1$ 
 $g_2' = x_1 + 1$ 
 $g_3' = x_2 + 2$ 
 $g_4' = x_3 + 3$ 
 $g_5' = x_4^4$ |
| 2 | $g_1 = x_0 - 3x_4$ 
 $g_2 = x_1 + 3x_4^2 - 2x_4 + 3$ 
 $g_3 = x_2 - 3x_4^2 - 3x_4$ 
 $g_4 = x_3 - 2$ 
 $g_5 = x_4^3 - 3x_4$ | $g_1' = x_0 - x_4$ 
 $g_2' = x_1 + 3x_4^2 - 2x_4 + 3$ 
 $g_3' = x_2 - 3x_4^2 - 3x_4$ 
 $g_4' = x_3 - 2$ 
 $g_5' = x_4^3 - 3x_4$ |
| 14 | $g_1 = x_0 + 2$ 
 $g_2 = x_1 + 2$ 
 $g_3 = x_2 + x_4^3 - 2x_4^2 + 3x_4 - 2$ 
 $g_4 = x_3 - 3$ 
 $g_5 = x_4^4 - x_4$ | $g_1' = x_0 + 2$ 
 $g_2' = x_1 + 2$ 
 $g_3' = x_2 + x_4^3 - 2x_4^2 + 3x_4 - 2$ 
 $g_4' = x_3 - 3$ 
 $g_5' = x_4^4 + x_4$ |

Table 21: Failure examples from the $\mathcal{D}_n(\mathbb{F}_{31})$ test sets for $n = 2, 3, 4, 5$ (ground truth v.s. prediction).

| ID | $G$ (Ground Truth) | $G'$ (Transformer) |
|---|---|---|
| 0 | $g_1 = x_0 - 14x_1^4 + 9x_1^3 + 12x_1^2 + 14x_1 + 15$
$g_2 = x_1^5 + 15x_1^4 - 5x_1^3 - 7x_1^2$ | $g_1' = x_0 - 14x_1^4 + 9x_1^3 + 12x_1^2 + 14x_1 + 15$
$g_2' = x_1^5 - 11x_1^4 - 11x_1^3$ |
| 1 | $g_1 = x_0 - 9$
$g_2 = x_1$ | $g_1' = x_0 - 9$
$g_2' = x_1 - 7$ |
| 3 | $g_1 = x_0 - 8x_1^2 + 8x_1$
$g_2 = x_1^5 - 14x_1^3 + 10x_1^2 + 2x_1 + 5$ | $g_1' = x_0 - 8x_1^2 + 8x_1$
$g_2' = x_1^5 - 3x_1^4 + 10x_1^2 + 2x_1 + 5$ |
| 0 | $g_1 = x_0 + 5$
$g_2 = x_1 - 8x_2^4 + 15x_2^2 - 5x_2 + 11$
$g_3 = x_2^5 - 2x_2^2 + 15x_2 + 4$ | $g_1' = x_0 + 5$
$g_2' = x_1 - 8x_2^4 - 12x_2^2 - 8x_2 - 13$
$g_3' = x_2^5 - 2x_2^2 + 15x_2 + 4$ |
| 1 | $g_1 = x_0 - 14x_2^4 + 13x_2^3 - 10x_2^2 - 6x_2 - 10$
$g_2 = x_1 - 14x_2^4 - 4x_2^3 - 5x_2^2 + 8x_2 + 3$
$g_3 = x_2^5 + 7x_2^4 + 7x_2^2 - 8$ | $g_1' = x_0 - 2x_2^4 + x_2^3 - 10x_2^2 - 6x_2 - 10$
$g_2' = x_1 - 14x_2^4 - 4x_2^3 - 5x_2^2 + 8x_2 + 3$
$g_3' = x_2^5 - 2x_2^4 - 2x_2^3 - 2x_2^2 - 2$ |
| 2 | $g_1 = x_0 + 15x_2^4 + 12x_2^3 - 13x_2^2 - 10x_2 - 7$
$g_2 = x_1 + 9x_2^2$
$g_3 = x_2^5 - 15x_2^3 + 5x_2^2$ | $g_1' = x_0 + 15x_2^4 + 12x_2^3 - 13x_2^2 - 10x_2 - 7$
$g_2' = x_1 + 9x_2^2$
$g_3' = x_2^5 + 14x_2^4 + 14x_2^3 + 14x_2^2 + 6$ |
| 1 | $g_1 = x_0 + 11x_3^4 - 4x_3^3 - 8x_3^2$
$g_2 = x_1 - 7$
$g_3 = x_2 + 2$
$g_4 = x_3^5 + 5x_3^4 - 6x_3^2$ | $g_1' = x_0 + 11x_3^4 - 4x_3^3 - 8x_3^2$
$g_2' = x_1 - 7$
$g_3' = x_2 + 2$
$g_4' = x_3^5 + 5x_3^4 - 11x_3^2$ |
| 3 | $g_1 = x_0 - 6x_3^2 + 13x_3$
$g_2 = x_1 + 12x_3^4 - 11x_3^3 + 10x_3^2 - 15x_3 + 10$
$g_3 = x_2 - 8x_3^2 - 15x_3$
$g_4 = x_3^5 - 2x_3^4 - 2x_3^2 - 5x_3$ | $g_1' = x_0 - 6x_3^2 + 13x_3$
$g_2' = x_1 + 12x_3^4 - 11x_3^3 + 10x_3^2 - 15x_3 + 10$
$g_3' = x_2 - 8x_3^4 - 8x_3^3 - 8x_3^2 - 15x_3$
$g_4' = x_3^5 - 2x_3^4 - 2x_3^2 - 5x_3$ |
| 5 | $g_1 = x_0 - 5x_3 + 13$
$g_2 = x_1 + 11x_3^3 - 13$
$g_3 = x_2 - 5x_3 - 9$
$g_4 = x_3^5 - 2x_3^4 - 11x_3^3 - 6x_3 - 4$ | $g_1' = x_0 - 5x_3 + 13$
$g_2' = x_1 + 11x_3^3 - 13$
$g_3' = x_2 - 5x_3 - 9$
$g_4' = x_3^5 - x_3^4 - 11x_3^3 - 10x_3 - 4$ |
| 1 | $g_1 = x_0 - 10x_4^4 - 12x_4^2 - 11x_4 - 13$
$g_2 = x_1 + 2x_4$
$g_3 = x_2 + 7x_4^4 - 11x_4^3 - 12x_4^2 + 7x_4 - 13$
$g_4 = x_3 + 5x_4^4 - x_4^3 - 15x_4^2 + 4x_4 - 5$
$g_5 = x_4^5 + 13x_4^4 + 8$ | $g_1' = x_0 - 10x_4^4 - 12x_4^2 - 11x_4 - 13$
$g_2' = x_1 + 2x_4$
$g_3' = x_2 + 7x_4^4 - 11x_4^3 - 12x_4^2 + 7x_4 - 13$
$g_4' = x_3 + 5x_4^4 - x_4^3 - 15x_4^2 - 2x_4 - 5$
$g_5' = x_4^5 + 13x_4^4 + 8$ |
| 2 | $g_1 = x_0 + 7x_4^4 + 3x_4^3 + 15x_4^2 + 3x_4 + 9$
$g_2 = x_1 + 3x_4^4 - 12x_4^3 - 6x_4^2 - x_4 - 15$
$g_3 = x_2 + 6$
$g_4 = x_3 - 15x_4^4 + 3x_4^3 - 13x_4^2 + 9x_4$
$g_5 = x_4^5 - 7x_4^3 - 5x_4^2 - 13x_4$ | $g_1' = x_0 - 2x_4^4 + 3x_4^3 + 15x_4^2 + 3x_4 + 9$
$g_2' = x_1 + 6x_4^4 + 11x_4^3 + 12x_4^2 - 5x_4 + 11$
$g_3' = x_2 + 6$
$g_4' = x_3 - 15x_4^4 + 3x_4^3 - 13x_4^2 - 2x_4$
$g_5' = x_4^5 - 7x_4^3 - 5x_4^2 - 13x_4$ |
| 3 | $g_1 = x_0 + 13x_4^4 + 10x_4^3 + 11x_4^2 - 5x_4$
$g_2 = x_1 + 10x_4^4 - 13x_4^3 + 12$
$g_3 = x_2 + 12x_4^4 + 12x_4^3 - 7$
$g_4 = x_3 + 4$
$g_5 = x_4^5 + 9x_4^4 - 3x_4^2$ | $g_1' = x_0 + 13x_4^4 + 10x_4^3 + 11x_4^2 - 5x_4$
$g_2' = x_1 + 10x_4^4 - 13x_4^3 + 12$
$g_3' = x_2 + 12x_4^4 + 12x_4^3 - 7$
$g_4' = x_3 - 15$
$g_5' = x_4^5 + 9x_4^4 - 3x_4^2$ |

Table 22: Runtime comparison (in seconds) of forward generation (F.) and Transformer on timeout examples. Timeout limit: 100 seconds. Red cells indicate timeout. For each configuration, up to first 10 unique samples are shown. Transformer successfully predicted the correct Gröbner bases for these examples.

| Sample ID | $\mathcal{D}_n^-(\mathbb{Q})$ $n=4$ ($\sigma=0.3$) | | | |
|---|---|---|---|---|
| | F. (STD) | F. (SLIMGB) | F. (STDFGLM) | Transformer |
| 23 | 0.005 | 100.0 | 0.008 | 0.110 |
| 49 | 100.0 | 0.018 | 0.008 | 0.253 |
| 64 | 0.006 | 100.0 | 0.008 | 0.147 |
| 171 | 100.0 | 100.0 | 0.015 | 0.238 |
| 341 | 100.0 | 0.011 | 0.008 | 0.212 |
| 384 | 100.0 | 100.0 | 0.009 | 0.142 |
| 423 | 0.123 | 100.0 | 0.008 | 0.167 |
| 530 | 100.0 | 0.005 | 0.010 | 0.197 |
| 542 | 0.005 | 100.0 | 0.008 | 0.212 |
| 552 | 100.0 | 100.0 | 0.008 | 0.185 |

| Sample ID | $\mathcal{D}_n^-(\mathbb{Q})$ $n=5$ ($\sigma=0.2$) | | | |
|---|---|---|---|---|
| | F. (STD) | F. (SLIMGB) | F. (STDFGLM) | Transformer |
| 61 | 0.005 | 100.0 | 0.009 | 0.170 |
| 76 | 0.004 | 100.0 | 0.008 | 0.292 |
| 130 | 100.0 | 0.014 | 0.008 | 0.227 |
| 153 | 100.0 | 19.384 | 0.009 | 0.220 |
| 175 | 100.0 | 34.825 | 0.192 | 0.390 |
| 208 | 100.0 | 0.013 | 0.008 | 0.223 |
| 269 | 100.0 | 0.006 | 0.069 | 0.320 |
| 295 | 0.005 | 100.0 | 0.008 | 0.282 |
| 320 | 0.005 | 100.0 | 0.008 | 0.309 |
| 333 | 0.135 | 100.0 | 0.008 | 0.296 |

| Sample ID | $\mathcal{D}_n^-(\mathbb{F}_7)$ $n=4$ ($\sigma=0.3$) | | | |
|---|---|---|---|---|
| | F. (STD) | F. (SLIMGB) | F. (STDFGLM) | Transformer |
| 407 | 100.0 | 4.788 | 0.013 | 0.189 |
| 717 | 100.0 | 0.004 | 0.007 | 0.135 |
| 765 | 100.0 | 0.004 | 0.007 | 0.122 |
| 915 | 100.0 | 5.832 | 0.009 | 0.136 |
| 916 | 100.0 | 0.022 | 0.008 | 0.266 |

| Sample ID | $\mathcal{D}_n^-(\mathbb{F}_7)$ $n=5$ ($\sigma=0.2$) | | | |
|---|---|---|---|---|
| | F. (STD) | F. (SLIMGB) | F. (STDFGLM) | Transformer |
| 74 | 100.0 | 100.0 | 0.045 | 0.314 |
| 121 | 100.0 | 100.0 | 0.011 | 0.285 |
| 256 | 100.0 | 100.0 | 0.008 | 0.144 |
| 267 | 100.0 | 0.005 | 0.008 | 0.284 |
| 274 | 100.0 | 0.006 | 0.007 | 0.209 |
| 433 | 100.0 | 38.895 | 0.013 | 0.285 |
| 438 | 0.432 | 100.0 | 0.011 | 0.270 |
| 492 | 100.0 | 0.020 | 0.007 | 0.144 |
| 568 | 100.0 | 0.008 | 0.007 | 0.223 |
| 766 | 100.0 | 100.0 | 0.009 | 0.144 |

| Sample ID | $\mathcal{D}_n^-(\mathbb{F}_{31})$ $n=4$ ($\sigma=0.3$) | | | |
|---|---|---|---|---|
| | F. (STD) | F. (SLIMGB) | F. (STDFGLM) | Transformer |
| 954 | 100.0 | 0.157 | 0.008 | 0.123 |

| Sample ID | $\mathcal{D}_n^-(\mathbb{F}_{31})$ $n=5$ ($\sigma=0.2$) | | | |
|---|---|---|---|---|
| | F. (STD) | F. (SLIMGB) | F. (STDFGLM) | Transformer |
| 196 | 100.0 | 0.051 | 0.007 | 0.267 |
| 715 | 100.0 | 26.161 | 0.008 | 0.160 |

Table 23: Dataset profile comparison between $\mathcal{D}^u(k)$ and $\mathcal{D}_n^-(k)$. The standard deviation is shown in the superscript.

| Metric | $\mathcal{D}^u(\mathbb{Q})$ | $\mathcal{D}_n^-(\mathbb{Q})$ | $\mathcal{D}^u(\mathbb{F}_7)$ | $\mathcal{D}_n^-(\mathbb{F}_7)$ | $\mathcal{D}^u(\mathbb{F}_{31})$ | $\mathcal{D}_n^-(\mathbb{F}_{31})$ |
|---|---|---|---|---|---|---|
| Size of $F$ | $2.64^{(\pm 0.73)}$ | $2.56^{(\pm 0.71)}$ | $3.06^{(\pm 0.82)}$ | $3.03^{(\pm 0.81)}$ | $3.04^{(\pm 0.79)}$ | $3.01^{(\pm 0.81)}$ |
| Max degree in $F$ | $4.97^{(\pm 1.65)}$ | $7.44^{(\pm 1.91)}$ | $5.46^{(\pm 1.77)}$ | $7.91^{(\pm 2.01)}$ | $5.56^{(\pm 1.77)}$ | $8.18^{(\pm 1.98)}$ |
| Min degree in $F$ | $2.50^{(\pm 1.50)}$ | $4.19^{(\pm 1.91)}$ | $2.70^{(\pm 1.61)}$ | $4.33^{(\pm 2.04)}$ | $2.85^{(\pm 1.63)}$ | $4.64^{(\pm 2.06)}$ |
| # of terms in $F$ | $10.56^{(\pm 6.02)}$ | $15.64^{(\pm 7.69)}$ | $13.46^{(\pm 7.14)}$ | $20.02^{(\pm 9.60)}$ | $13.73^{(\pm 7.10)}$ | $20.79^{(\pm 9.68)}$ |
| Gröbner ratio | $0^{(\pm 0)}$ | $0^{(\pm 0)}$ | $0.002^{(\pm 0.045)}$ | $0.002^{(\pm 0.045)}$ | $0^{(\pm 0)}$ | $0^{(\pm 0)}$ |
| Size of $G$ | $2^{(\pm 0)}$ | $2^{(\pm 0)}$ | $2^{(\pm 0)}$ | $2^{(\pm 0)}$ | $2^{(\pm 0)}$ | $2^{(\pm 0)}$ |
| Max degree in $G$ | $2.48^{(\pm 1.32)}$ | $4.09^{(\pm 1.31)}$ | $2.53^{(\pm 1.33)}$ | $3.94^{(\pm 1.30)}$ | $2.53^{(\pm 1.38)}$ | $4.09^{(\pm 1.30)}$ |
| Min degree in $G$ | $1.21^{(\pm 0.54)}$ | $2.56^{(\pm 1.24)}$ | $1.20^{(\pm 0.53)}$ | $2.37^{(\pm 1.21)}$ | $1.21^{(\pm 0.57)}$ | $2.60^{(\pm 1.24)}$ |
| # of terms in $G$ | $3.69^{(\pm 1.65)}$ | $6.65^{(\pm 2.32)}$ | $3.72^{(\pm 1.64)}$ | $6.31^{(\pm 2.27)}$ | $3.74^{(\pm 1.73)}$ | $6.70^{(\pm 2.32)}$ |
| Gröbner ratio | $1^{(\pm 0)}$ | $1^{(\pm 0)}$ | $1^{(\pm 0)}$ | $1^{(\pm 0)}$ | $1^{(\pm 0)}$ | $1^{(\pm 0)}$ |

Table 24: Accuracy and support accuracy of Transformers on out-distribution evaluation set $\mathcal{D}_2^+(k)$.

| Metric | $\mathcal{D}_2^+(\mathbb{Q})$ | $\mathcal{D}_2^+(\mathbb{F}_7)$ | $\mathcal{D}_2^+(\mathbb{F}_{31})$ |
|---|---|---|---|
| accuracy | 83.8 | 52.9 | 36.1 |
| support acc. | 86.7 | 62.0 | 54.4 |

Table 25: Katsura-n on $\mathbb{Q}[x_1, \ldots, x_n]$. Katsura-5 is not presented because of its complexity.

| $n$ | $F$ | $G$ |
|---|---|---|
| 2 | $f_1 = x_0 + 2x_1 - 1$ 
 $f_2 = x_0^2 - x_0 + 2x_1^2$ | $g_1 = x_0 + 2x_1 - 1$ 
 $g_2 = x_1^2 - \frac{1}{3}x_1$ |
| 3 | $f_1 = x_0 + 2x_1 + 2x_2 - 1$ 
 $f_2 = x_0^2 - x_0 + 2x_1^2 + 2x_2^2$ 
 $f_3 = 2x_0x_1 + 2x_1x_2 - x_1$ | $g_1 = x_0 - 60x_2^3 + \frac{158}{7}x_2^2 + \frac{8}{7}x_2 - 1$ 
 $g_2 = x_1 + 30x_2^3 - \frac{79}{7}x_2^2 + \frac{3}{7}x_2$ 
 $g_3 = x_2^4 - \frac{10}{21}x_2^3 + \frac{1}{84}x_2^2 + \frac{1}{84}x_2$ |
| 4 | $f_1 = x_0 + 2x_1 + 2x_2 + 2x_3 - 1$ 

 $f_2 = x_0^2 - x_0 + 2x_1^2 + 2x_2^2 + 2x_3^2$ 

 $f_3 = 2x_0x_1 + 2x_1x_2 - x_1 + 2x_2x_3$ 

 $f_4 = 2x_0x_2 + x_1^2 + 2x_1x_3 - x_2$ | $g_1 = x_0 - \frac{53230079232}{1971025}x_3^7 + \frac{10415423232}{1971025}x_3^6 + \frac{9146536848}{1971025}x_3^5 - \frac{2158574456}{1971025}x_3^4 - \frac{838935856}{5913075}x_3^3 + \frac{275119624}{5913075}x_3^2 + \frac{4884038}{5913075}x_3 - 1$ 
 $g_2 = x_1 - \frac{97197721632}{1971025}x_3^7 + \frac{73975630752}{1971025}x_3^6 - \frac{12121915032}{1971025}x_3^5 - \frac{2760941496}{1971025}x_3^4 + \frac{814792828}{1971025}x_3^3 - \frac{1678512}{1971025}x_3^2 - \frac{9158924}{1971025}x_3$ 
 $g_3 = x_2 + \frac{123812761248}{1971025}x_3^7 - \frac{79183342368}{1971025}x_3^6 + \frac{7548646608}{1971025}x_3^5 + \frac{3840228724}{1971025}x_3^4 - \frac{2024910556}{5913075}x_3^3 - \frac{132524276}{5913075}x_3^2 + \frac{30947828}{5913075}x_3$ 
 $g_4 = x_3^8 - \frac{8}{11}x_3^7 + \frac{4}{33}x_3^6 + \frac{131}{5346}x_3^5 - \frac{70}{8019}x_3^4 + \frac{1}{3564}x_3^3 + \frac{5}{42768}x_3^2 - \frac{1}{128304}x_3$ |

