# OpenReview forum: "Learning to compute Gröbner bases"
_NeurIPS.cc/2024/Conference — NeurIPS 2024 poster_

### Official Review · Reviewer_q3D1 · 2024-07-02

**Soundness:** 2
**Presentation:** 1
**Contribution:** 2
**Rating:** 3
**Confidence:** 4

**Summary:**

This paper provides a machine learning algorithm to compute Gröbner bases of 0-dimensional ideals in shape position.
To address the backward Gröbner problem, the authors propose an algorithm based on 1) random generation of Gröbner bases by sampling univariate polynomials related to the shape position, then on 2) random generation of a polynomial matrix to generate non-Gröbner sets.
The associated algorithm running time is derived in Theorem 4.8.
After generating tuples with the two above-mentioned steps, the authors rely on Transformers for learning to compute Gröbner bases.

**Strengths:**

- The benchmarks show that computing Gröbner bases with the proposed backward approach can be substantially more efficient than with the usual "forward" algorithms.

**Weaknesses:**

- The paper contains many typos, grammatical mistakes and missing articles. The authors are invited to carefully rewrite the paper to increase the overall quality. Two examples (among many) are in p6, l255: "Consider polynomial ring", "Given dataset size".

- The training approach, via Transformers, does not come with any theoretical guarantees related to global/local convergence, running time, error bounds. No explanation is provided regarding the cases where the method works or fails. The ML algorithms related to transformers are mostly described in the appendix and only vaguely in the main text.

- The paper does not address concrete benchmarks of target applications (e.g. in cryptography or biological systems) where computing Gröbner bases is out of reach or very time consuming. Providing such experiments would definitely make the approach credible.

- Many important aspects (theorem proofs, limitations) are postponed to the appendix. Overall I believe that this work, possibly sound and surely interesting, would be worth publishing but the current conference format (coming together with a short allocated review time) and focus are certainly not the best fit.

**Questions:**

- The framework is limited to 0-dimensional ideals in shape position. It is well known that the shape position assumption is not so strong but the 0-dimensional ideal one is very strong.
Random search applied to higher-dimensional ideals would be more challenging and interesting. Could the authors provide more insights to address such problems?

- Could the authors derive a killer application for which their framework would provide new results or results obtained in a more efficient way than with the forward approach?

**Limitations:**

- There is no section dedicated about the limitations of this work in the main text. Section H is not part of the main text.

---

> ### Author Rebuttal · Authors · 2024-08-06
>
> # NeurIPS 2024 Rebuttal (Reviewer q3D1)
>
> We appreciate your thorough review and valuable feedback. Below, we answer the weaknesses and questions.
>
> ## **On Weaknesses**
>
> **Theoretical guarantees on training.** Providing theoretical guarantees on global/local convergence, running time, and error bounds for the training of Transformers (or deep neural networks) is generally difficult.
> To our knowledge, such guarantees have been (partially) given only for simple models, e.g., two-layer networks or infinitely wide networks. Hence, this should be beyond the scope of our study. The tight page limitation forced us to send the description of Transformers and its training to the appendix, but as mentioned around [l.335], they follow a standard one. The exception is when hybrid embedding is used, where we have a small MLP at the input embedding, a regression head for predicting coefficients, and MSE loss. These are described in Section 5 and Appendix D, but we will review the manuscript to give further details.
>
> **Benchmarks and applications.**
> Showing the superiority of using Transformers for large-scale problems in benchmarks and applications is indeed important. However, to achieve this, we need a few more fundamental steps, e.g.,
> 1. Designing dataset generation algorithms tailored for target applications.
> 2. Discovering an efficient tokenization of large polynomial systems.
>
> On 1), since the characteristics of polynomial systems vary across applications, one needs to design a special way of generating them. Our work did this for zero-dimensional radical ideals as a first step because 0-dimensionality is common in several applications (cf. Global Response).
>
> On 2), we need an efficient representation of polynomial systems for scale-up. The scope of this study is to establish the problem of learning to compute Gröbner bases, and thus, we tested its learnability based on a standard model and training. The tokenization of polynomials also follows a standard one; for example, $\{x^2-10, y\}$ is tokenized as $\texttt{[x, **, 2, +, -10, <sep>, y]}$, (cf. Section 5). Eventually, a single $n$-variate term requires $O(n)$ tokens, leading to a long input sequence for large $n$. As well known in Transformer literature, the space complexity of the attention mechanism scales as $O(L^2)$ with input length $L$. A potential approach to shorten the input is to embed a term by its coefficient and inject the exponent part as position vectors because this is literally a position in the term space. We are currently working on it in our follow-up paper.
>
> It is worth noting that the current study already observed that Gröbner basis computation is computationally costly for $n=5$ in Table 1, where about 24% of the instances encountered timeout for two standard algorithms. Further, Table 22 shows that for all the algorithms, there are several instances for which Transformers run accurately and significantly faster. Thus, we have already partially observed the Transformer's supremacy.
>
> **Writing and format.**
> We will carefully clean up our manuscript manually and systematically (e.g., using LLM). A grammar checker applied before submission does not seem to work well for sentences with latex codes. As for the placement of the proofs and limitations, in my limited experience, it is common for the NeurIPS papers to have them in Appendix. The paper checklist justifies this; see [l.947],
> > The proofs can either appear in the main paper or the supplemental material, but if they appear in the supplemental material, the authors are encouraged to provide a short proof sketch to provide intuition.
>
> In our manuscript, the intuition is provided below each theorem. About the limitations, because this study tackles a new problem, we needed a thorough discussion of open questions, which led us to send it to Appendix. The checklist does not clarify whether the limitation section should be included in the main text.
>
> According to the submission instructions, accepted papers will get another content page, so we may be able to include the proofs and/or limitations in the main text in such a case.
>
> ## **On Questions**
>
> **0-dimensional ideals.** If an ideal is in shape position, it is also 0-dimensional. Thus, the assumption of shape-positioned ideals cannot be weaker than that of 0-dimensionality. We assume that you intend to mean "assuming 0-dim is strong; further assuming shape position is OK." As several reviewers ask about the assumption, we provide the Global Response to address it. Please kindly refer to it.
>
> **Killer applications.**
> One of the potential killer applications is cryptanalysis. The security of
> cryptography is often reduced to the difficulty of solving certain mathematical problems.
> Recently, the SALSA project [1] has shown that a Transformer-based approach can solve the LWE (Learning With Errors) problem, which is the basis for the security of lattice-based cryptographies. The overview of their approach is as follows:
>
> 1. From the cryptosystem to be analyzed, generate samples of LWE and their solutions as training data and perform training.
> 2. Input the LWE corresponding to the ciphertext to be attacked into the model constructed in Step 1. If the output gives the secret, the cryptosystem is vulnerable.
>
> The computational difficulty of the Gröbner basis problem for 0-dimensional ideals is the basis for the security of AES [2], which is currently the mainstream symmetric key encryption, and multivariate polynomial encryption [3], which provided candidates in the NIST Post-Quantum Cryptography Standardization process. Our future target is to make our proposed efficient data set generation and machine learning the basis for some security analysis in cryptography.
>
> [1] E. Wenger, et al., SALSA: Attacking lattice cryptography with transformers, 2022.
>
> [2] J. Buchmann, et al., Block ciphers sensitive to Gröbner basis attacks. 2005
>
> [3] A. Kipnis, et al., Unbalanced oil and vinegar signature schemes, 1999

---

> > ### Comment · Reviewer_q3D1 · 2024-08-07
> > **Answer to authors' rebuttal**
> >
> > Thanks for your comments, I will maintain my score.

---

> > > ### Author Response · Authors · 2024-08-08
> > >
> > > Thank you for reading our rebuttal and responding promptly.
> > >
> > > We again clarify that your comments/questions are very critical to improving the current work and also directing future work. We believe that we've addressed your comments/questions. Particularly, it is important for us that you kindly comprehend the scope of this study and its value within this scope, with which your evaluation of our work hopefully increases.
> > >
> > > We'd be happy to engage further.

---

### Official Review · Reviewer_pNNy · 2024-07-11

**Soundness:** 4
**Presentation:** 4
**Contribution:** 4
**Rating:** 8
**Confidence:** 5

**Summary:**

This article investigates the use of machine learning techniques to compute Gröbner basis of polynomial systems. This problem consists in, given a term order and a finite set of polynomials $f_1, \cdots, f_m$, to compute another set of polynomials $g_1, \cdots, g_m$ that have the following desirable properties $(P)$:

$(P)$ the $g_i$'s all belong the ideal generated by $f_1, \cdots, f_m$, and such that any leading term in the ideal generated by the $f_i$'s is generated by the $g_i$'s.

To do so, the authors propose a method to first sample training data, which was previously problematic. Then, after embedding the set of polynomials $f_i$'s and $g_i$'s that they embed (after tokenization and a hybrid embedding technique that they detail in the appendix), they make use of a transformer to output the $g_i$'s given the $f_i$'s.

**Strengths:**

- The article proposes a serious study, is well-written and very comprehensible for non-experts in computational algebra and Gröbner basis.

- The introduction and comparison to existing work is clear.

- The articles presents interesting ideas to generate training datasets, using a backward approach. Instead of first sampling the $f_i$'s (non-Gröbner set) and then an associated Gröbner basis, they first generate a set of polynomials formed by some gi's and then generate an associated system of polynomials fi's that thave property (P) (in other words, first generate the solutions, then the problem).

- The authors included example of success for, but also examples of failure, which can be of use for future improvements.

- The authors share in details the results and conclusion of their study, in particular their experiments. Therefore, this article may be of great interest beyond the computational algebra community, including those who have to solve polynomial systems of questions for various applications.

**Weaknesses:**

I do not see a major weakness of this article. It constitutes a serious study of the use of machine learning advancements for computational algebra. One comment (not a weakness): Section D, there is a reference to a Table that is missing (line 728).

**Questions:**

- Line 115, the authors mention the notion of reduced Gröbner basis. Do they mean normalized and in shape position?  The authors may want to clarify this.

- The authors may want (if they think this is relevant) to introduce some short additional comment for non-experts about the Gröbner basis definition line 109. This would help to understand the motivation of the definition. Namely: Understanding and finding solutions of a polynomial system can be adressed via studying the ideal generated by the fi's. For example, if one is able to find if the constant 1 is in the ideal generated by the fi's, then there is no solution for the system. It turns out the condition on the leading terms in the defintion are equivalent to, given a polynomial h, that the remainder after multivariate division of h by the gi's to be zero if and only if the h is in the ideal generated by the fi's (see for instance Definition 1 in [70]).
Hence one can answer the above question (if 1 is in the ideal) -- as well as others-- by performing a multivariate division algorithm taking on 1 and the Gröbner basis. It the output is 0, then 1 belong to the ideal and there is no solution.

**Limitations:**

The authors clearly explain the apparent limitation of their work, in particular to which extent the experiments they conducted are successful. They report extensive experimental results in the appendix, in particular failed examples.

---

> ### Author Rebuttal · Authors · 2024-08-06
>
> We sincerely appreciate your thorough review and strongly positive assessment of our work. We are grateful that you listed many strengths of our work.
>
> **Reduced Gröbner basis.** The reduced Gröbner basis is defined independently from shape position. A Gröbner basis $G$ is called reduced when i) the leading term coefficients are all one and ii) there is no redundancy (see. Definition A.13.) in that any terms of $g \in G$ is not divisible by the leading term of other polynomials in $G$. We will include a few follow-up lines to make it more friendly to readers unfamiliar with Grönber bases.
>
> **Gröbner basis definition for non-experts.** It is indeed very important to broaden the scope of readers by introducing some intuition on the definition of a Gröbner basis. We prepare the paragraph at [l.118] to give such an intuition of Gröbner bases, but it is true that this does not give an intuition of Definition 3.2. The suggested explanation elegantly presents the point of the definition, and we will integrate it into our explanation. Once our paper gets accepted, there will be an additional content page. We will exploit it.
>
> We will clean up our manuscript by taking into account your comments on missing table references and providing a more friendly introduction to Gröbner bases. Again, we thank you for your strong support of our work.

---

> > ### Comment · Reviewer_pNNy · 2024-08-09
> >
> > Thanks for your reply. I will maintain my score and will argue in favor of this work if needed.

---

> > > ### Author Response · Authors · 2024-08-10
> > >
> > > Thank you for your reading and support!

---

### Official Review · Reviewer_WoAv · 2024-07-12

**Soundness:** 3
**Presentation:** 3
**Contribution:** 3
**Rating:** 8
**Confidence:** 2

**Summary:**

The paper proposes to learn to compute Gröbner bases (as the title says). This includes two important problems. 1) Generating the dataset and 3) Finding an appropriate encoding of the problem to feed into a transformer architecture. That is finding an encoding for a system of polynomials to be solved.

The paper gives adequate solution for both of these issues and implements it,

**Strengths:**

The paper tackles a really tough and important problem with a novel approach. This is done while also placing the work nicely in the existing literature.

I enjoyed reading the open questions section in the Appendix.

**Weaknesses:**

The method does not provide any guarantees. In this regard it is then not entirely clear what the benefit would be of having "hints at the right solution".

**Questions:**

1) Could the authors comment on my point raised in the "weakness". Specifically, how do you envisage to use learned Gröbner basis to be useful for any downstream tasks? Would you use them as a heuristic to exactly solve polynomial systems?

2) Why did you limit the rings to Q,R F_p. Why not N or p-adic numbers?

**Limitations:**

Exemplary!

---

> ### Author Rebuttal · Authors · 2024-08-06
>
> Thank you for your strongly positive evaluation of our work!
>
> **Benefits.**
> In general, if the terms of the input polynomial system $f_1,\ldots,f_m$ are fixed and the coefficients $\{c_{i,\alpha}\}$, where $f_i = \sum_{\alpha} c_{i,\alpha} x^{\alpha}$, are considered as parameters, it is known that by using the comprehensive Gröbner basis theory, i.e., the parameterized Gröbner basis theory, the terms appearing in the reduced Gröbner basis $G$ of the ideal $\langle f_1,\ldots,f_m \rangle$ are identical for general coefficient values of $\{c_{i,\alpha}\}$. In other words, this means that in most cases, the combinations of terms on a reduced Gröbner basis are determined only by the combinations of terms in the input polynomial system. This can also be observed from the relatively high support accuracy in our experiments.
>
> From the above, there is still no concrete guarantee that our method will give a correct model, including the coefficients, but it is believed that there is some guarantee that the terms will be guessed correctly. In algebra, guessing the terms of the Gröbner basis gives a global invariant of the ideal, such as the initial ideal or the Hilbert polynomial. Therefore, we can consider using our model as an oracle or heuristic to provide global invariants of ideals (initial ideals, Hilbert polynomials) for downstream tasks. Indeed, there is an accelerated Gröbner basis computation algorithm using Hilbert polynomials [1]. In future work, we can construct efficient polynomial solving and Gröbner basis computation theory by assuming further information (e.g., entire supports) given by Transformers.
>
> **Other fields.**
> The proposed dataset generation works for other coefficient fields, including $\mathbb{Q}_p$. However, as $\mathbb{N}$ is not a field (or even a ring), the set $\mathbb{N}[x_1,\ldots, x_n]$ is beyond the scope of Gröbner basis theory. The ring $\mathbb{Q}_p[x_1, \ldots, x_n]$ is associated with a non-Archimedean geometry, so it should be interesting to see how the learning goes. In our experiments, the learning was more successful with $\mathbb{Q}$, which equips a canonical metric, than with $\mathbb{F}_p$, which has no non-trivial metrics. An experiment on $\mathbb{Q}_p[x_1, \ldots, x_n]$ may serve as the midpoint between them.
>
> [1] C. Traverso, Hilbert functions and the Buchberger algorithm, Journal of Symbolic Computation, 1997.

---

> > ### Comment · Reviewer_WoAv · 2024-08-11
> >
> > I thank the reviewer for their detailed overall rebuttal and intend to maintain my score.
> >
> > (and yes N is not a field, apologies)

---

### Official Review · Reviewer_YTLb · 2024-07-12

**Soundness:** 2
**Presentation:** 3
**Contribution:** 3
**Rating:** 5
**Confidence:** 3

**Summary:**

Gröbner bases are a tool of fundamental importance in the field of computational algebra. Unfortunately known algorithms for computing Gröbner bases are very ineficient, having a running time that is double-exponential on the number of variables. In this work, the authors propose a machine-learning based approach for the computation problem. Instead of devising an algorithm that is always guaranteed to correctly output a Gröbner basis, the authors propose a learning algorithm based on transformers.

To address this learning problem, the authors address other interesting problems such as random generation of Gröbner bases and the Backward Gröbner problem.

**Strengths:**

The paper provide an interesting experimental evaluation on how transformers can be used to learn Gröbner bases. The random generation of Gröbner bases is also interesting.

**Weaknesses:**

The main weakness, in my opinion, is that the paper does not attempt to provide a characterization of a subclass of polynomial systems that can be efficiently solved using the transformed-based approach. Therefore it is difficult to have an idea of what properties the input polynomial system must satisfy in order for the approach to give reasonable results.

The number of variables considered in the experiments (n=2,3,4,5) is way to low to give an idea about the complexity of the problem.  For this number of variables it is not clear whether the machine learning method provides any advantage at all with respect to traditional computational algebraic approaches.

The theoretical part of the paper is much more concentrated on the issue of random generation of Gröbner bases than on the learning part. For this reason, it is my impression that the authors are putting too much emphasis on the part of the paper where results are not very satisfactory (the learning theory part). The paper would be more solid if concentrated on random generation of Gröbner bases with the learning part as an interesting application.

**Questions:**

1) What is the largest number of variables for which the machine learning method gave interesting results?

2) How does the accuracy of the model decay with the number of variables?

3) What is the relation between the number of variables and the time necessary to train the model?

**Limitations:**

Yes.

---

> ### Author Rebuttal · Authors · 2024-08-06
>
> We sincerely appreciate your thorough review and insightful comments. You raise the characterization of a subclass of polynomial systems as the main concern, and our rebuttal mostly focuses on this point. We would like you to refer to the Global Response as well.
>
> ### **On Weaknesses**
>
> **Characterization of a subclass of polynomial systems.**
> We appreciate your fundamental remark. We acknowledge that our paper does not provide a comprehensive characterization of the subclass of polynomial systems that can be efficiently solved using our transformer-based approach. Generally speaking, it is difficult to answer what problems can be solved by Transformers or what sample distributions are learnable by Transformers.
>
> Instead, our study currently approaches the question: for what class of polynomial systems can we prepare a dataset efficiently for Transformer training? The trainablity in this sense should come before the learnability from a practical point of view. As an initial work, we break down the trainability question into two components: Gröbner basis sampling and backward Gröbner problem. These can be answered affirmatively through the design of algorithms. For the former, we suggest ideals in shape position, Cauchy module, and potentially, ideals of points (cf. Appendix~G). For the latter, we suggest 0-dimensional ideals (Theorem 4.7).
>
> **Number of variables**
> This is also an important point. Ultimately, we envision that Transformers will be used for large-scale problems (i.e., problems with many variables and equations) that mathematical algorithms cannot address.
>
> However, the number of variables that Transformer can handle is currently restrictive. The restriction comes from the current simple tokenization of polynomial systems: for example, {$x^2-10, y$} is tokenized as $\texttt{[x, **, 2, +, -10, <sep>, y]}$, (cf. Section 5). Eventually, a single $n$-variate term requires $O(n)$ tokens, leading to a long input sequence for large $n$. As well known in Transformer literature, the space complexity of the attention mechanism scales in $O(L^2)$, where $L$ denotes the input length.
>
> The scope of this study is to establish the problem of learning to compute Gröbner bases, and thus, we tested its learnability based on a standard model, including tokenization. For scale-up, one may be able to embed a term by its coefficient (i.e., single token) and inject the exponent part as a position vector because this is literally a position in the term space. We are currently working on it in our follow-up paper.
>
> It is also worth noting that even for $n=5$, we have observed a potential advantage of using machine learning. Table 22 shows that there are several instances in which Transformer models successfully compute Gröbner bases in less than a second, whereas algebraic methods take much longer (100 seconds and timeout for most cases).
>
> **Balance between theoretical and learning parts.** Thank you for your fair evaluation and great writing suggestions. We indeed have a great algebraic interest in random generation of (non-Gröbner basis, Gröbner basis pair). We adopted the current structure by taking into account the potential readers (i.e., NeurIPS readers), who may be more interested in the learning part. We will reconsider the presentation. Thank you very much.
>
> ### **On Questions**
> We appreciate your insightful questions. We believe the learning approach of Gröbner bases has an advantage over algebraic algorithms for large-scale problems. However, as described above, the training on large $n$ requires an efficient input embedding of polynomial systems. We could also try larger $n$ by using a few more GPUs, but we considered this to be very interesting. It should be more reasonable to design an efficient input embedding for polynomial systems first and then see the impact of the increase in $n$ on the learning complexity. Here, we only provide quick answers to your questions below.
>
> **Largest number of variables with interesting results:**
> We have only tested up to $n=5$. Even at this point, we have already observed interesting results, such as the contrast between infinite and finite fields (Table 2) and Transformer supremacy (Table 22).
>
> **Accuracy decay with the number of variables:**
> The accuracy change with respect to the number of variables is given in Table 2, but the results would also change the density parameter $\sigma$. This is also a consequence of the input length restriction.
>
> **Relation between the number of variables and training time.** We collected the training time from our experiment log and summarized it below. Note that the training time here can have been affected by the other processes running in parallel, so these numbers can be regarded as upper bounds. The table shows that we generally need longer training time for larger $n$. Not shown here, but we also tried longer training but we only observed a subtle improvement.
>
> |  | $n=2$ | $n=3$ | $n=4$ | $n=5$ |
> | -------- | -------- | -------- | -------- |-------- |
> | $\mathbb{Q}[x_1,\ldots, x_n]$     | 5.8     | 8.4      | 9.8     | 12.5    |
> | $\mathbb{F}_7[x_1,\ldots, x_n]$   | 6.2     | 12.6     | 8.6     | 9.5     |
> | $\mathbb{F}_{31}[x_1,\ldots, x_n]$| 7.3     | 10.4     | 10.6    | 11.8    |
> *The training time in hours.
>
> Thanks to the questions, we realize that there are two interleaving factors that affect the complexity of learning. For larger $n$, the Gröbner basis computation becomes algebraically more difficult. At the same time, dataset generation algorithms generate larger systems (i.e., longer input/target sequences), making learning more difficult from a machine learning perspective. We have to find a way to separate these factors for future discussion.

---

> > ### Comment · Reviewer_YTLb · 2024-08-13
> >
> > Thanks for the clarifications. I will keep my current recommendation.

---

### Official Review · Reviewer_BQDD · 2024-07-14

**Soundness:** 3
**Presentation:** 3
**Contribution:** 3
**Rating:** 5
**Confidence:** 3

**Summary:**

This paper presents a Transformer based method to compute Gröbner basis, a known NP-hard problem.  The authors focus on polynomials with 0-dimensional radical ideals and propose efficient algorithms to generate training samples of polynomial systems and their corresponding Gröbner basis. The main novelties include (1) an efficient backward transformation algorithm from a Gröbner basis to an associated non-Gröbner set; (2) a hybrid input embedding for both discrete and continuous-value tokens. The transformer, which was trained on millions of Gröbner basis pairs generated by the authors, performed well on some types of rings.

**Strengths:**

The paper introduced a novel method to compute a significant problem in computational algebra, demonstrating high accuracy for certain types of rings. The efficient backward sampling methods are less explored and may facilitate further ML studies in this field. Additionally, this backward approach helps restrict the Gröbner basis to applications of interest. The paper is well-written and easy to follow.

**Weaknesses:**

* I have serval questions about the choice of the 0-dimensional ideal as the scope of the study :
(1) the authors first mentioned “…, and thus, we should focus on a particular class of ideals and pursue in-distribution accuracy” and then claimed “… and thus, we focus on the generic case and leave the specialization to future work”.  Do you regard this choice as a “particular” or “general” case of the Gröbner basis computation?
(2) What is motivation of focusing on 0-dimensional ideals (since it is already mentioned that this work is meant to be general)? Is it purely because of the sampling easiness? Also, what are the difficulties of working on the more general Gröbner basis computation problem?
(3) The discussion of, whether Transformers, or more generally, ML methods, can help NP-hard problems generally or only in-distribution, is interesting. There are some results from the SAT field that end-to-end ML models may be able to generalize to out-of-distribution problems [1]. As you claim here “in-distribution accuracy” is what we are after, it is important to include some experiments across different applications, to support this claim.

* The hybrid input embedding for both discrete and continuous-value tokens is claimed to be one of the paper’s contributions. However, this idea is not new, as other research has explored similar problems (e.g.,  [2, 3, 4]). The authors do not provide a review of existing methods or distinguish their approach from these works.

It is interesting to note that many incorrect results are reasonable. This suggests that incorporating the Transformer method with planning (5) or RL (6) may help the performance through feedbacks.


Reference
[1] Cameron, Chris, et al. "Predicting propositional satisfiability via end-to-end learning." Proceedings of the AAAI Conference on Artificial Intelligence. Vol. 34. No. 04. 2020.
[2] Charton, François. "Linear algebra with transformers." arXiv preprint arXiv:2112.01898 (2021).
[3] Golkar, Siavash, et al. "xval: A continuous number encoding for large language models." arXiv preprint arXiv:2310.02989 (2023).
[4] McLeish, Sean, et al. "Transformers Can Do Arithmetic with the Right Embeddings." arXiv preprint arXiv:2405.17399 (2024).
[5] Kamienny, Pierre-Alexandre, et al. "Deep generative symbolic regression with Monte-Carlo-tree-search." International Conference on Machine Learning. PMLR, 2023.
[6] Jha, Piyush, et al. "RLSF: Reinforcement Learning via Symbolic Feedback." arXiv preprint arXiv:2405.16661 (2024).

**Questions:**

* With the proposed sampling methods based on shape position and Cauchy module, how diverse is this generated set? Does these generating methods further restrict the scope less than the 0-dimensional radical ideals?

**Limitations:**

It is not clear whether this work can be generalized to beyond 0-dimensional ideals.

---

> ### Author Rebuttal · Authors · 2024-08-06
>
> We appreciate your insightful and constructive comments and thorough review, as well as many pointers to related papers.
>
> ### **On Weaknesses**
>
> **Choice of 0-dimensional ideals.**
> We received several comments/questions about this assumption from several reviewers; please refer to the Global Response for the motivation for the choice.
>
>
> **Particular v.s. generic.**
> You seem to find contradictive policies from our claims:
> > [l.178]
> > ... focus on a particular class of ideals..."
> >
> > [l.183]
> > ..., the form of non-Gröbner sets varies across applications, and thus, we focus on the generic case and leave the specialization to future work.
>
> The former claims the need to focus on a particular class of ideals as it is difficult to cover all the cases in a single work. The latter says that our way of particularization is more driven by a computer-algebraic viewpoint and not biased toward a single application. We will update our manuscript to avoid such confusion.
>
>
> **Hybrid input embedding.**
> Thank you for completing our literature survey! In our reading, [2,4] rely on traditional discrete embedding based on mantissa (between 0 and 9999) and exponent, which requires keeping a large vocabulary and learning the numbers' relation from scratch. We found [3; Golkar+, Oct. 2023] to be close to our idea. The difference is that in their work, a real value is represented as the length of the embedding vector of the number token, while we use an MLP to find an embedding directly and have more degree of freedom (i.e., length and direction). Interestingly, Figure 1 shows that our embedding also encodes the values by the scale. We need experiments of Gröbner basis computation learning to check if their method outperforms ours and also discrete embedding. We will review and include them in our references.
>
> **In-distribution accuracy.**
> We appreciate your pointer to an interesting work. It seems that the paper (particularly Table 2) indicates a potential generalization even for NP-hard problems. It is also very surprising to me that models trained on a 100-variable dataset perform better in a 600-variable dataset than in a 100-variable dataset. We are not sure if this is called generalization (although it is great).
> > As you claim here "in-distribution accuracy" is what we are after, it is important to include some experiments across different applications, to support this claim.
>
> We don't claim that a generalization of learning is unachievable for NP-hard problems; we only present our picture of using Transformer for Gröbner basis computation. The learning approach may provide faster computation than classical algorithms, but the problem itself has been proven NP-hard, so machine learning models cannot break this.
>
> ### **On Questions**
> **Dataset diversity.**
> Table 6 provides several statistics of the datasets (we will make it cleaner in the update). The degrees and the number of terms have certain variances, which empirically shows diversity. Tables 4 and 22 also present the existence of hard samples for mathematical algorithms, which supports a certain diversity of the generated samples.
>
> Theoretically, defining and measuring the diversity of generators is not very clear. For Gröbner basis sampling, we performed uniform sampling for degree, the number of terms, and so on, so the Gröbner bases were sampled uniformly randomly in a sense. For non-Gröbner sets computed from the Gröbner bases, a reasonable starting point would be the question on the reachability: can we sample all possible generating set $F$ such that $\langle F \rangle = \langle G \rangle$ from a Gröbner basis $G$? As of now, we suspect the answer is negative because we are based on Theorem 4.2 (2), a sufficient condition. We are now tackling designing an efficient algorithm based on Theorem 4.2 (3), a necessary and sufficient condition, but this requires a deeply algebraic discussion.
>
> **Generalization to the positive-dimensional case.**
> Please kindly refer to the Global Response.

---

### Author Rebuttal · Authors · 2024-08-06

# Global Response

We sincerely appreciate the reviewers' time and efforts in reviewing our manuscript. We have received many insightful comments and questions, which have been carefully considered in this rebuttal and the next manuscript update.

While we will answer most of the comments and questions from each reviewer individually, here we would like to elaborate on our assumption of 0-dimensional ideals, as this topic appears to be of interest to several reviewers (particularly Reviewers BQDD and q3D1).

### **On 0-dimensional ideal settings**
This study focuses on 0-dimensional ideals, and several reviewers asked about its motivation and generalization to positive-dimensional ideals. A short answer is that the choice of 0-dimensional ideals is reasonable from both computational algebra and application standpoints, and generalization to the positive-dimensional case is challenging but interesting future work.

### **Motivation**
Our study is the first to approach Gröbner basis computation via end-to-end learning. We found that dataset generation poses unexplored algebraic tasks, i.e., sampling of Gröbner bases and their back-transformation to non-Gröbner sets. It is very difficult to resolve this in the most general case in a single work, so we naturally had to restrict ourselves to a particular case. We chose 0-dimensional ideals for two reasons:

**i)** From a computer algebraic perspective, 0-dimensional ideals are the most popular class of ideals to study. This is partly because of the ease of analysis. As Definition A.5 shows, 0-dimensional ideals relate to finite-dimensional vector spaces, and thus, analysis and algorithm design can be essentially addressed by matrices and linear algebra. As a consequence, we have more useful statements and algorithms for this case. As such, we focused on the 0-dimensional case as a reasonable starting point, where many facts and tools are accessible. It is also worth noting that on a finite field, any ideal becomes 0-dimensional by including the field equations (i.e., polynomials restricting the solutions to the finite field; e.g., $x(x-1)$ for $\mathbb{F}_2[x]$) to the generators. Such extensions of generators are perfectly acceptable when we are interested in the solutions, not the Gröbner bases themselves.

**ii)** From an application perspective, 0-dimensional ideals are again popular objects to study. Namely, many applications share a motivation of finding solutions for polynomial systems with an implicit assumption of finitely many solutions. If a system (or its associated ideal) is 0-dimensional, it has finitely many solutions (and the reverse is also true if the coefficient field is algebraically closed). For example, the multivariate cryptosystem, a promising candidate for signature-based post-quantum cryptosystem standardization by NIST, is based on 0-dimensional ideals [1]. In control theory, finding equilibria of rational dynamical systems is reduced to polynomial system solving with an assumption that there are finite equilibria. The estimation of the domain of attraction around an equilibrium is also reduced to polynomial system solving [2]. In machine learning, the computation of generators of 0-dimensional ideals has been studied as a feature extraction method [3, 4, 5] (see Appendix G for more), including an ICML'13 best paper.

### **Generalization**
Generalization to positive-dimensional ideals is non-trivial, and we leave it to future work. One of our contributions to the community is posing this new open problem with motivation from machine learning. We currently have no particular idea to address this. Perhaps we might be able to design some constructive algorithms for binomials, which appear in applications to combinatorics and algebraic statistics.


[1] T. Yasuda, X. Dahan, Y.-J. Huang, T. Takagi, and K. Sakurai. MQ challenge: hardness evaluation of solving multivariate quadratic problems. Cryptology ePrint Archive, 2015.

[2] D. Nešić, I.M.Y. Mareels, T. Glad, M. Jirstrand, The Gröbner Basis Method in Control Design: an Overview, IFAC, 2002,

[3] R. Livni, D. Lehavi, S. Schein, H. Nachliely, S. Shalev-Shwartz, and A. Globerson. Vanishing component analysis. ICML, 2013. (Best Paper Award)

[4] H. Kera and Y. Hasegawa. Gradient boosts the approximate vanishing ideal. AAAI, 2020

[5] E. S. Wirth and S. Pokutta. Conditional gradients for the approximately vanishing ideal. AISTATS, 2022

---

### Decision · Program_Chairs · 2024-09-25

**Decision:**

Accept (poster)

**Comment:**

In computational algebra, a crucial problem with various applications is the computation of a Gröbner basis of a polynomial system. The computational barrier of this NP-hard problem is the doubly exponential time complexity in the number of variables. In order to overcome this barrier, this paper focuses on the class of zero-dimensional ideals and proposes a Transformer-based approach to compute Gröbner bases. There are two key challenges behind this learning approach: generating a sample set of labeled examples and tokenizing polynomials on infinite fields. These challenges are addressed by providing a computationally efficient solution to the “backward” Gröbner problem and a continuous embedding scheme. Experiments conducted on polynomial systems of different fields validate the effectiveness of this approach.

Based on the reviewers’ comments and the discussion with the authors, the strengths of this paper outweigh its weaknesses. Notably, most reviewers agree on the novelty of this approach. From a theoretical viewpoint, the sampling method obtained by solving the backward Gröbner problem is elegant. From an empirical viewpoint, the experimental evaluation indicating how transformers can learn to compute Gröbner bases is also interesting.

However, several concerns have been raised by reviewers, and I strongly encourage the authors to revise the paper accordingly. Notably:
* Presentation: significant effort has been spent on providing a manuscript that is accessible to the machine learning community by providing clear notation and a smooth flow of algebraic concepts. Yet, it is highly likely that such a study will be read by algebraists. Therefore, some key notions about Transformers should be included in the paper, for example, in Section 5, before embarking on tokenization issues.
* Motivation: regarding the choice of zero-dimensional ideals, the global authors' response is insightful and some arguments in this response should be included in the paper to clarify the choice.
* Positioning: as the key theoretical result of this paper lies in an efficient way of generating sample sets for supervised learning of Gröbner bases, it might be relevant to re-organize the paper according to this fundamental result. The empirical evaluation with Transformers is also interesting, but the current state-of-the-art in computational learning theory does not provide probabilistic guarantees of the sample/computational complexities of these models.
* Scalability: a discussion about the fact that, for the moment, datasets include only a small number of variables should be included in the experimental section. The authors' clarification of this issue in the rebuttal could be used for this purpose.
* Applicability: The above scalability issue naturally raises the question of whether such a transformer-based approach can be targeted to practical applications (e.g., in Biology or Cryptography). A discussion on this question could be incorporated in the conclusion.